# Rewiring of the ubiquitinated proteome determines ageing in *C. elegans*

Seda Koyuncu[1], Rute Loureiro[1], Hyun Ju Lee[1], Prerana Wagle[1], Marcus Krueger[1,2] & David Vilchez[1,2,3 ✉]

Ageing is driven by a loss of cellular integrity[1]. Given the major role of ubiquitin modifications in cell function[2], here we assess the link between ubiquitination and ageing by quantifying whole-proteome ubiquitin signatures in *Caenorhabditis elegans*. We find a remodelling of the ubiquitinated proteome during ageing, which is ameliorated by longevity paradigms such as dietary restriction and reduced insulin signalling. Notably, ageing causes a global loss of ubiquitination that is triggered by increased deubiquitinase activity. Because ubiquitination can tag proteins for recognition by the proteasome[3], a fundamental question is whether deficits in targeted degradation influence longevity. By integrating data from worms with a defective proteasome, we identify proteasomal targets that accumulate with age owing to decreased ubiquitination and subsequent degradation. Lowering the levels of age-dysregulated proteasome targets prolongs longevity, whereas preventing their degradation shortens lifespan. Among the proteasomal targets, we find the IFB-2 intermediate filament[4] and the EPS-8 modulator of RAC signalling[5]. While increased levels of IFB-2 promote the loss of intestinal integrity and bacterial colonization, upregulation of EPS-8 hyperactivates RAC in muscle and neurons, and leads to alterations in the actin cytoskeleton and protein kinase JNK. In summary, age-related changes in targeted degradation of structural and regulatory proteins across tissues determine longevity.

Ageing can be delayed by evolutionary conserved pathways such as dietary restriction and reduced insulin/insulin-like growth factor (IGF-1) signalling[1]. The attachment of the small protein ubiquitin to lysine residues of specific targets is a central pathway by which cellular decisions are made[2], but the effect of ubiquitination in ageing remains unclear. Numerous proteins are ubiquitinated in a dynamic and tightly regulated manner through a sequential mechanism that involves E3 ubiquitin ligases, which are responsible for substrate selection[2]. However, deubiquitinating enzymes (DUBs) can reverse this process[6].

The ubiquitination cascade also links additional molecules to the internal lysine sites of the primary ubiquitin. Ubiquitin has seven lysine residues, all of which can form polyubiquitin chains. A Lys48-linked polyubiquitin chain is the primary signal for degradation by the proteasome—the main selective proteolytic system in eukaryotic cells[2,3]. As such, the ubiquitin–proteasome system regulates the levels of several structural and short-lived regulatory proteins. Here we asked whether ageing modifies the ubiquitination and targeted degradation of regulatory proteins that, in turn, could actively influence longevity. To this end, we used an antibody that recognizes di-glycine moieties linked by an isopeptide bond to lysine sites of proteins[7]. These epitopes, which constitute remnants of ubiquitination followed by tryptic digestion, can be engaged in biochemical isolation and mass spectrometry to provide site-specific information and quantification of ubiquitin modifications across the proteome[7].

## Ub-proteome remodelling during ageing

The median lifespan of wild-type *C. elegans* is 19 days[8]. Thus, we compared the ubiquitin (Ub)-modified proteome of worms at the first day of adulthood with young (day 5), mid-age (day 10) and aged adults (day 15). Moreover, we assessed age-matched long-lived genetic models of dietary restriction (*eat-2(ad1116)*) and reduced insulin/IGF-1 signalling (*daf-2(e1370)*) (Fig. 1a). With high reproducibility between biological replicates, our assay identified ubiquitination sites for 3,373 peptides that correspond to 1,485 distinct proteins (Extended Data Fig. 1a, Supplementary Table 1). The levels of multiple Ub-peptides changed with age in wild-type and long-lived mutant worms compared with their respective day-1 adults (Fig. 1b, Supplementary Table 1). In wild-type worms, the total number of differentially abundant Ub-peptides increased after day 5. Most of these changes were linked to downregulated ubiquitination levels. By contrast, long-lived mutants had fewer downregulated Ub-peptides during ageing. In fact, *daf-2* worms had an increased number of upregulated Ub-peptides with age (Fig. 1c, d). Given that *C. elegans* undergoes a widespread proteome remodelling during ageing[9,10], we quantified the total amounts of individual proteins to compare with Ub-peptides (Supplementary Table 2). In many instances, differences in ubiquitination levels could not be simply ascribed to a similar change in the protein amounts (Extended Data Fig. 1b–d, Supplementary Table 3).

[1]Cologne Excellence Cluster for Cellular Stress Responses in Aging-Associated Diseases (CECAD), University of Cologne, Cologne, Germany. [2]Center for Molecular Medicine Cologne (CMMC), University of Cologne, Cologne, Germany. [3]Faculty of Medicine, University Hospital Cologne, Cologne, Germany. ✉e-mail: dvilchez@uni-koeln.de

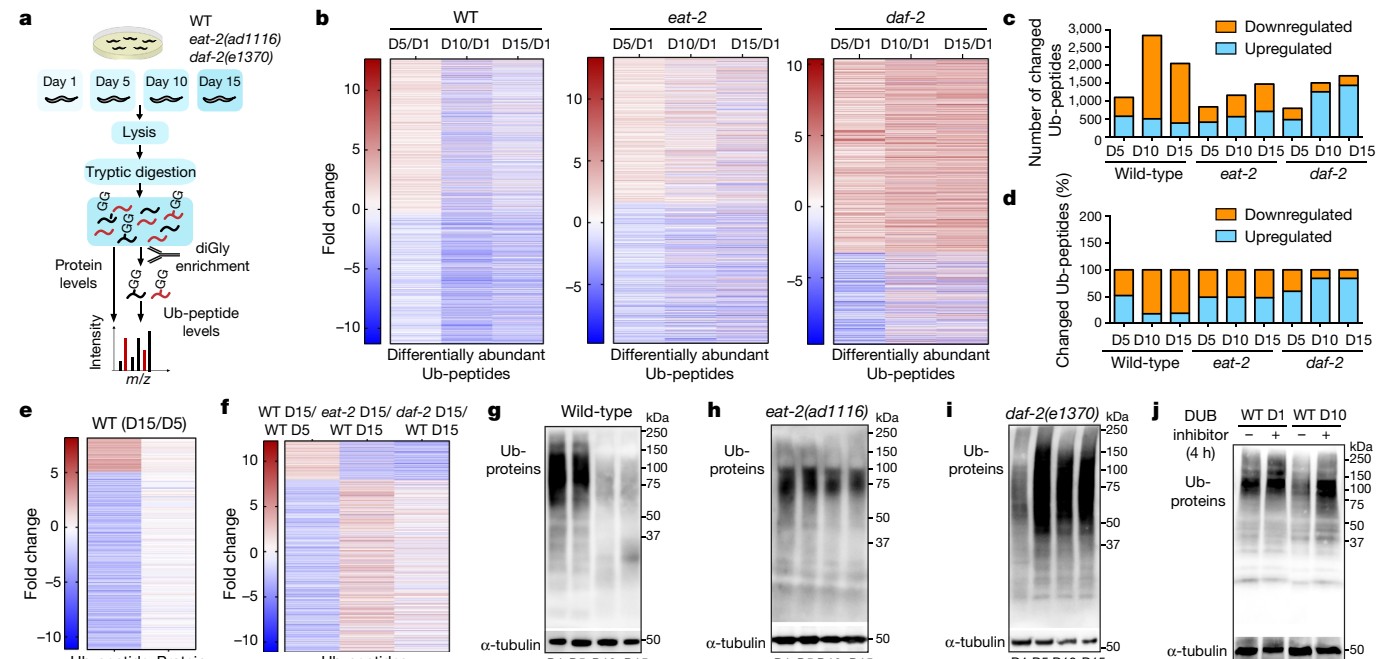

**Fig. 1 | Rewiring of the Ub-proteome with age. a**, Scheme of ubiquitin proteomics by di-glycine (diGly) peptide enrichment in wild-type (WT) and long-lived mutant worms. **b**, Heat maps representing log$_2$-transformed fold changes in Ub-peptide levels at different days (D) of adulthood compared with the corresponding day-1 adult strain. For each strain, only Ub-peptides significantly changed in at least one age are shown. **c**, Number of significantly downregulated and upregulated Ub-peptides compared with the respective day-1 adult strain. **d**, Percentage of downregulated and upregulated Ub-peptides among the total number of significantly changed Ub-peptides per condition. **e**, The log$_2$-transformed fold changes of differentially abundant Ub-peptides and their corresponding total protein levels comparing day-15 and day-5 wild-type worms. **f**, The log$_2$-transformed fold changes of differentially abundant Ub-peptides in day-15 wild-type worms and comparison with age-matched *eat-2* and *daf-2* mutants. In **b**–**f**, n = 4; two-sided *t*-test, false discovery rate (FDR) < 0.05 was considered significant. **g**–**i**, Immunoblot of Ub-proteins in wild-type (**g**), *eat-2(ad1116)* (**h**) and *daf-2(e1370)* (**i**) worms at different days of adulthood. α-tubulin was used as a loading control. The images are representative of four independent experiments. **j**, Immunoblot of Ub-proteins in wild-type worms treated with 13.7 µg ml$^{-1}$ PR-619 (broad-spectrum DUB inhibitor) or vehicle control (dimethyl sulfoxide, DMSO) for 4 h before lysis. Representative of three independent experiments. For gel source data, see Supplementary Fig. 1.

Because we detected a higher number of ubiquitination changes after day 5, we directly compared day-5 and day-15 wild-type worms. In aged worms, 350 Ub-peptides were upregulated, whereas 1,813 Ub-peptides were downregulated, which supports the idea that ageing is particularly associated with a loss of ubiquitination (Fig. 1e, Supplementary Table 4). Only 123 upregulated and 582 downregulated Ub-peptides correlated with a change in the total protein levels in the same direction. The other differentially abundant Ub-peptides were inversely correlated with protein levels or corresponded to proteins that did not change in abundance with age (Fig. 1e, Extended Data Fig. 1e, f, Supplementary Table 5). The amounts of Ub-peptides can be influenced by transcription, translation and proteolysis, which are dysregulated during ageing and rescued by longevity paradigms[1,3,11,12]. Notably, pro-longevity pathways also prevented age-related changes in ubiquitination (Fig. 1f, Extended Data Fig. 1g, Supplementary Table 6).

## DUBs diminish ubiquitination with age

To further assess age-related changes in ubiquitination, we performed western blot analysis. Wild-type worms exhibited a global decrease in levels of Ub-protein after day 8 of adulthood (Fig. 1g, Extended Data Fig. 2a). However, Ub-protein levels remained similar in *eat-2* mutants during ageing, and increased in *daf-2* mutants after day 1 (Fig. 1h, i, Extended Data Fig. 2b, c). In contrast to their global downregulated ubiquitination, aged wild-type worms expressed higher or similar levels of ubiquitin-encoding genes (*ubq-1*, *ubq-2* and *ubl-1*) compared with either young wild-type worms or age-matched long-lived mutants. Similarly, ageing and longevity paradigms did not change

the expression of *usp-14*, a gene induced by ubiquitin deficiency[13] (Extended Data Fig. 2d–g). Moreover, aged wild-type worms did not exhibit decreased total levels or half-life of protein ubiquitin itself (Extended Data Fig. 2h–j).

E3 ligases and DUBs directly modulate ubiquitination levels. In *C. elegans*, there are more than 170 E3 ligases[14] but we only found significant changes in the expression of 12 E3s with age. By contrast, there are 45 DUBs in *C. elegans*[6], but a higher proportion (14 DUBs) were upregulated in aged wild-type worms (Extended Data Fig. 3a, b, Supplementary Table 7). Dietary restriction rescued the levels of CSN-6, a component of the COP9 signalosome that removes ubiquitin-like proteins from cullin E3 complexes, inhibiting their activity and the subsequent ubiquitination of proteasomal targets[15]. In addition to CSN-6, reduced insulin/IGF-1 signalling prevented the upregulation of most of the age-dysregulated DUBs (Extended Data Fig. 3b). Knockdown of distinct age-dysregulated DUBs ameliorated loss of ubiquitination during ageing, including *csn-6* (which encodes the worm orthologue of human COPS6), *H34C03.2* (USP4), *F07A11.4* (USP19), *math-33* (USP7), *usp-5* (USP5), *usp-48* (USP48) and *otub-3* (OTUD6A) (Extended Data Fig. 3c–g). To examine whether increased DUB activity underlies the age-associated decline in ubiquitination, we used the broad-spectrum DUB inhibitor PR-619 (ref. [16]). Notably, treatment with the DUB inhibitor in old worms was sufficient to rescue low ubiquitination levels and extend lifespan (Fig. 1j, Extended Data Fig. 3h).

## Impaired targeted degradation with age

Given the high number of downregulated Ub-peptides during ageing, a subset of these events could reduce selective degradation by the

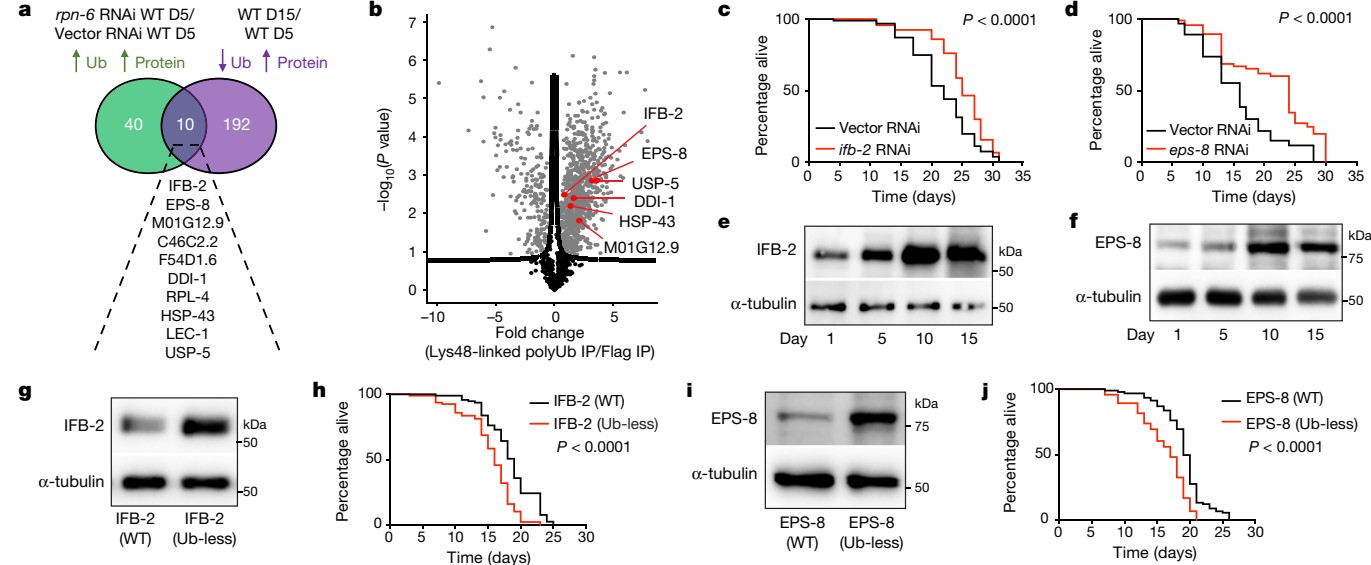

**Fig. 2 | Age-related deubiquitination impairs targeted degradation of longevity regulators. a**, Ten Ub-proteins increase after *rpn-6* RNAi treatment in young wild-type adults (*rpn-6* RNAi/Vector RNAi) and become less ubiquitinated but more abundant with age (day 15/day 5). **b**, Volcano plot of proteins containing Lys48-linked polyUb in day-5 adult wild-type worms (*n* = 3, FDR < 0.05). The $-\log_{10}(P \text{ value})$ of a two-sided *t*-test is plotted against the $\log_2$-transformed fold change values from immunoprecipitation (IP) with an antibody against Lys48-linked polyUb compared with a control anti-Flag antibody. Red dots indicate age-dysregulated proteasome targets. **c**, **d**, Knockdown of either *ifb-2* (**c**) or *eps-8* (**d**) after development extends lifespan (*P* < 0.0001). **e**, Western blot analysis with an antibody against IFB-2 of wild-type worms at different days of adulthood. α-tubulin is the loading control. Representative of four independent experiments. **f**, Western blot analysis with an antibody against EPS-8 of wild-type worms. Representative of

three independent experiments. **g**, Western blot analysis with an antibody against IFB-2 of wild-type and IFB-2(K255R/K341R) (Ub-less) mutant worms at day 2 of adulthood. Representative of three independent experiments. **h**, Ubiquitin-less IFB-2 mutant worms have a shorter lifespan than wild-type worms (*P* < 0.0001). **i**, Western blot analysis with an antibody against EPS-8 of worms expressing endogenous wild-type EPS-8::HA or EPS-8(K524R/K583R/K621R::HA) (Ub-less) at day 1 of adulthood. Representative of three independent experiments. **j**, EPS-8 (Ub-less) mutants are short-lived (*P* < 0.0001). In lifespan experiments, *P* values were determined by two-sided log-rank test; *n* = 96 worms per condition. Lifespan data are representative of at least two independent experiments. Supplementary Table 11 contains replicate data of independent lifespan experiments. For gel source data, see Supplementary Fig. 1.

proteasome. We found that 192 proteins were less ubiquitinated in at least one of their lysine sites during ageing, whereas the total levels of the protein increased (Fig. 2a, Extended Data Fig. 1e). If these proteins are age-dysregulated proteasomal targets, defects in proteasome activity could diminish their degradation in young worms. To decrease proteasome function, we knocked down the *rpn-6* proteasome subunit[17,18]. Loss of proteasome activity resulted in widespread changes in the proteome of day-5 young adults (Extended Data Fig. 4a, Supplementary Table 8). In addition to potential indirect effects caused by proteasome dysfunction, upregulated proteins could include direct proteasome targets, particularly if they also have increased Ub-peptide levels. We identified 40 proteins that exhibit an increase in both their total and Ub-peptide levels after *rpn-6* RNA interference (RNAi) in young adults. By integrating data from untreated aged worms, we found that 10 proteasome-modulated proteins became more abundant with age, and at least one of their lysine sites was less ubiquitinated (Fig. 2a, Supplementary Table 8). The age-dysregulated proteasome targets were IFB-2, EPS-8, RPL-4, M01G12.9, C46C2.2, F54D1.6, DDI-1, LEC-1, HSP-43 and USP-5. Notably, ageing or *rpn-6* knockdown did not increase the mRNA levels of most of these targets, including IFB-2 and EPS-8 (Extended Data Fig. 4b, c).

Several age-dysregulated proteasome targets such as EPS-8 contained Lys48-linked polyUb chains for proteasomal recognition. By contrast, they did not have Lys63-linked polyUb, with the exception of LEC-1 and IFB-2 that contained both Lys63 and Lys48 ubiquitin linkages (Fig. 2b, Extended Data Fig. 4d, Supplementary Table 9). Together, our data indicate that the age-related deubiquitination of distinct proteins reduces their recognition and subsequent degradation by the proteasome.

## Proteasome targets determine lifespan

Age-dysregulated proteasome targets such as EPS-8, IFB-2, RPL-4 and F54D1.6 are required for normal development[19–23]. Whereas these factors endow benefits early in life, we asked whether their age-associated upregulation is detrimental for adult lifespan. We hypothesized that if age-dysregulated proteasome targets are not essential for adult viability, lowering their levels after development could prolong longevity. Both the chaperone *hsp-43* and *usp-5* are essential for cell viability[24,25] and their knockdown in adult worms shortened lifespan (Extended Data Fig. 4e). By contrast, single knockdown of *ifb-2*, *eps-8*, *rpl-4*, *M01G12.9*, *C46C2.2* or *F54D1.6* during adulthood was sufficient to extend lifespan (Fig. 2c, d, Extended Data Fig. 4e).

Intrigued by the robust effects of IFB-2 and EPS-8 on lifespan, we confirmed by western blot that their protein levels increase with age (Fig. 2e, f, Extended Data Fig. 4f–h). We then asked whether the age-associated deubiquitination that affects proteins such as IFB-2 and EPS-8 is conserved across tissues. Using a worm tissue expression prediction tool[26], we classified proteins with ubiquitination changes according to their tissue expression. The bioinformatic analysis indicated that tissues such as the germline, muscle, intestine, epidermis or neurons express several proteins that contain downregulated Ub-peptides with age (Extended Data Fig. 5a, Supplementary Table 10). Similar to lysates from whole worms, the global amounts of Ub-proteins decreased in isolated germlines, intestines and heads during ageing, which indicates that the loss of ubiquitination occurs throughout the organism (Extended Data Fig. 5b). The amounts of the ubiquitously expressed EPS-8 protein[20] were increased in different somatic tissues with age (Extended Data Fig. 5c), which correlates with global deubiquitination across tissues.

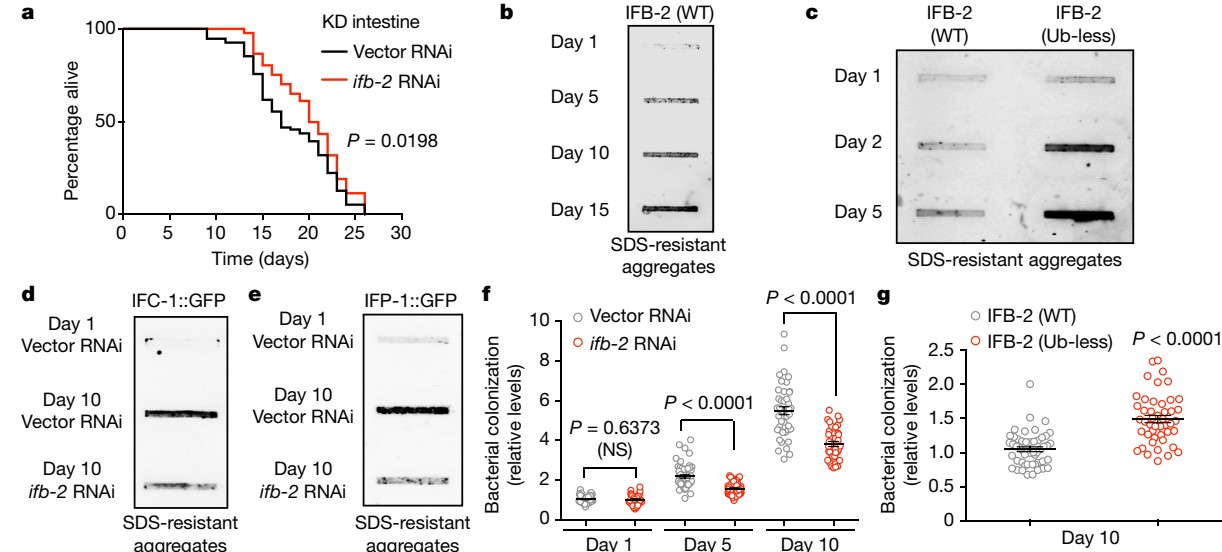

**Fig. 3 | Increased IFB-2 levels induce age-related intestinal alterations.** **a**, Intestinal-specific knockdown (KD) of *ifb-2* extends lifespan. *P* value determined by two-sided log-rank test; *n* = 96 worms per condition. Lifespan data are representative of two independent experiments. Supplementary Table 11 contains replicate data of independent experiments. **b**, Filter trap analysis with an antibody against IFB-2 of wild-type worms at different ages. Representative of eight independent experiments. **c**, Filter trap analysis with an antibody against IFB-2 of wild-type and IFB-2 (Ub-less) mutant worms. Representative of four independent experiments. **d**, **e**, Filter trap experiments with an antibody against GFP of worms expressing IFC-1::GFP under the *ifc-1* promoter (**d**) or IFP-1::GFP under the *ifp-1* promoter (**e**). Representative of two independent experiments. **f**, Quantification of bacterial colonization. Fluorescence of mCherry-expressing *E. coli* within the intestine relative to day 1 (D1) Vector RNAi. Data are mean ± s.e.m. D1 Vector RNAi, *n* = 56 worms from 3 independent experiments; D1 *ifb-2* RNAi, *n* = 35; D5 Vector RNAi, *n* = 53; D5 *ifb-2* RNAi, *n* = 55; D10 Vector RNAi, *n* = 45; D10 *ifb-2* RNAi, *n* = 39. **g**, Bacterial colonization relative to day-10 adult wild-type worms. Data are mean ± s.e.m. Wild-type, *n* = 50 worms from 3 independent experiments; Ub-less IFB-2, *n* = 47. In **f** and **g**, *P* values were determined by two-sided *t*-test. NS, not significant. In all experiments, RNAi was initiated at day 1 of adulthood.

Because the *ifb-2* gene is specifically expressed in intestinal cells[4], protein levels of IFB-2 were upregulated in the intestine of old worms but were not detectable in other tissues (Extended Data Fig. 5d).

Tissue-specific knockdown of *rpn-6* in the intestine, epidermis, neurons or muscle increased the amounts of EPS-8 in young worms, which indicates that the proteasome modulates EPS-8 in all of these tissues (Extended Data Fig. 5e–h). According to its intestinal-specific expression, only knockdown of *rpn-6* in the intestine upregulated IFB-2 levels (Extended Data Fig. 5e–h). Together, these experiments support the idea that the ubiquitin–proteasome system acts in a cell-autonomous manner to regulate IFB-2 and EPS-8. Besides intracellular regulation, inter-organ communication also influences organismal ageing. For example, neurons elicit signals that modulate the ageing of distal tissues[27]. To assess whether inter-organ communication influences global ubiquitination levels, we used *unc-13* mutant worms that are deficient in the release of neurotransmitters[28]. Whereas blocking neurotransmission did not affect ubiquitination levels in young worms, it exacerbated the age-associated decline in older worms (Extended Data Fig. 5i). Concomitantly, the amounts of EPS-8 and IFB-2 were further upregulated in aged *unc-13* mutants compared with wild-type worms, which suggests that cell non-autonomous mechanisms impinge on organismal ubiquitination levels (Extended Data Fig. 5j, k).

To confirm a direct link between loss of ubiquitination in IFB-2 and EPS-8 with the regulation of longevity, we blocked the ubiquitination of endogenous IFB-2 and EPS-8 by generating lysine (K) to arginine (R) mutants. Among the three ubiquitinated sites identified in IFB-2, Lys255 and Lys341 exhibited a pronounced deubiquitination during ageing (Supplementary Table 1). Notably, the K255R/K341R double mutation increased IFB-2 protein levels in young adult worms and shortened lifespan (Fig. 2g, h, Extended Data Fig. 6a). Thus, upregulation of IFB-2 levels was sufficient to decrease lifespan, as further supported by overexpression experiments (Extended Data Fig. 6b).

Within EPS-8, Lys524, Lys583 and Lys621 showed a robust deubiquitination in aged worms (Extended Data Fig. 1f). Ubiquitin-less mutations

did not result in loss of EPS-8 function, as EPS-8(K524R/K583R/K621R) mutants did not exhibit embryonic lethality or developmental arrest (Extended Data Fig. 6c, d). However, EPS-8(K524R/K583R/K621R) worms had upregulated EPS-8 protein levels at young adult stages, resulting in a short-lived phenotype (Fig. 2i, j). Notably, knockdown of *eps-8* after development was sufficient to rescue the short-lived phenotype of EPS-8(K524R/K583R/K621R) mutants (Extended Data Fig. 6e). Thus, the age-related deubiquitination and subsequent impaired degradation of IFB-2 and EPS-8 determine lifespan.

## Intestinal alteration by increased IFB-2

Because age-dysregulated proteasome targets have different roles, they could act in a tissue-specific manner. Indeed, RNAi against the intermediate filament *ifb-2* in the intestine alone was sufficient to extend longevity, whereas RNAi in other tissues did not affect lifespan (Fig. 3a, Extended Data Fig. 7a, b). IFB-2 and other interacting intermediate filament proteins form the intermediate filament-rich layer—an evolutionary conserved region in the apical cytoplasm of intestinal cells that is essential for intestinal morphogenesis and integrity[4]. During adulthood, pathological conditions impair the network of intestinal intermediate filaments such as IFB-2, which loses its typical enrichment in the apical region[4]. Because loss of intestinal integrity is also a characteristic of ageing[29], we examined the intracellular distribution of IFB-2. We found that ageing triggers the mislocalization of IFB-2 from the apical part to the rest of the cytoplasm and its accumulation into foci (Extended Data Fig. 7c, d). By filter trap experiments, we confirmed the aggregation of endogenous IFB-2 during ageing, a process accelerated by ubiquitin-less IFB-2 mutations (Fig. 3b, c, Extended Data Fig. 7e).

With age, other intestinal intermediate filaments (IFC-1, IFC-2 and IFP-1) and the intestinal filament organizer IFO-1 also lost their apical localization and accumulated into aggregates (Extended Data Fig. 7f–k). Notably, knockdown of *ifb-2* after development ameliorated these age-related changes (Fig. 3d, e, Extended Data Fig. 8a–c). Because

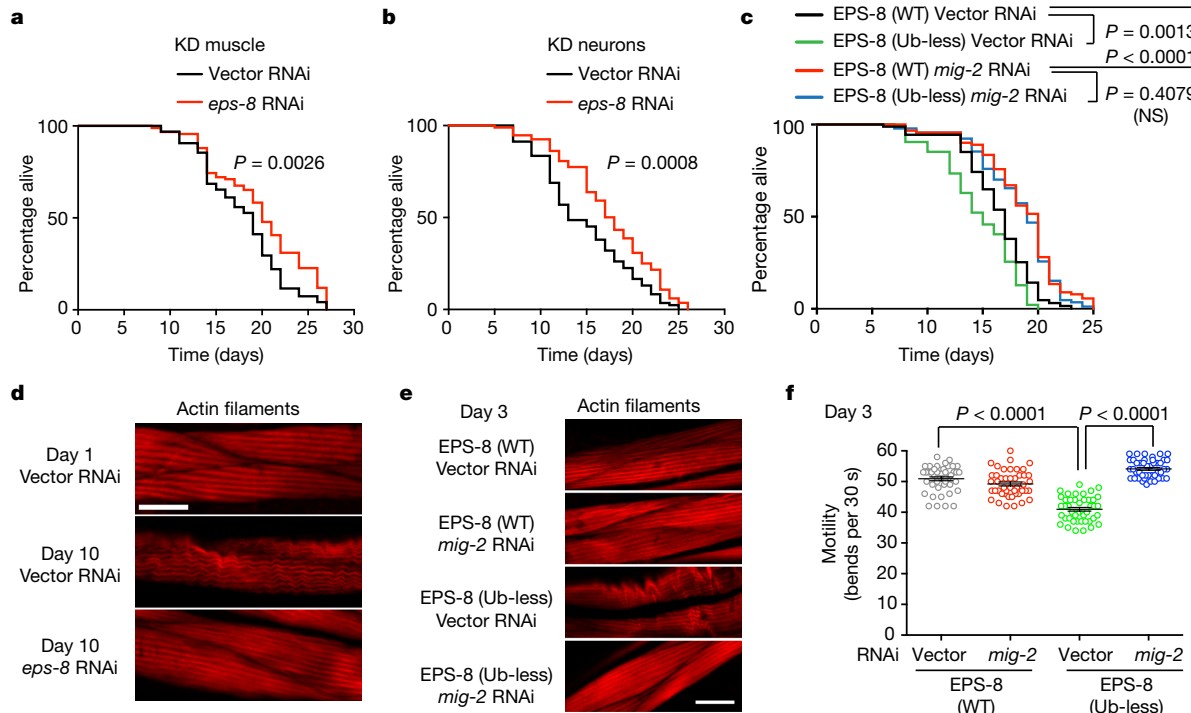

**Fig. 4 | Increased EPS-8 levels shorten lifespan through RAC hyperactivation. a, b**, Muscle-specific (**a**) and neuronal-specific (**b**) knockdown of *eps-8* after development extends lifespan. **c**, Knockdown of *mig-2* after development extends longevity and rescues the short lifespan induced by ubiquitin-less EPS-8. *P* values in **a**–**c** were determined by two-sided log-rank test; *n* = 96 worms per condition. Supplementary Table 11 contains replicate data of independent experiments. **d**, Staining of filamentous actin with phalloidin. *eps-8* RNAi prevents age-associated destabilization of actin filaments. Scale bar, 20 μm. Representative of two independent experiments. **e**, Knockdown of *mig-2* rescues the disruption of actin filaments in day-3 adult worms induced by ubiquitin-less EPS-8. Scale bar, 20 μm. Representative of two independent experiments. **f**, Thrashing movements per 30 s (*n* = 45 worms per condition from three independent experiments). Knockdown of *mig-2* suppresses motility deficits induced by ubiquitin-less EPS-8 in young adult worms. Data are mean ± s.e.m. *P* values in **f** determined by two-sided *t*-test. In all experiments, RNAi was initiated at day 1 of adulthood.

intestinal alterations trigger bacterial colonization[29], we asked whether IFB-2 influences this deleterious process. Notably, knockdown of *ifb-2* diminished bacterial invasion in the intestine of aged worms, whereas ubiquitin-less IFB-2 mutations exacerbated this phenotype (Fig. 3f, g, Extended Data Fig. 8d, e). Thus, increased levels of IFB-2 could underlie the collapse of other intermediate filaments during ageing, leading to loss of intestinal integrity.

## Increased EPS-8 levels hyperactivate RAC

Although the levels of EPS-8 increased in distinct tissues with age, it regulated organismal lifespan through its activity within muscle and neurons (Fig. 4a, b, Extended Data Fig. 9a, b). EPS-8 promotes the exchange of GDP for GTP on RAC protein, which then becomes active to regulate a wide range of pathways[30]. *C. elegans* expresses three *rac*-like genes (*rac-2*, *mig-2* and *ced-10*) required for development[31]. Similar to EPS-8, knockdown of either *rac-2* or *mig-2* in muscle and neurons after development extended lifespan (Fig. 4c, Extended Data Fig. 9c–f). Moreover, knockdown of *mig-2* prevented the short lifespan induced by the ubiquitin-less EPS-8 variant, which supports the idea that EPS-8 decreases longevity through RAC hyperactivation (Fig. 4c).

RAC induces phosphorylation and subsequent activation of the protein kinase JNK, which regulates transcription factors involved in cell survival and death signalling[30,32]. In correlation with upregulated EPS-8 levels, we observed increased phosphorylation of JNK during ageing (Extended Data Fig. 9g). By contrast, the knockdown of *eps-8* reduced JNK phosphorylation in aged worms (Extended Data Fig. 9h). The worm JNK homologue KGB-1 protects against stress during development, but its activity becomes detrimental with the onset of adulthood[33]. Knockdown of *kgb-1* during adulthood ameliorated the short-lived phenotype

of ubiquitin-less EPS-8 mutants (Extended Data Fig. 9i), which indicates that RAC signalling influences longevity through JNK activation.

RAC also promotes the polymerization and remodelling of the actin cytoskeleton[34]. With age, muscle cells exhibit unorganized actin filaments that impair organismal motility[35]. Notably, *eps-8* knockdown from adulthood prevented the age-associated destabilization of muscle actin networks and associated myosin filaments, ameliorating deficits in motility (Fig. 4d, Extended Data Fig. 10a, b). By contrast, knockdown of *eps-8* did not rescue age-associated changes in actin organization within intestinal and epidermal cells (Extended Data Fig. 10c, d). Filter trap experiments indicated that actin protein also aggregates during ageing, whereas knockdown of *eps-8* rescued this phenotype (Extended Data Fig. 10e). To determine in which tissues the increased EPS-8 levels trigger actin aggregation, we used tissue-specific RNAi. Knockdown of *eps-8* in the muscle or neurons reduced actin aggregation during ageing, whereas knockdown in other tissues did not prevent aggregation (Extended Data Fig. 10f–i).

Similar to *eps-8*, lowering the levels of hyperactivated RAC after development prevented the destabilization of muscle actin cytoskeleton during ageing (Extended Data Fig. 10j, k). Moreover, knockdown of *mig-2* rescued the accelerated disruption of actin filaments, aggregation of actin and motility deficits induced by ubiquitin-less EPS-8 (Fig. 4e, f, Extended Data Fig. 10l). Thus, hyperactivation of RAC by increased EPS-8 levels could result in excessive actin remodelling and polymerization, leading to altered actin networks with age.

## Discussion

Our study has demonstrated a global deubiquitination across tissues during ageing that impairs targeted proteasomal degradation

of lifespan regulators. Besides IFB-2 and EPS-8, we identified other dysregulated proteasome targets such as the cytokine receptor F54D1.6 and the solute carrier C46C2.2 that could have synergistic effects on ageing. Because ubiquitination also tags proteins for degradation through autophagy, our datasets have implications for understanding the link between autophagy and longevity[3]. Notably, we identified more than 1,000 ubiquitination changes during ageing that do not induce alterations in protein levels. These ubiquitination events could modulate protein activity and localization, and lead to longevity modifiers.

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

# Methods

## *C. elegans* strains and maintenance

*C. elegans* were grown and maintained at 20 °C on standard Nematode Growth Medium (NGM) seeded with *E. coli* (OP50)[36]. Wild-type (N2), DA1116 (*eat-2 (ad1116)II*), and RW1596 (*myo-3(st386)*V; *stEx30[myo-*3p::GFP::*myo-3* + *rol-6(su1006)*]) were provided by the *Caenorhabditis* Genetics Center (CGC) (University of Minnesota), that is supported by the NIH Office of Research Infrastructure Programs (P40 OD010440). CF1041 (*daf-2(e1370)III*) was provided by C. Kenyonu8). MT7929 (*unc-13(e51)I*) outcrossed four times to wild-type N2 was produced by T. Hoppe. AGD1657 (*unc-119(ed3)*III; uthSi13[*gly-19p*::LifeAct::mRuby::*unc-54* 3'UTR::*cb-unc-119*(+)]IV) and AGD1654 (*unc-119*(*ed3*) III; uthSi10[*col-19p*::LifeAct::mRuby::*unc-54* 3'UTR::*cb-unc-119*(+)] IV)[37] were a gift from A. Dillin. BJ49 (*kcIs6[ifb-2p::ifb-2a*::CFP]IV)[38], BJ186 (*kcIs30[ifo-1p*::*ifo-1*::YFP;*myo-3p*::mCherry::*unc-54*]III)[39], BJ324 (*kcEx78[ifc-1p*::*ifc-1*::eGFP; *unc-119(ed3)*+];*unc-119(ed3)III*), BJ316 (*ifc-2(kc16[ifc-2a/e*::YFP*])X*), and BJ312 (*kcIs40[ifp-1p*::*ifp-1*::eGFP*]IV*)[40] were a gift from R. E. Leube.

To generate the strains DVG197 (N2, *ocbEx162[sur-5p::ifb-2, myo-3p*::GFP]) and DVG198 (N2, *ocbEx163[sur-5p::ifb-2, myo-3p*::GFP]), a DNA plasmid mixture containing 70 ng μl$^{-1}$ of the plasmids *sur5-p::ifb-2* (pDV199) and 20 ng μl$^{-1}$ pPD93_97 (*myo3-p*::GFP) was injected into the gonads of adult N2 hermaphrodite worms by standard methods[41]. To construct the *C. elegans* plasmid (pDV199) for overexpression of *ifb-2*, pPD95.77 from the Fire Lab kit was digested with SphI and XmaI to insert 3.6-kb of the *sur-5* promoter. The resultant vector was then digested with KpnI and EcoRI to excise GFP and insert a multi-cloning site. *F10C1.7 (ifb-2)* was PCR-amplified from cDNA to include 5' NheI and 3' NotI restriction sites and then cloned into the aforementioned vector. The construct was sequence verified. The corresponding control DVG9 strain (N2, *ocbEx9[myo3p*::GFP*]*) was generated by microinjecting N2 worms with 20 ng μl$^{-1}$ pPD93_97 (ref. [42]).

For tissue-specific RNAi assays, we used either *rde-1* or *sid-1* mutant worms in which wild-type *rde-1* or *sid-1* expression has been rescued using tissue-specific promoters, respectively. Tissue-specific knockdown was validated in the original publications. In the VP303 strain (*rde-1(ne219)*V; *kbIs7[nhx-2p::rde-1* + *rol-6(su1006)]*), RNAi treatment is only effective in the intestine[43]. For muscle-specific knockdown, we used the WM118 strain (*rde-1(ne300)*V; *neIs9[myo-3p*::HA::RDE-1 + *rol-6(su1006)]*)[44]. For neuronal-specific RNAi, we used the TU3401 strain (*sid-1(pk3321)*V; *uIs69*[pCFJ90(*myo-2p*::mCherry) + *unc-119p*::*sid-1*])[45]. For epidermal-specific RNAi experiments, we used the NR222 strain (*rde-1(ne219)*V; kzIs9 [(pKK1260) *lin-26p*::NLS::GFP + (pKK1253) *lin-26p*::*rde-1* + *rol-6(su1006)*]))[46]. All the tissue-specific RNAi strains were provided by the CGC.

Ubiquitin-less IFB-2 and EPS-8 mutant worms were generated at SunyBiotech (http://www.sunybiotech.com) by CRISPR–Cas9 methodology. The IFB-2(K255R/K341R) mutant strain (VDL07, *ifb-2(syb2876)*II)) was generated using the sgRNAs sg1-CCACCAGAGTTGATCTTGAGACA and sg2-CCTCCGATTTATGCGTGAAGACT. The strain expressing endogenous wild-type EPS-8 tagged with 3xHA (VDL05, *eps-8(syb2901)*IV)) was generated by CRISPR–Cas9 using sg1-CCTGTATTCACTATTAACCCAAT and sg2-CACTATTAACCCAATTCTCTAGG. The strain expressing endogenous EPS-8(K524R/K583R/K621R::3xHA) (VDL06, *eps-8(syb2901, syb3149)* IV) was generated using sg1-GGAGAGTCGTTTGAGGCATGAGG and sg2-AGGAGCTGACTGTTCACAAGGGG. Because *eps-8* generates several different isoforms, the numbering of ubiquitination sites corresponds to the leading EPS-8 isoform identified in our proteomics data (that is, transcript Y57G11C.24g.1, EPS-8 protein isoform g). The strain expressing endogenous IFB-2 tagged with GFP (VDL08, *ifb-2(syb3973)*II)) was also generated at SunyBiotech using sg1-CCAGACGACGGTCGCTTCTTCCC and sg2-CCGTTAAGAAATCTCCATCATCT. In all the aforementioned strains, the editing was confirmed by sequencing the *ifb-2* and *eps-8* genes in both directions.

## Lifespan assays

Young hermaphrodite adults were randomly picked from maintenance plates and transferred onto plates with OP50 *E. coli*. After 6 h of egg laying, adult worms were removed from the plates. Then, synchronized larvae were raised and fed OP50 *E. coli* at 20 °C until they developed into hermaphrodite adults. Once worms reached adulthood, they were randomly picked and transferred onto plates with HT115 *E. coli* carrying empty vector or RNAi clones for lifespan assays. For the non-integrated lines DVG9, DVG197 and DVG198, GFP-positive worms were selected for lifespan studies. All the lifespan assays of adult worms were conducted at 20 °C. In total, 96 worms were assessed per condition and scored every day or every other day[47]. Sample sizes were determined according to our previous studies and other publications[8,47]. Lifespan experiments were not blinded.

From the initial worm population, we censored the worms that were lost or burrowed into the medium as well as those that exhibited bagging or 'protruding vulva'. Supplementary Table 11 indicates the total number of uncensored worms and total number (uncensored + censored) of worms observed in each experiment. GraphPad PRISM 6 software was used to determine median lifespan and generate lifespan graphs. OASIS software (version 1) was used for statistical analysis to determine mean lifespan[48]. *P* values were calculated using the log-rank (Mantel–Cox) method. The *P* values refer to experimental and control worms in a single experiment. In the main and extended data figures, each graph shows a representative lifespan experiment. See Supplementary Table 11 for statistical analysis and replicate data.

## RNAi constructs

For RNAi experiments, hermaphrodite worms were fed *E. coli* (HT115) containing an empty control vector (L4440) or expressing double-stranded RNAi when they reached adulthood. *M01G12.9*, *C46C2.2, F54D1.6, ddi-1, usp-5, rac-2, csn-6, math-33, H34C03.2, usp-48, prp-8, cyk-3* and *F07A11.4* RNAi constructs were obtained from the Ahringer RNAi library. *ifb-2, eps-8, rpn-6.1, hsp-43, rpl-4, lec-1, mig-2, ced-10, kgb-1, cyld-1, usp-50, csn-5, otub-3* and *eif-3.F* RNAi constructs were obtained from the Vidal RNAi library. All RNAi constructs were sequence verified. See Supplementary Table 12 for further details about double-stranded RNAi constructs used for knockdown assays.

## Synchronization of large populations

To obtain large populations of synchronized hermaphrodite worms for proteomics, western blot, filter trap and quantitative PCR (qPCR), we used the bleaching technique[49]. In brief, young adult worms were collected by transferring random chunks of agar from maintenance plates and let them grow until there were sufficient young hermaphrodites for bleaching. Then, young worms were treated with alkaline hypochlorite solution (3 ml of bleach 4% (Fischer), 1.5 ml 5 N KOH, 5.5 ml dH$_2$O) for 4 min to destroy adult tissues and obtain eggs. After four washes with M9 buffer, eggs were maintained on M9 buffer without food overnight to allow hatching but prevent further development. Synchronized L1 larvae were randomly collected, raised and fed OP50 *E. coli* at 20 °C until late L4 larvae stage. Then, worms were transferred onto plates with OP50 *E. coli* (alternatively, HT115 *E. coli* for RNAi experiments) covered with 100 μg ml$^{-1}$ 5-fluoro-2'deoxyuridine (FUdR) to prevent the development of progeny. Every five days, adult worms were transferred onto fresh plates.

## Quantitative proteomics of the Ub-modified proteome and analysis

For label-free quantitative proteomics, wild-type, *daf-2(e1370)* and *eat-2(ad1116)* hermaphrodite worms were randomly collected with M9 buffer at days 1, 5, 10 and 15 of adulthood. For proteomics analysis of proteasome-less worms, we collected day 5 adult worms treated with either Vector RNAi or *rpn-6* RNAi. To remove bacteria, we washed the

worms five times with M9 buffer. Then, worms were resuspended in 10 M urea containing 50 mM triethylammonium bicarbonate (TEAB) and 25 mM *N*-ethylmaleimide. Protein was extracted using a Precellys 24 homogenizer (Bertin Technologies). Samples were centrifuged at 18,000*g* for 10 min at room temperature and supernatants were collected. Then, we determined protein concentrations with Pierce BCA protein assay (Thermo Scientific). For each sample, 22 mg of total protein were used as starting material and treated with 5 mM dithiothreitol (DTT) for 30 min at room temperature to reduce disulfide bonds. Carbamidomethylation was performed by incubation with 30 mM chloroacetamide for 30 min at room temperature. The urea concentration was diluted to 2 M with 50 mM ammonium bicarbonate and samples were digested with 220 µg trypsin (1:100 (w/w) enzyme:substrate ratio) for 3 h at room temperature. Then, we further added 220 µg trypsin and digested the samples overnight. After digestion, we added 0.5% formic acid and centrifuged 3,500*g* for 5 min the samples to remove precipitate. Peptides were desalted with 500-mg tC18 Sep-Pak cartridge (Waters). Then, 200 µg of peptides was separated and cleaned up using C18 Stage Tips (Thermo Fischer) for label-free proteomics of total protein levels. The rest of the material was used for enrichment of Ub-modified peptides. First, peptides were frozen at −80 °C for 3 h and then completely dried by vacuum centrifugation. Dried samples were dissolved in immunoaffinity purification solution (IAP). For enrichment of Ub-modified peptides, we used PTMScan Ubiquitin Remnant Motif (K-ε-GG) Antibody Bead Conjugate (Cell Signaling Technology). This antibody has specificity for diGly tag, which is the remnant of ubiquitin left on proteins after trypsin digestion. For each sample, 40 µl of antibody–bead conjugates were added and incubated for 6 h with gentle rotation at 4 °C. Then, beads were washed three times with ice-cold PBS and ice-cold water. Ub-modified peptides were eluted by incubating twice with 100 µl of 0.15% trifluoroacetic acid. Finally, peptides were desalted using C18 Stage Tips and analysed by label-free quantitative proteomics.

The liquid chromatography tandem mass spectrometry (LC−MS/MS) equipment consisted of an EASY nLC 1000 coupled to the quadrupole based QExactive Plus Orbitrap instrument (Thermo Scientific) via a nano-spray electroionization source. Peptides were separated on an in-house packed 50 cm column (2.7 µm C18 beads, Dr. Maisch) using a binary buffer system: (A) 0.1% formic acid and (B) 0.1% formic acid in 80% acetonitrile. The content of buffer B was raised from 5% to 30% within 65 min and followed by an increase to 50% within 10 min. Then, within 1 min, buffer B fraction was raised to 95% and then followed by washing and column equilibration for 15 min. Eluting peptides were ionized by an applied voltage of 2.2 kV. The capillary temperature was 275 °C and the S-lens RF level was set to 60. MS1 spectra were acquired using a resolution of 70,000 at 200 *m/z*, an automatic gain control target of $3 \times 10^6$ and a maximum injection time of 20 ms in a scan range of 300–1,750 Th. The ten most intense peaks were selected for isolation and fragmentation in the higher collisional dissociation cell using a normalized collision energy of 27 at an isolation window of 1.8 Th. Dynamic exclusion was enabled and set to 20 s. The MS/MS scan properties were: 17,500 resolution at 200 *m/z*, an automatic gain control target of $5 \times 10^5$ and a maximum injection time of 60 ms. For protein identification and LFQ in ubiquitin-proteomics experiments, we used the LFQ mode and MaxQuant (version 1.5.3.8) default settings[50]. For proteomics data sets of total protein levels in ageing and proteasome-less worms, we used Spectronaut 11 (Biognosys) with the BGS Factory Settings and MaxQuant (version 1.5.3.8) with default settings, respectively. MS2 spectra were searched against the *C. elegans* Uniprot database, including a list of common contaminants. FDRs on protein and peptide–spectrum match level were estimated by the target-decoy approach to 0.01% (protein FDR) and 0.01% (peptide–spectrum match FDR), respectively. The minimal peptide length was set to 7 amino acids and the match-between runs option was enabled. Carbamidomethylation (C) was considered as a fixed modification, whereas oxidation (M), acetylation (protein

N-term) and GlyGly (K) were included as variable modifications. All the downstream analyses of the resulting output were performed with R program (version 4.0.5) and Perseus (version 1.6.2.3). Protein groups flagged as reverse', 'potential contaminant' or 'only identified by site' were removed. LFQ values were $\log_2$-transformed and missing values were replaced using an imputation-based approach. Significant differences between the experimental groups were assessed by two-sided Student's *t*-test for samples. A permutation-based FDR approach was applied to correct for multiple testing. For worm tissue expression prediction, we used the database http://worm.princeton.edu[26]. Sample sizes for proteomics experiments were chosen based on previous studies on ubiquitin proteomics in mammalian cells[51], global proteomics changes during ageing in *C. elegans*[9,10] and our previous work on proteomics analysis of *C. elegans*[8]. Sample collection was not performed in a blinded manner. Once the samples were processed for proteomics, the mass spectrometry was performed by the CECAD Proteomics Facility in a blinded manner.

### Immunoprecipitation of Lys48 and Lys63-linked polyUb and proteomics analysis

Wild-type worms were randomly collected at day 5 of adulthood with M9 buffer and washed five times with M9 to remove bacteria. Then, worms were lysed in protein lysis buffer (50 mM Tris-HCl at pH 6.7, 150 mM NaCl, 1% NP40, 0.25% sodium deoxycholate, 1 mM EDTA, 1 mM NaF, 25 mM *N*-ethylmaleimide, and protease inhibitor cocktail (Roche)) using a Precellys 24 homogenizer. Samples were centrifuged at 18,000*g* for 10 min at 4 °C and supernatants were collected. We measured protein concentrations and used 2.5 mg of protein as starting material for each sample. The samples were incubated on ice for 3 h with anti-ubiquitin antibody, Lys48-specific, clone Apu2 (Merck, 05-1307, 1:50. RRID: AB_1587578) or anti-ubiquitin antibody, Lys63-specific, clone Apu3 (Merck, 05-1308, 1:50. RRID: AB_1587580). As a co-immunoprecipitation control, the same amount of protein was incubated with anti-Flag antibody (Sigma, F7425, 1:100; RRID: AB_439687) in parallel. Then, samples were incubated with 50 µl of µMACS MicroBeads (Miltenyi Biotec, 130-071-001) for 1 h on the overhead shaker at 4 °C. Subsequently, samples were loaded to pre-cleared µMACS column (Miltenyi Biotec, 130-042-701). Beads were washed three times with 50 mM Tris (pH 7.5) buffer containing 150 mM NaCl, 5% glycerol and 0.05% Triton and then washed five times with 50 mM Tris (pH 7.5) and 150 mM NaCl. After that, columns were subjected to in-column tryptic digestion containing 7.5 mM ammonium bicarbonate, 2 M urea, 1 mM DTT and 5 ng ml$^{-1}$ trypsin. Peptides were eluted using two times 50 µl of elution buffer 1 containing 2 M urea, 7.5 mM ambic and 15 mM chloroacetamide and incubated overnight at room temperature with mild shaking. Then, samples were stage-tipped the next day for label-free quantitative proteomics assay.

All samples were analysed on a Q-Exactive Plus (Thermo Scientific) mass spectrometer coupled to an EASY nLC 1200 UPLC (Thermo Scientific). Peptides were loaded with solvent A (0.1% formic acid in water) onto an in-house packed analytical column (50 cm × 75 µm I.D., filled with 2.7 µm Poroshell EC120 C18 (Agilent)). Peptides were chromatographically separated at a constant flow rate of 250 nl min$^{-1}$ using the 150-min method: 3–5% solvent B (0.1% formic acid in 80% acetonitrile) within 1 min, 5–30% solvent B (0.1% formic acid in 80% acetonitrile) within 65 min, 30–50% solvent B within 13 min and 50–95% solvent B within 1 min, followed by washing and column equilibration. The mass spectrometer was operated in data-dependent acquisition mode. The MS1 survey scan was acquired from 300 to 1,750 *m/z* at a resolution of 70,000. The top 10 most abundant peptides were subjected to higher collisional dissociation fragmentation at a normalized collision energy of 27% and the automatic gain control target was set to $5 \times 10^5$ charges. Product ions were detected in the Orbitrap at a resolution of 17,500. All mass spectrometric raw data were processed with MaxQuant (version 1.5.3.8) using default parameters as described above. LFQ was

performed using the LFQ mode and MaxQuant default settings. All downstream analyses were carried out on LFQ values with Perseus (version 1.6.2.3). Sample sizes for immunoprecipitation experiments followed by label-free quantitative proteomics were determined according to our previous publications[52,53]. Sample collection was not performed in a blinded manner.

Once the samples were processed for proteomics, the mass spectrometry was performed by the CECAD Proteomics Facility in a blinded manner.

### Cycloheximide and DUB inhibitor treatment
To assess the half-life of protein ubiquitin, we blocked ubiquitin synthesis by cycloheximide treatment[54]. Synchronized adult worms by bleaching technique were randomly transferred onto plates with OP50 bacteria covered with a final concentration of 54.5 µg ml$^{-1}$ cycloheximide (Sigma-Aldrich) during the indicated times in the corresponding figure. For DUB inhibitor experiments, synchronized adult worms were randomly transferred onto plates with OP50 bacteria covered containing a final concentration of 13.7 µg ml$^{-1}$ PR-619[16] (Merck) or vehicle control (DMSO) for 4 h and then collected for western blot analysis.

### RNA isolation and quantitative RT–PCR
Total RNA was isolated using RNAbee (Tel-Test) from approximately 2,000 adult hermaphrodite worms synchronized by bleaching technique at the ages indicated in the corresponding figures. Sample size was determined according to our previous publications[8,42]. cDNA was generated using a qScript Flex cDNA synthesis kit (Quantabio). SybrGreen real-time qPCR experiments were performed with a 1:20 dilution of cDNA using the CFC384 Real-Time System (Bio-Rad). Data were analysed with the comparative $2\Delta\Delta C_t$ method using the geometric mean of *cdc-42*, *pmp-3* and *Y45F10D.4* as housekeeping genes[55]. qPCR experiments were not blinded. See Supplementary Table 13 for details about the primers used for quantitative RT–PCR.

### Western blot
*C. elegans* were synchronized by bleaching technique and collected randomly at the indicated ages in the corresponding figures. Then, worms were lysed in protein lysis buffer (50 mM Hepes at pH 7.4, 150 mM NaCl, 1 mM EDTA, 1% Triton X-100, 2 mM sodium orthovanadate, 1 mM PMSF and protease inhibitor cocktail (Roche)) using a Precellys 24 homogenizer. Worm lysates were centrifuged at 8,000$g$ for 5 min at 4 °C and the supernatant was collected. Protein concentrations were determined with Pierce BCA protein assay (Thermo Scientific). Total protein (30 µg) was separated by SDS–PAGE, transferred to polyvinylidene difluoride membranes (Millipore) and subjected to immunoblotting. Western blot analysis was performed with anti-IFB-2 (Developmental Studies Hybridoma Bank, MH33, 1:1,000, RRID: AB_528311), anti-EPS8L2 (Abcam, ab85960, 1:1,000, RRID: AB_1924963), anti-GFP (AMSBIO, 210-PS-1GFP, 1:5,000, RRID: AB_10013682) and anti-α-tubulin (Sigma, T6199, 1:5,000, RRID: AB_477583). For analysis of JNK phosphorylation, we used the antibodies anti-JNK (Cell Signaling, 9252, 1:1,000, RRID: AB_2250373) and anti-phospho-JNK (Thr183/Tyr185) (Cell Signaling, 9251, 1:1,000; RRID: AB_331659) previously validated in *C. elegans*[56]. When indicated in the corresponding figure, we also performed analysis of total IFB-2 levels with antibody MH33 in whole *C. elegans* lysates without the centrifugation step.

For assessing ubiquitinated proteins by western blot, worms lysed in lysis buffer (50 mM Tris-HCl, pH 7.8, 150 mM NaCl, 1% Triton X100, 0.25% sodium deoxycholate, 1 mM EDTA, 25 mM *N*-ethylmaleimide, 2 mM sodium orthovanadate, 1 mM PMSF and protease inhibitor cocktail (Roche)) using Precellys 24 homogenizer. The cell debris was removed by centrifugation at 10,600$g$ for 10 min at 4 °C and the supernatant was collected. Protein concentrations were determined with standard BCA protein assay (Thermo Scientific). Then, 30 µg of total protein was separated by SDS–PAGE, transferred to nitrocellulose membranes (Millipore) and subjected to immunoblotting. Western blot analysis was performed with anti-ubiquitin (Sigma, 05-944, clone P4D1-A11, 1:1,000, RRID: AB_441944) and α-tubulin (Sigma, T6199, 1:5,000; RRID AB_477583). Sample size was determined according to our previous publications[8,53]. Western blot experiments were not blinded.

### Isolation of tissues for western blot analysis
Adult hermaphrodite worms synchronized by egg laying technique were collected and washed with M9 buffer. Then, worms were randomly picked and suspended into a droplet of M9 buffer on a glass slide. To cut off the head, two 27-gauge needles were moved in scissors motion. Decapitation was normally followed by the extrusion of at least one germline arm and the intestine. Heads, germlines and intestines were carefully separated using 27-gauge needles. A 20 µl pipette was used for collection of these tissues, which were then transferred to 1.5 ml Eppendorf tubes. Tissues were fast frozen and kept at −80 °C until a total of approximately 1,000 samples of each isolated tissue could be collected and combined to have enough material for western blot analysis. The collected tissues were then lysed in protein lysis buffer and protein concentrations were determined with Pierce BCA as detailed in the previous section for western blot analysis.

### Filter trap
Synchronized adult hermaphrodite worms by bleaching technique were randomly collected and washed with M9 buffer and worm pellets were frozen with liquid N2. Frozen worm pellets were thawed on ice and worm lysates were generated in non-denaturing protein lysis buffer (50 mM HEPES (pH 7.4), 150 mM NaCl, 1 mM EDTA, 1% Triton X-100) supplemented with 2 mM sodium orthovanadate, 1 mM PMSF and protease inhibitor cocktail (Roche)) using a Precellys 24 homogenizer. Worm lysates were centrifuged at 8,000$g$ for 5 min at 4 °C to remove debris and the supernatant was collected. Then, 100 µg of total protein was supplemented with 0.5% SDS and loaded onto a cellulose acetate membrane assembled in a slot blot apparatus (Bio-Rad). The membrane was then washed with 0.2% SDS and aggregates were assessed by immunoblotting using anti-IFB-2 (Developmental Studies Hybridoma Bank, MH33, 1:1,000, RRID: AB_528311), anti-GFP (AMSBIO, 210-PS-1GFP, 1:5,000, RRID: AB_10013682), and anti-β-actin (Abcam, ab8226, 1:5,000; RRID AB_306371). Sample size was determined according to our previous publications[42,53]. Filter trap experiments were not blinded.

### Motility assay
*C. elegans* were synchronized on *E. coli* (OP50) bacteria until L4 stage by egg laying technique and then randomly transferred to *E. coli* (HT115) bacteria containing either control empty vector or RNAi for the rest of the experiment. At the day of adulthood indicated in the corresponding figures, *C. elegans* were randomly picked and transferred to a drop of M9 buffer and after 30 s of adaptation the number of body bends was counted for 30 s. A body bend was defined as change in direction of the bend at the mid-body[53,57]. Sample size was determined according to our previous publications[42,53]. Motility assays were not blinded.

### Phalloidin staining
Adult hermaphrodite *C. elegans* were synchronized by egg laying technique and randomly collected at the ages indicated in the corresponding figures, washed with M9 buffer and fixed with ice cold 4% formaldehyde solution for 15 min. Worms were permeabilized with 2% Tween-20 in 1× PBS (pH 7.2) for 30 min at room temperature. Then, worms were treated with β-mercaptoethanol solution (120 mM Tris-HCl (pH 6.8), 5% β-mercaptoethanol, 1% Triton X-100) for 10 min. The worms were washed three times with 0.2% 1× PBS-Tween (pH 7.2) and three times with 1% BSA in 0.2% 1× PBS-Tween (pH 7.2). Worms were stained with rhodamine-phalloidin (Thermo Fischer, R415, 1:100) for 45 min. Three washes using 0.2% 1× PBS-Tween (pH 7.2) were followed before the cover slips were mounted on FluorSave Reagent (Merck, 345789).

The sample size for imaging experiments was determined according to previous publications[8,42]. Imaging experiments were not blinded.

## Bacterial colonization assay

Bacterial colonization or invasion experiments were performed as previously described[29]. All HT115 *E. coli* bacteria were cultured under carbenicillin selection. After overnight growth at 37 °C, bacterial cultures were induced with 1 mM IPTG for 4 h, collected and concentrated 10 times by centrifugation. Concentrated RNAi cultures were mixed at a ratio of 4:1 with concentrated HT115 bacteria expressing mCherry from the pDP151 plasmid. NGM agar plates were seeded with the bacterial mixed culture (4 parts HT115 RNAi, 1 part HT115 expressing mCherry) and allowed to dry for 24 h at room temperature. L4 larvae synchronized by egg laying technique were then randomly transferred onto these plates and analysed by fluorescence microscopy at days 1, 5 and 10 of adulthood. Before the analysis, worms were gently collected from the RNAi or mCherry agar plates, washed three times with M9 buffer, and placed onto OP50 *E. coli* plates to feed for 2 h to remove residual fluorescent bacteria from the intestine. Then, worms were immobilized using 0.1% azide in M9 buffer on 2% agarose pad. Images of worms were acquired with a Zeiss Axio Zoom.V16 fluorescence microscope. For quantification of fluorescence signal, worms were outlined and quantified using ImageJ software (version 1.5|s). Sample size for quantification of fluorescence reporters was determined according to our previous studies and other publications[8,58]. Bacterial colonization assays were not blinded.

## Quantification of hatched eggs and development into adults

Synchronized worms by egg laying technique were raised and fed OP50 *E. coli* at 20 °C until late L4 stage. Then, late L4 larvae were picked in a random manner and singly plated. After 24 h, the adult worm was removed and the number of eggs per plate was measured. The plate was kept for another 24 h, when the number of live progeny, visible as L1 larvae, was scored to assess the percentage of hatched eggs. L1 larvae were cultured for 50 h and the number of adult worms was scored in each plate. Sample size was determined according to previous work from our laboratory[8]. These experiments were not blinded.

## Reporting summary

Further information on research design is available in the Nature Research Reporting Summary linked to this paper.

## Data availability

There is no restriction on data availability. Readers can interact with the ubiquitin and global proteomics data using the following Shiny Web apps by downloading the datasets provided in the apps: https://vilchezlab.shinyapps.io/shiny-volcanoplot/ and https://vilchezlab.shinyapps.io/shiny-heatmap/. Proteomics data have been deposited in the ProteomeXchange Consortium via the PRIDE partner repository with the dataset identifiers PXD024338 (ubiquitin proteomics of ageing and long-lived worms), PXD025128 (global protein proteomics of ageing and long-lived worms), PXD024094 (ubiquitin proteomics upon *rpn-6* RNAi), PXD024095 (global protein proteomics upon *rpn-6* RNAi), PXD024093 (immunoprecipitation Lys48-linked polyUb) and PXD024045 (immunoprecipitation Lys63-linked polyUb). MS2 spectra in proteomics experiments were searched against the *C. elegans* Uniprot database (https://www.uniprot.org/proteomes/UP000001940). For worm tissue expression analysis, we used the database http://worm.princeton.edu. Source data are provided with this paper.

## Code availability

Custom code used in this article can be accessed at https://github.com/Vilchezlab/UbProteomics2021.

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

**Acknowledgements** The European Research Council (ERC Starting Grant-677427 StemProteostasis), the Deutsche Forschungsgemeinschaft (DFG) (VI742/4-1 and Germany's Excellence Strategy-CECAD, EXC 2030-390661388) and the Center for Molecular Medicine Cologne (C16) supported this research. We are grateful to the CECAD Proteomics Facility and C. Klein for their advice and contribution to proteomics experiments and analysis. We thank R. E. Leube for providing the *C. elegans* strains expressing intestinal intermediate filaments with fluorescent tags. We also thank P. M. Douglas for providing the HT115 bacteria driving mCherry expression. We thank R. Higuchi-Sanabria for comments on the manuscript.

**Author contributions** S.K. and D.V. performed most of the experiments, data analysis and interpretation. R.L. generated the *ifb-2*(OE) strains and helped with other experiments. H.J.L. performed experiments of isolated tissues. P.W. generated the Shiny Web apps and contributed to analysis of proteomics data. M.K. provided critical advice on the project. The manuscript was written by D.V. All authors discussed the results and commented on the manuscript.

**Funding** Open access funding provided by Universität zu Köln.

**Competing interests** The authors declare no competing interests.

**Additional information**
**Correspondence and requests for materials** should be addressed to D.V.

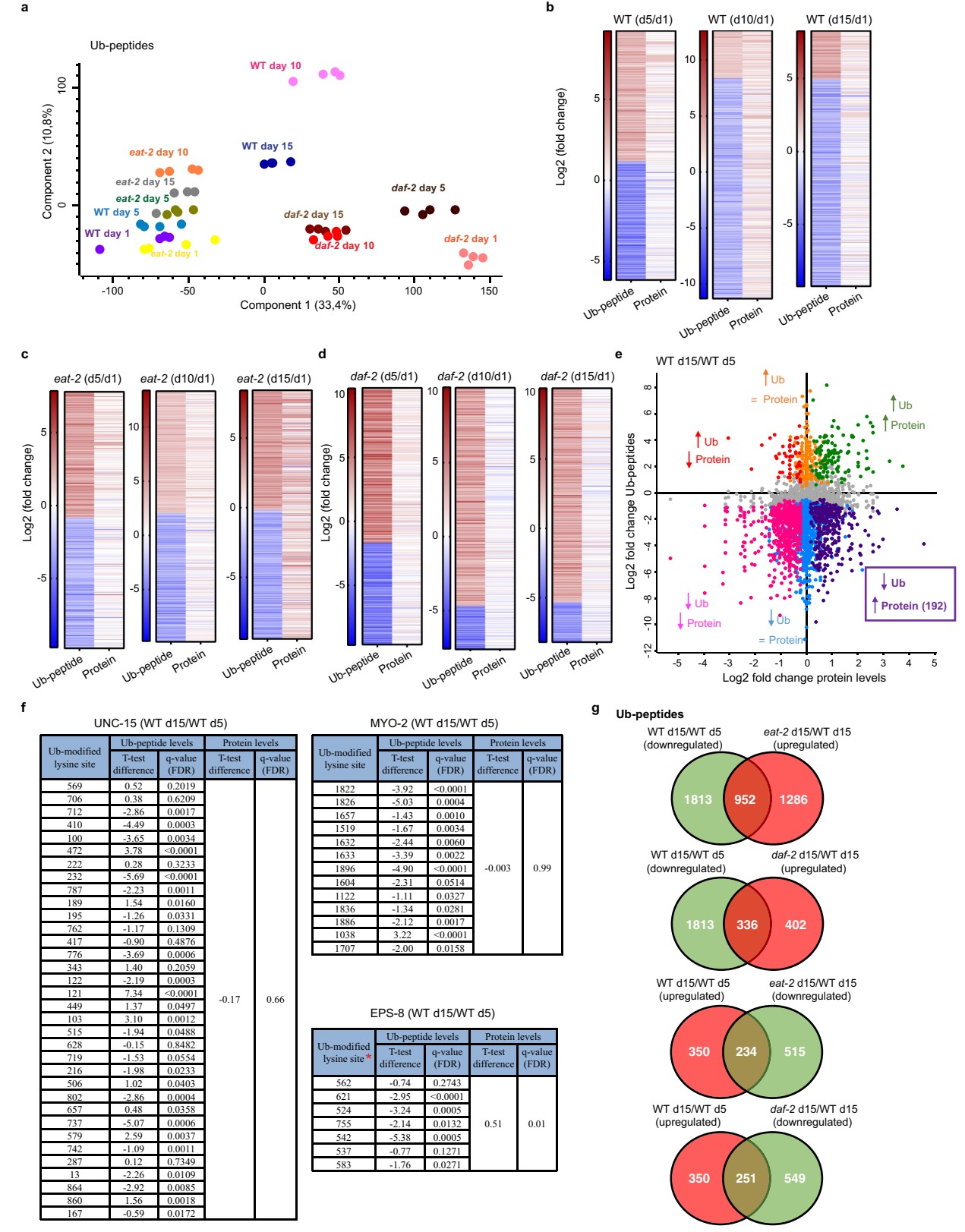

**Extended Data Fig. 1** | See next page for caption.

**Extended Data Fig. 1 | Differences in the levels of Ub-peptides often do not directly correlate with a change in the total amounts of the protein.**
**a**, Principal component analysis indicates high reproducibility in proteomics of Ub-peptides among the four biological replicates for each experimental condition (wild type, *eat-2(ad1116)*, *daf-2(e1370)* at day 1, 5, 10 and 15 of adulthood). **b**–**d**, Heat maps representing $\log_2$-transformed fold changes of all the differentially abundant Ub-peptides and their corresponding total protein levels at the indicated day and genetic background ($n = 4$, two-sided $t$-test, FDR <0.05 was considered significant). In each heat map, only significantly changed Ub-peptides are shown. Supplementary Table 3 contains complete list of all identified Ub-peptides and levels of the respective protein. **e**, Integrated proteomics analysis of $\log_2$-transformed fold changes in the levels of Ub-peptides and corresponding proteins comparing day-15 and day-5 wild-type worms ($n = 4$, two-sided $t$-test, FDR < 0.05). **f**, The total protein levels of myosin MYO-2 and paramyosin UNC-15 remain similar during ageing, but they contain multiple downregulated and upregulated Ub-sites. EPS-8 protein becomes more abundant with age, and most of its Ub-sites are significantly downregulated ($n = 4$, two-sided $t$-test, FDR < 0.05 was considered significant). Because EPS-8 has many different isoforms, we show the ubiquitination sites numbered according to the leading isoform identified in our proteomics data (EPS-8 protein isoform g). **g**, Among the 1,813 downregulated ubiquitin modifications in aged wild-type worms (WT d15/WT d5), age-matched *eat-2* (*eat-2* d15/WT d15) and *daf-2* mutants (*daf-2* d15/WT d15) exhibited increased ubiquitination for 952 and 336 peptides, respectively. Among the 350 upregulated ubiquitin modifications in aged wild-type worms, age-matched *eat-2* and *daf-2* worms exhibited decreased ubiquitination for 234 and 251 peptides, respectively.

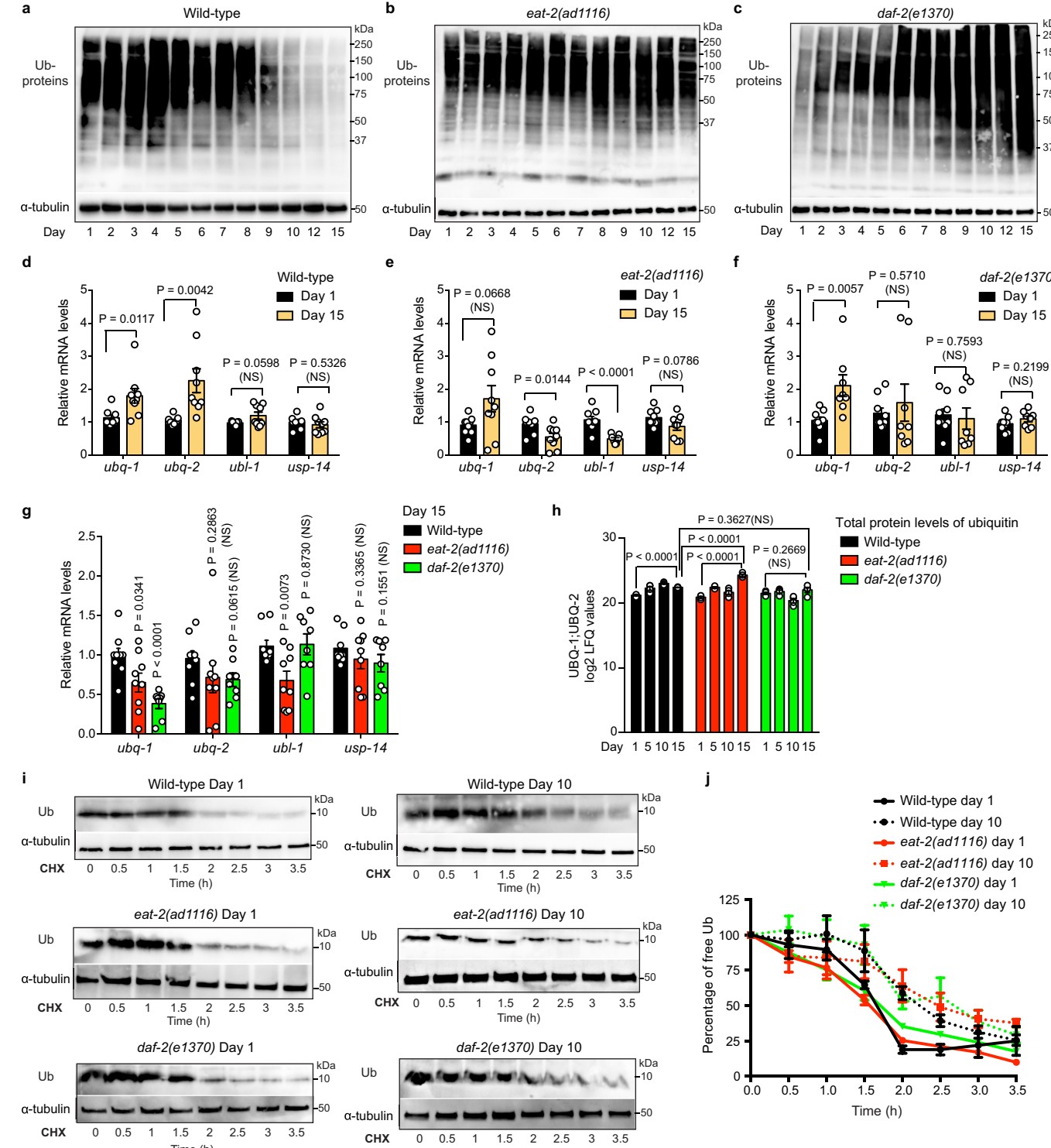

**Extended Data Fig. 2 |** See next page for caption.

**Extended Data Fig. 2 | Loss of ubiquitination in aged wild-type worms is not associated with lower expression or half-life of protein ubiquitin itself.**
**a**, Immunoblot of Ub-proteins in wild-type worms at the indicated days of adulthood. The images are representative of four independent experiments.
**b**, Immunoblot of Ub-proteins in *eat-2(ad1116)* mutant worms. Representative of four independent experiments. **c**, Immunoblot of Ub-proteins in *daf-2(e1370)* mutant worms. Representative of three independent experiments. **d**, qPCR analysis of ubiquitin (*ubq-1*), ubiquitin-ribosomal fusion (*ubq-2*, *ubl-1*) and ubiquitin-stress response (*usp-14*) genes comparing day-1 and day-15 wild-type adult worms. Graph represents the relative expression to day 1 adult wild-type worms (mean ± s.e.m., *n* = 9). *P* values: *ubq-1* (*P* = 0.0117), *ubq-2* (*P* = 0.0042), *ubl-1* (*P* = 0.0598), *usp-14* (*P* = 0.5326). **e**, qPCR analysis comparing day-1 and day-15 *eat-2* mutant worms. Graph represents the relative expression to day 1 adult *eat-2* mutant worms (mean ± s.e.m., *n* = 9). *P* values: *ubq-1* (*P* = 0.0668), *ubq-2* (*P* = 0.0144), *ubl-1* (*P* < 0.0001), *usp-14* (*P* = 0.0786). **f**, qPCR analysis comparing day 1 and day 15 *daf-2* mutant worms. Graph represents the relative expression to day 1 adult *daf-2* mutant worms (mean ± s.e.m., day 1 (*n* = 9); day 15 (*n* = 8)). *P* values: *ubq-1* (*P* = 0.0057), *ubq-2* (*P* = 0.5710), *ubl-1* (*P* = 0.7593), *usp-14* (*P* = 0.2199). **g**, qPCR analysis comparing day 15 wild-type worms with age-matched long-lived mutant worms. Graph represents the relative expression to wild-type worms (mean ± s.e.m., wild-type (*n* = 9); *eat-2* (*n* = 9); *daf-2* (*n* = 8)). Wild-type worms express higher levels of *ubq-1* gene at day 15 of adulthood (wild-type versus *eat-2*: *P* = 0.0341; wild-type versus *daf-2*: *P* < 0.0001). **h**, The log$_2$-transformed label-free quantification (LFQ) values of protein ubiquitin (UBQ-1; UBQ-2) from global proteomics analysis of wild-type, *eat-2(ad1116)* and *daf-2(e1370)* worms at the indicated ages (*n* = 4, mean ± s.e.m.). The total amounts of protein ubiquitin itself slightly increase in wild-type worm with age (wild-type d1 versus wild-type d15: *P* < 0.0001). Old wild-type and aged-matched *daf-2* mutant worms have similar levels of total protein ubiquitin (wild-type d15 versus *daf-2* d15: *P* = 0.2669). Label-free proteomics of total protein levels (Supplementary Table 2) was performed in input samples separated from the same lysates used for analysis of the Ub-modified proteome before the enrichment with an anti-diGly antibody. In **d**–**h**, *P* values were determined by two-sided *t*-test. **i**, Western blot of free ubiquitin (Ub) in worms treated with 54.5 μg ml$^{-1}$ cycloheximide (CHX) to block ubiquitin synthesis. Worms were lysed at the indicated time after CHX treatment. The images are representative of three independent experiments. **j**, Quantification of the western blots presented in the previous figure. Graph represents the percentage of free Ub levels relative to time point 0 h of CHX treatment (mean ± s.e.m., *n* = 3 independent experiments). For gel source data, see Supplementary Fig. 1.

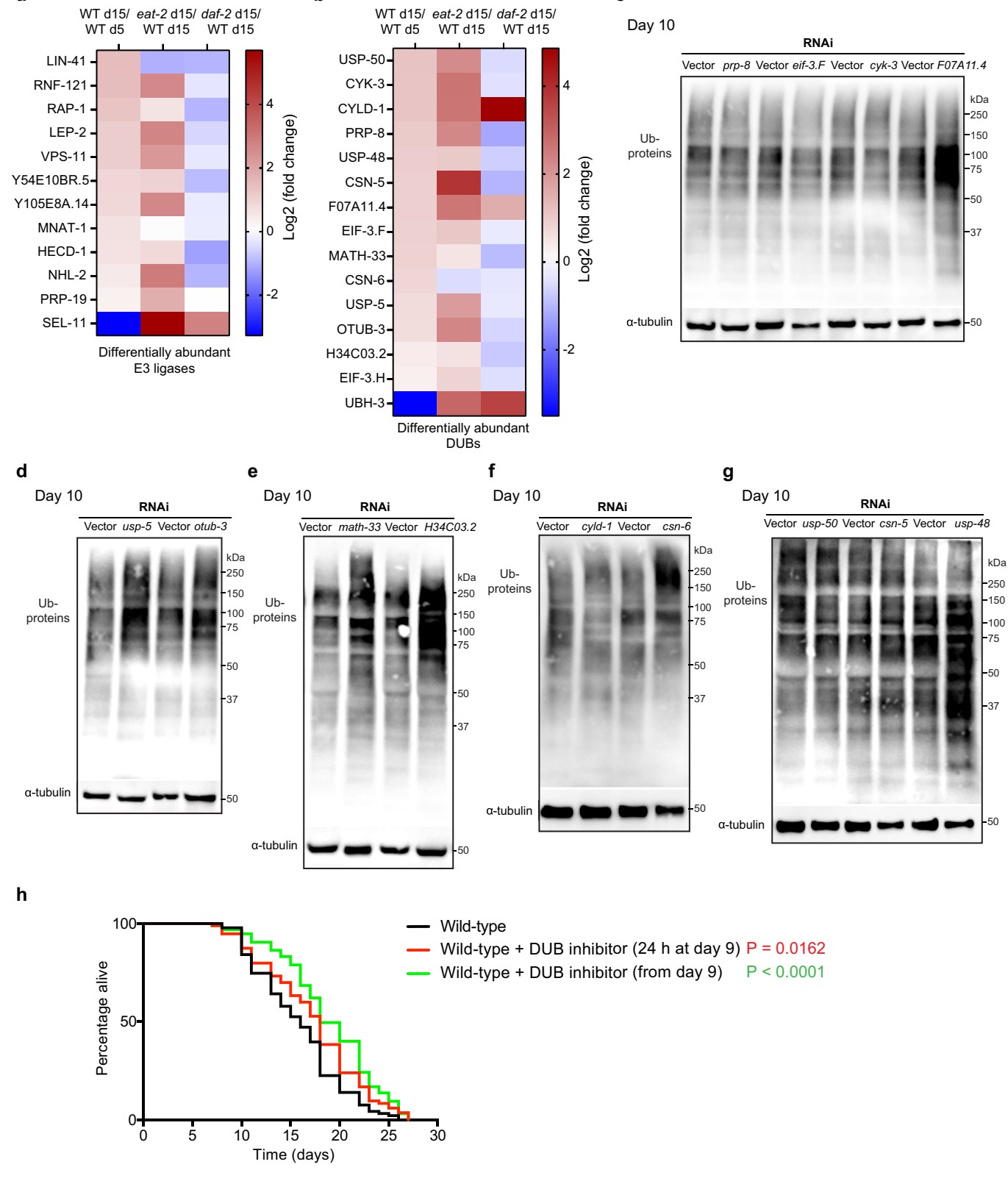

**Extended Data Fig. 3** | See next page for caption.

**Extended Data Fig. 3 | Single knockdown of specific age-dysregulated DUBs ameliorates the global decline in Ub-protein levels of aged wild-type worms. a**, Heat map representing all the differentially abundant E3 ubiquitin ligases in old (day 15) wild-type worms compared with young (day 5) wild-type worms ($n = 4$, two-sided $t$-test, FDR < 0.05 was considered significant). The levels of differentially abundant E3s in old wild-type worms were also compared with age-matched long-lived *eat-2* and *daf-2* mutants. **b**, Heat map representing all the differentially abundant DUBs in old wild-type worms compared with young wild-type worms ($n = 4$, two-sided $t$-test, FDR < 0.05 was considered significant). The levels of differentially abundant DUBs in old wild-type worms were also compared with age-matched long-lived *eat-2* and *daf-2* mutants. **c**, Immunoblot of Ub-proteins and α-tubulin in day 10 wild-type worms upon knockdown of *prp-8*, *eif-3.F*, *cyk-3* and *F07A11.4* after development. The images are representative of three independent experiments. **d**, Immunoblot of Ub-proteins in day-10 wild-type worms upon knockdown of *usp-5* and *otub-3* after development. Representative of three independent experiments. **e**, Immunoblot of Ub-proteins in day-10 wild-type worms upon knockdown of *math-33* and *H34C03.2* after development. Representative of four independent experiments. **f**, Immunoblot of Ub-proteins in day-10 wild-type worms upon knockdown of *cyld-1* and *csn-6* after development. Representative of four independent experiments. **g**, Immunoblot of Ub-proteins in day-10 wild-type worms upon knockdown of *usp-50*, *csn-5* and *usp-48* after development. Representative of four independent experiments. In **c**–**g**, RNAi was initiated at day 1 of adulthood. **h**, Wild-type worms treated with 13.7 μg ml$^{-1}$ of the broad-spectrum DUB inhibitor PR-619 at day 9 of adulthood for 24 h ($P = 0.0162$) or from day 9 of adulthood until the end of the experiment ($P < 0.0001$) live longer compared with untreated wild-type worms. $P$ values: two-sided log-rank test, $n = 96$ worms per condition. Supplementary Table 11 contains statistics and replicate data of independent lifespan experiments. For gel source data, see Supplementary Fig. 1.

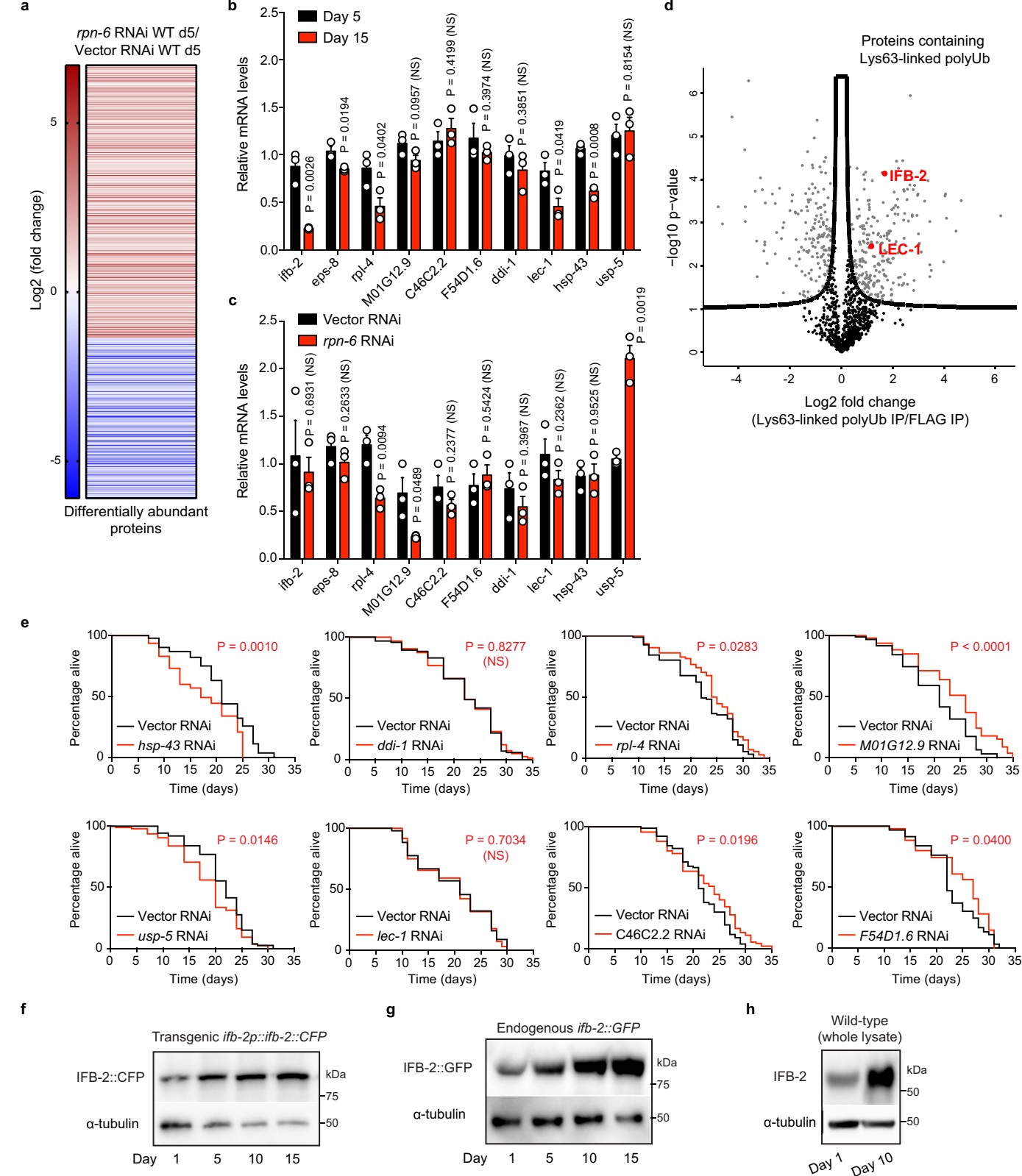

**Extended Data Fig. 4** | See next page for caption.

**Extended Data Fig. 4 | Ageing does not increase the mRNA levels of age-dysregulated proteasome targets. a**, Heat map representing $\log_2$-transformed fold changes of all the differentially abundant proteins comparing *rpn-6* RNAi-treated wild-type worms with Vector RNAi-treated wild-type worms at day 5 of adulthood ($n = 3$, two-sided *t*-test, FDR < 0.05 was considered significant). Loss of proteasome activity results in widespread changes in the individual protein levels of young adults, including 297 downregulated proteins and 509 upregulated proteins. See Supplementary Table 8 for complete list of all the detected proteins. **b**, The mRNA levels of age-dysregulated proteasome targets do not increase with age. In fact, *ifb-2* ($P = 0.0026$) and *eps-8* ($P = 0.0194$) mRNA levels decrease in aged worms. Graph represents the relative expression to young (day 5) adult worms (mean ± s.e.m., $n = 3$). **c**, Loss of *rpn-6* does not upregulate the mRNA levels of age-dysregulated proteasome targets in young worms (day 5), with the exception of the deubiquitinase *usp-5* ($P = 0.0019$) that could indicate a compensatory mechanism to ameliorate ubiquitin deficits triggered by dysregulated proteasome activity. Graph represents the relative expression to Vector RNAi control (mean ± s.e.m., $n = 3$). RNAi treatment was initiated at day 1 of adulthood. In **b**, **c**, *P* values were determined by two-sided *t*-test. **d**, Volcano plot of proteins containing Lys63-linked polyUb chains in wild-type worms at day 5 of adulthood ($n = 3$, FDR < 0.05). $-\log_{10}(P$ value) of a two-sided *t*-test is plotted against the $\log_2$-transformed fold change of LFQ values from immunoprecipitation experiments using an antibody against Lys63-linked polyUb compared with anti-Flag antibody. Red dots indicate age-dysregulated proteasome targets. **e**, Knockdown of either *hsp-43* ($P = 0.0010$) or *usp-5* ($P = 0.0146$) during adulthood shortens lifespan, indicating that these genes are essential for adult viability. Knockdown of either *ddi-1* ($P = 0.8277$) or *lec-1* ($P = 0.7034$) does not significantly affect lifespan. In contrast, single knockdown of *rpl-4* ($P = 0.0283$), *M01G12.9* ($P < 0.0001$), *C46C2.2* ($P = 0.0196$) and *F54D1.6* ($P = 0.0400$) extends lifespan. In each lifespan experiment, RNAi was initiated at day 1 of adulthood. *P* values were determined by two-sided log-rank test; $n = 96$ worms per condition. Lifespan data are representative of two independent experiments. Supplementary Table 11 contains statistics and replicate data of independent lifespan experiments. **f**, Western blot analysis with an antibody against GFP of integrated transgenic worms expressing *ifb-2p::ifb-2a*::CFP. α-tubulin is the loading control. Representative of two independent experiments. **g**, Western blot with an antibody against GFP of worms expressing endogenous IFB-2 tagged with GFP. Representative of two independent experiments. **h**, Western blot analysis of total IFB-2 levels with an anti-IFB-2 antibody in whole wild-type *C. elegans* extracts without mild centrifugation to collect supernatant. Representative of three independent experiments. For gel source data, see Supplementary Fig. 1.

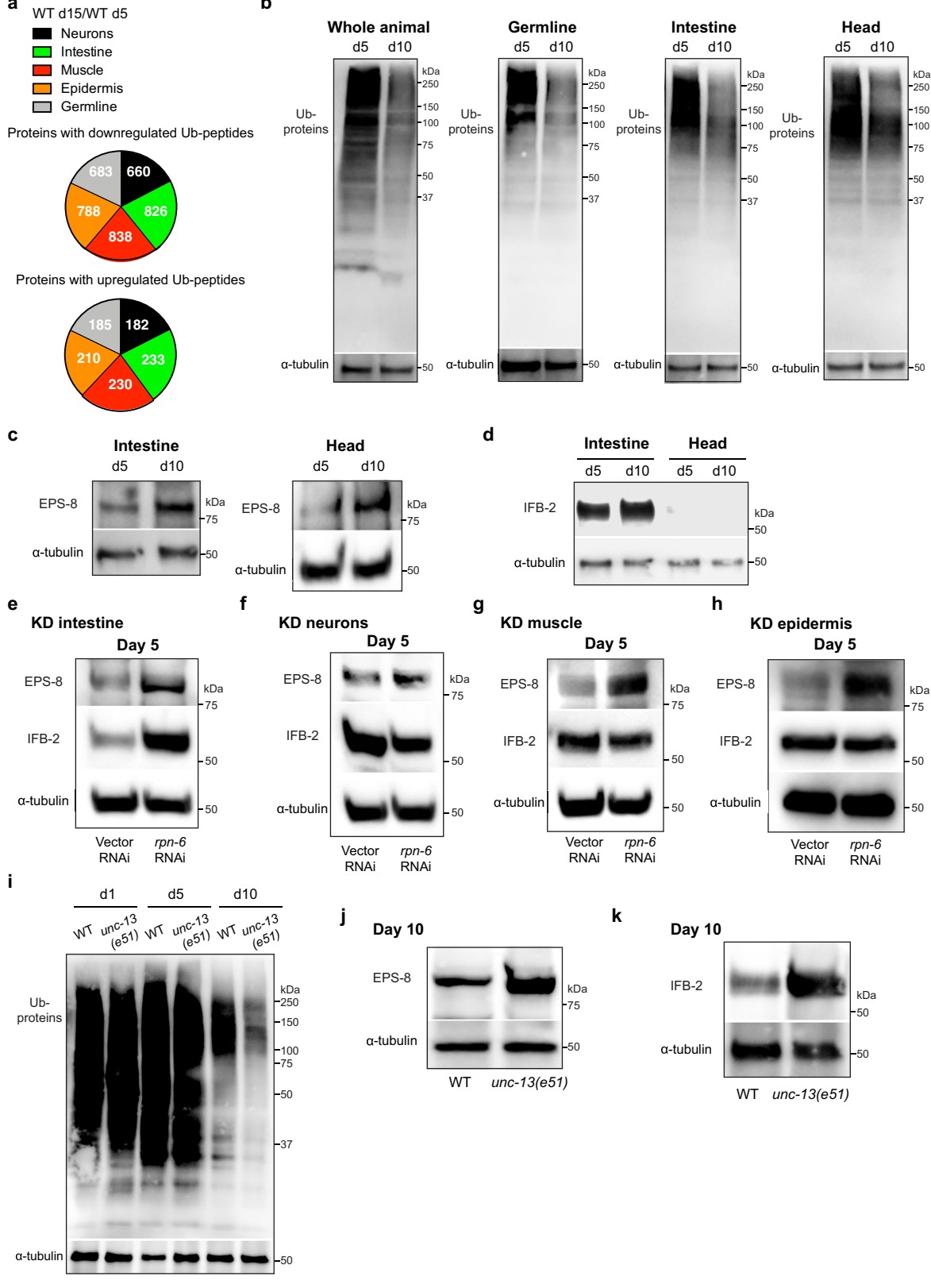

**Extended Data Fig. 5** | See next page for caption.

**Extended Data Fig. 5 | The age-associated decline in ubiquitination levels occurs across tissues. a**, Bioinformatics classification of proteins that exhibit differences in their Ub-peptides in aged wild-type worms (day 15) according to the tissues where these proteins are expressed. **b**, Immunoblot of Ub-proteins in lysates of whole worms or isolated germlines, intestines and heads from wild-type worms at the indicated days of adulthood. Representative of two independent experiments. **c**, Western blot analysis with an antibody against EPS-8 of isolated intestines and heads from wild-type worms at the indicated days. Representative of two independent experiments. **d**, Western blot analysis with an antibody against IFB-2 of isolated intestines and heads from wild-type worms at the indicated days. Representative of two independent experiments. **e**, Western blot with antibodies to EPS-8 and IFB-2 after intestinal-specific knockdown (KD) of *rpn-6*. RNAi rescued in the intestine of RNAi-deficient worms (*rde-1(ne219); nhx-2p::rde-1* strain). **f**, Western blot with antibodies to EPS-8 and IFB-2 after neuronal-specific KD of *rpn-6*. RNAi rescued in the neurons of RNAi-deficient worms (*sid-1(pk3321); unc-119p::sid-1* strain). **g**, Western blot with antibodies to EPS-8 and IFB-2 after muscle-specific KD of *rpn-6*. RNAi rescued in the muscle of RNAi-deficient worms (*rde-1(ne300); myo-3p::rde-1* strain). **h**, Western blot with antibodies to EPS-8 and IFB-2 after epidermal-specific KD of *rpn-6*. RNAi rescued in the epidermis of RNAi-deficient worms (*rde-1(ne219); lin-26p::rde-1* strain). In **e**–**h**, RNAi treatment was initiated at day 1 of adulthood. Worms were analysed at day 5 of adulthood. Representative of two independent experiments. **i**, Immunoblot of Ub-proteins in wild-type and *unc-13(e51)* mutant worms at the indicated days of adulthood. Representative of three independent experiments. **j**, Western blot with an antibody against EPS-8 in wild-type and *unc-13(e51)* mutant worms at day 10 of adulthood. Representative of three independent experiments. **k**, Western blot with an antibody against IFB-2 in wild-type and *unc-13(e51)* mutant worms at day 10 of adulthood. Representative of three independent experiments. For gel source data, see Supplementary Fig. 1.

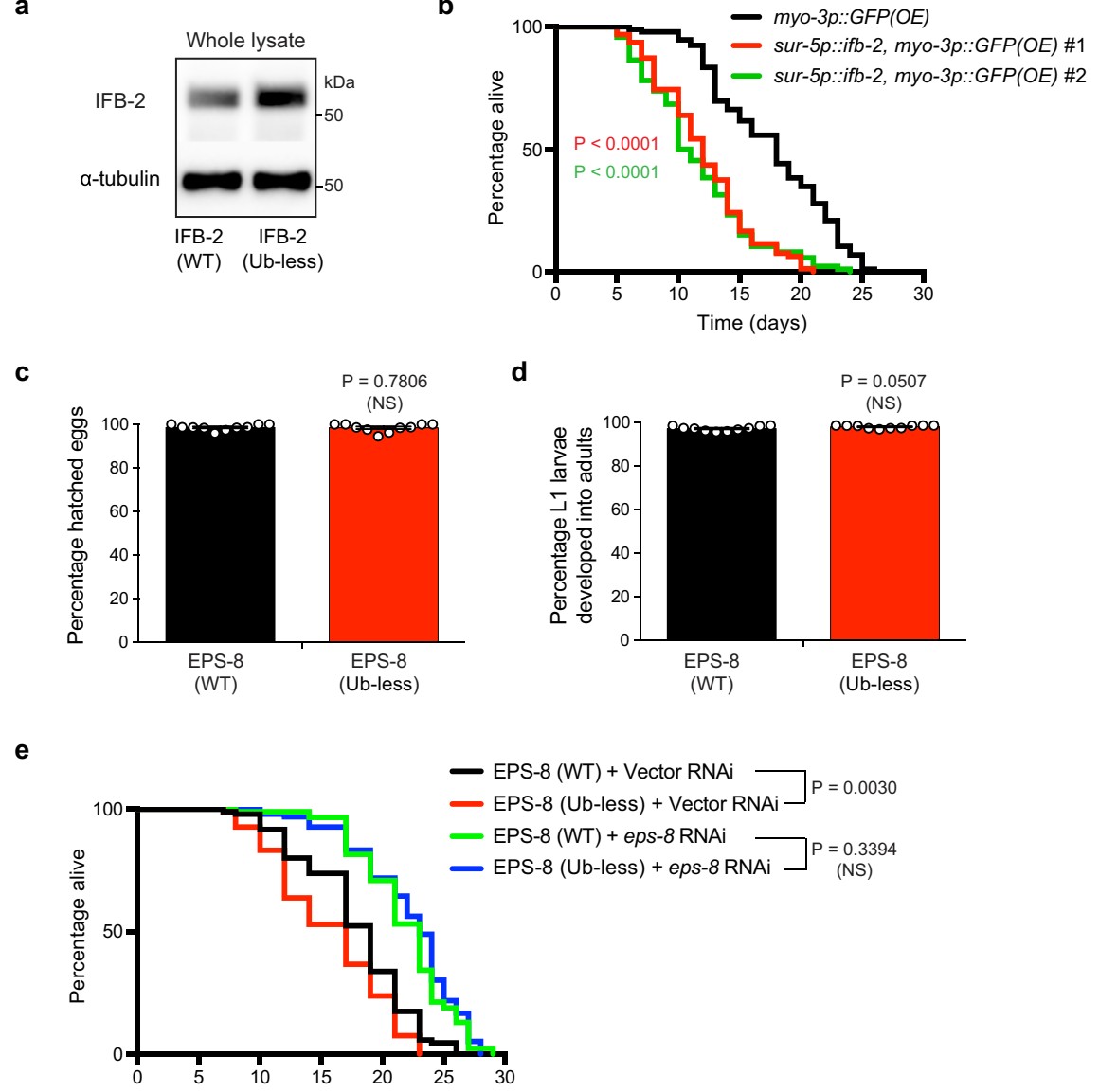

**Extended Data Fig. 6 | Increased levels of IFB-2 and EPS-8 shorten adult lifespan. a**, Western blot with an antibody against IFB-2 of whole lysates from wild-type worms and IFB-2 K255R/K341R (Ub-less) mutant worms at day 2 of adulthood. The images are representative of two independent experiments. **b**, Non-integrated transgenic DVG197 and DVG198 strains overexpressing *ifb-2* under *sur-5* promoter exhibit a short lifespan phenotype compared with the control strain (*P* < 0.0001). *P* values were determined by two-sided log-rank test; *n* = 96 worms per condition. Lifespan data are representative of two independent experiments. **c**, Percentage of hatched eggs (mean ± s.e.m., *n* = 10 worms scored per condition from 2 independent experiments). EPS-8(K524R/K583R/K621R) (Ub-less) mutants do not exhibit embryonic lethality (*P* = 0.7806). **d**, Percentage of L1 larvae that developed into adults

(mean ± s.e.m., *n* = 10 worms scored per condition from 2 independent experiments). Ubiquitin-less EPS-8 mutants do not exhibit developmental arrest (*P* = 0.0507). In **c**, **d**, *P* values were determined by two-sided *t*-test. **e**, Ubiquitin-less EPS-8 mutant worms live shorter than with worms that express wild-type EPS-8 (EPS-8 (WT) + Vector RNAi versus EPS-8 (Ub-less) + Vector RNAi, *P* = 0.0030). Knockdown of *eps-8* after development suppresses the deleterious effects on lifespan induced by Ub-less EPS-8 mutations (EPS-8 (WT) + *eps-8* RNAi versus EPS-8 (Ub-less) + *eps-8* RNAi, *P* = 0.3394). *P* values determined by two-sided log-rank test, *n* = 96 worms per condition. Supplementary Table 11 contains statistics and replicate data of independent lifespan experiments. For gel source data, see Supplementary Fig. 1.

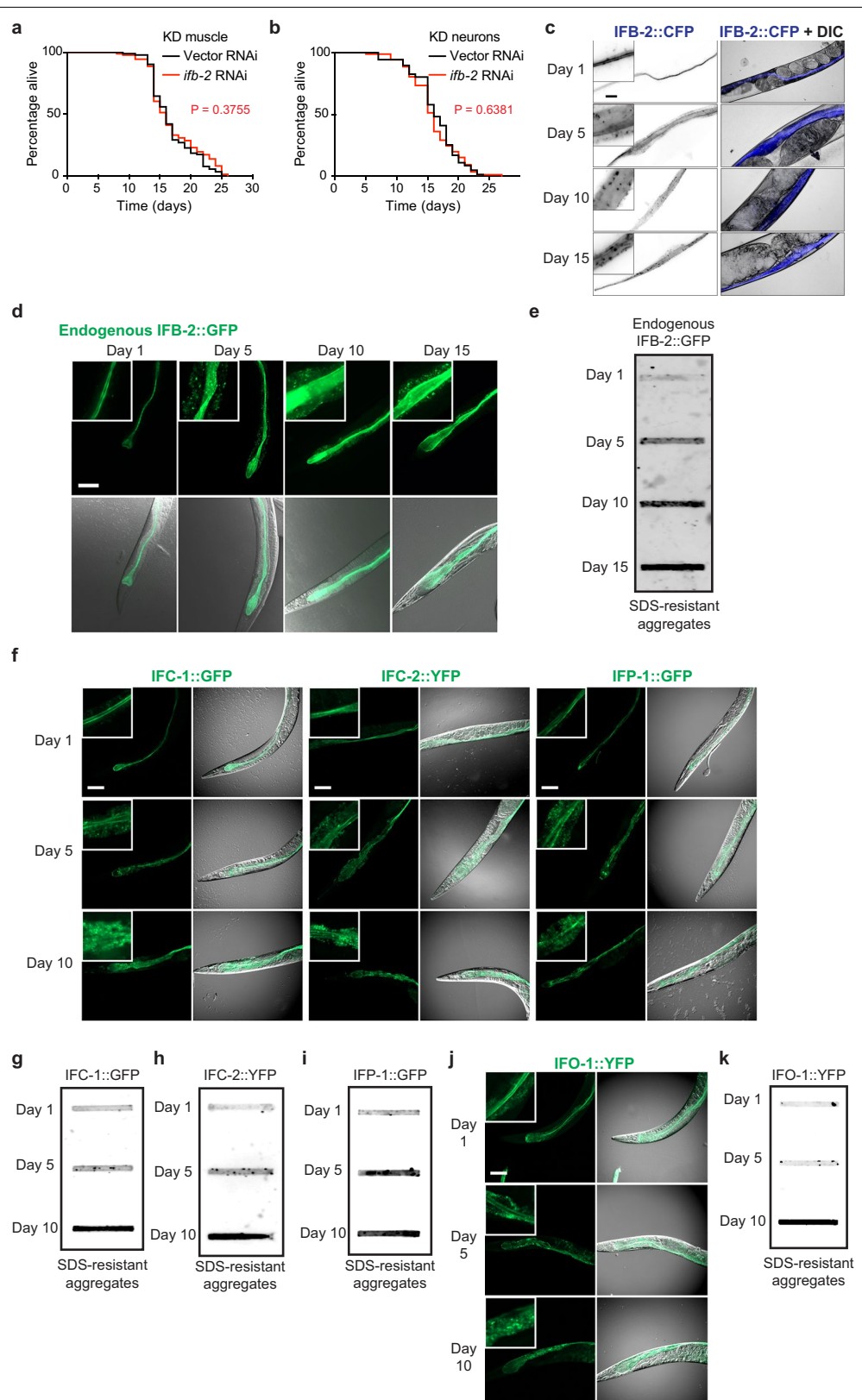

**Extended Data Fig. 7** | See next page for caption.

**Extended Data Fig. 7 | Ageing induces mislocalization and aggregation of IFB-2 and other intestinal intermediate filaments. a**, Muscle-specific knockdown of *ifb-2* does not affect lifespan (*P* = 0.3755). **b**, Neuronal-specific knockdown of *ifb-2* does not affect lifespan (*P* = 0.6381). In each lifespan experiment, RNAi was initiated during adulthood, *P* values determined by two-sided log-rank test, *n* = 96 worms per condition. Supplementary Table 11 contains statistics and replicate data of independent lifespan experiments. **c**, Representative images of transgenic worms expressing IFB-2::CFP under *ifb-2* promoter at different days of adulthood. On the left, IFB-2::CFP is presented in greyscale to show more visibly the IFB-2 aggregates. Merge images present IFB-2::CFP (blue) + DIC (differential interference contrast). Scale bar, 200 μm. Representative of three independent experiments. **d**, Images of worms expressing endogenous IFB-2 tagged with GFP at different days of adulthood. Scale bar, 100 μm. Representative of two independent experiments. **e**, Filter trap analysis with an antibody against GFP of worms expressing endogenous IFB-2 tagged with GFP. Representative of three independent experiments. **f**, Images of worms expressing endogenous IFC-2 tagged with YFP or the integrated transgenes *ifc-1p*::IFC-1::GFP and *ifp-1p*::IFP-1::GFP. Scale bar, 100 μm. Representative of two independent experiments. **g**, Filter trap analysis with an antibody against GFP of *C. elegans* expressing IFC-1::GFP under *ifc-1* promoter. Representative of two independent experiments. **h**, Filter trap with an antibody to GFP of *C. elegans* expressing endogenous intestinal intermediate filament IFC-2 tagged with YFP. Representative of three independent experiments. **i**, Filter trap with an antibody to GFP of *C. elegans* expressing IFP-1::GFP under *ifp-1* promoter. Representative of two independent experiments. **j**, Images of worms expressing the intestinal filament organizer IFO-1::YFP under *ifo-1* promoter. Scale bar, 100 μm. Representative of three independent experiments. **k**, Filter trap with an antibody to GFP of worms expressing IFO-1::YFP under *ifo-1* promoter. Representative of four independent experiments.

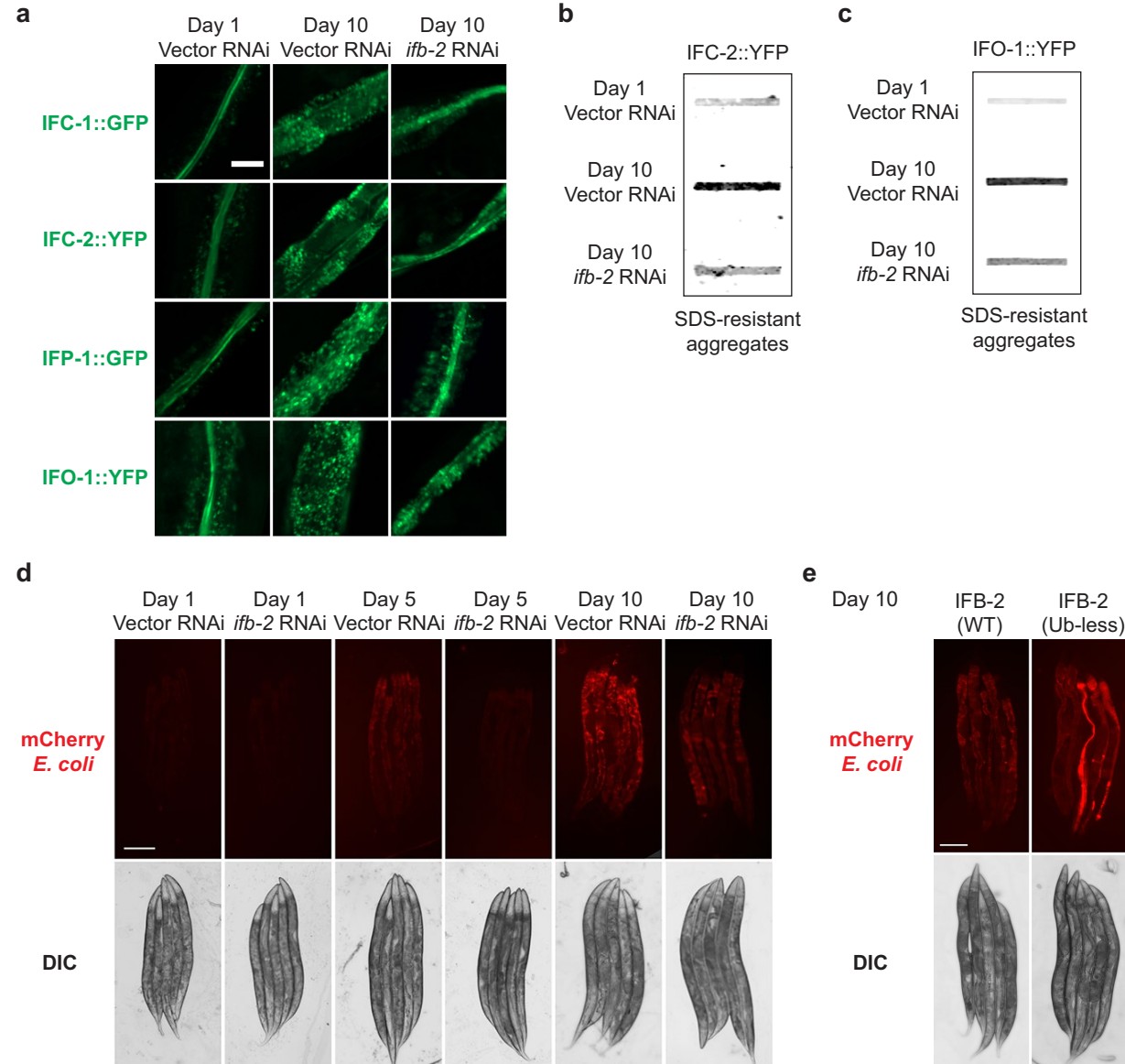

**Extended Data Fig. 8 | Downregulation of increased IFB-2 levels ameliorates loss of intestinal integrity and bacterial colonization. a**, *ifb-2* RNAi ameliorates age-related changes in the intracellular distribution and aggregation of the intestinal intermediate filaments IFC-1, IFC-2 and IFP-1 as well as the intestinal filament organizer IFO-1. We examined worms expressing endogenous IFC-2 tagged with YFP or the integrated transgenes *ifc-1p*::IFC-1::GFP, *ifp-1p*::IFP-1::GFP, and *ifo-1p*::IFO-1::GFP. *ifb-2* RNAi was initiated during adulthood. Scale bar, 25 μm. Representative of two independent experiments. **b**, Filter trap analysis with an antibody to GFP of *C. elegans* expressing endogenous intestinal intermediate filament IFC-2 tagged with YFP. *ifb-2* RNAi was initiated during adulthood. Representative of four independent experiments. **c**, Filter trap analysis with an antibody to GFP of worms expressing IFO-1::YFP under *ifo-1* promoter. *ifb-2* RNAi was initiated during adulthood. Representative of three independent experiments. **d**, Images of bacterial colonization in the intestine of wild-type *C. elegans* at different days of adulthood. RNAi was initiated during adulthood. Scale bar, 200 μm. Representative of three independent experiments. **e**, Images of bacterial colonization in the intestine of wild-type and IFB-2(K255R/K341R) (Ub-less) mutant worms at day 10 of adulthood. Scale bar, 200 μm. Representative of three independent experiments. Quantifications of bacterial colonization are shown in Fig. 3f, g.

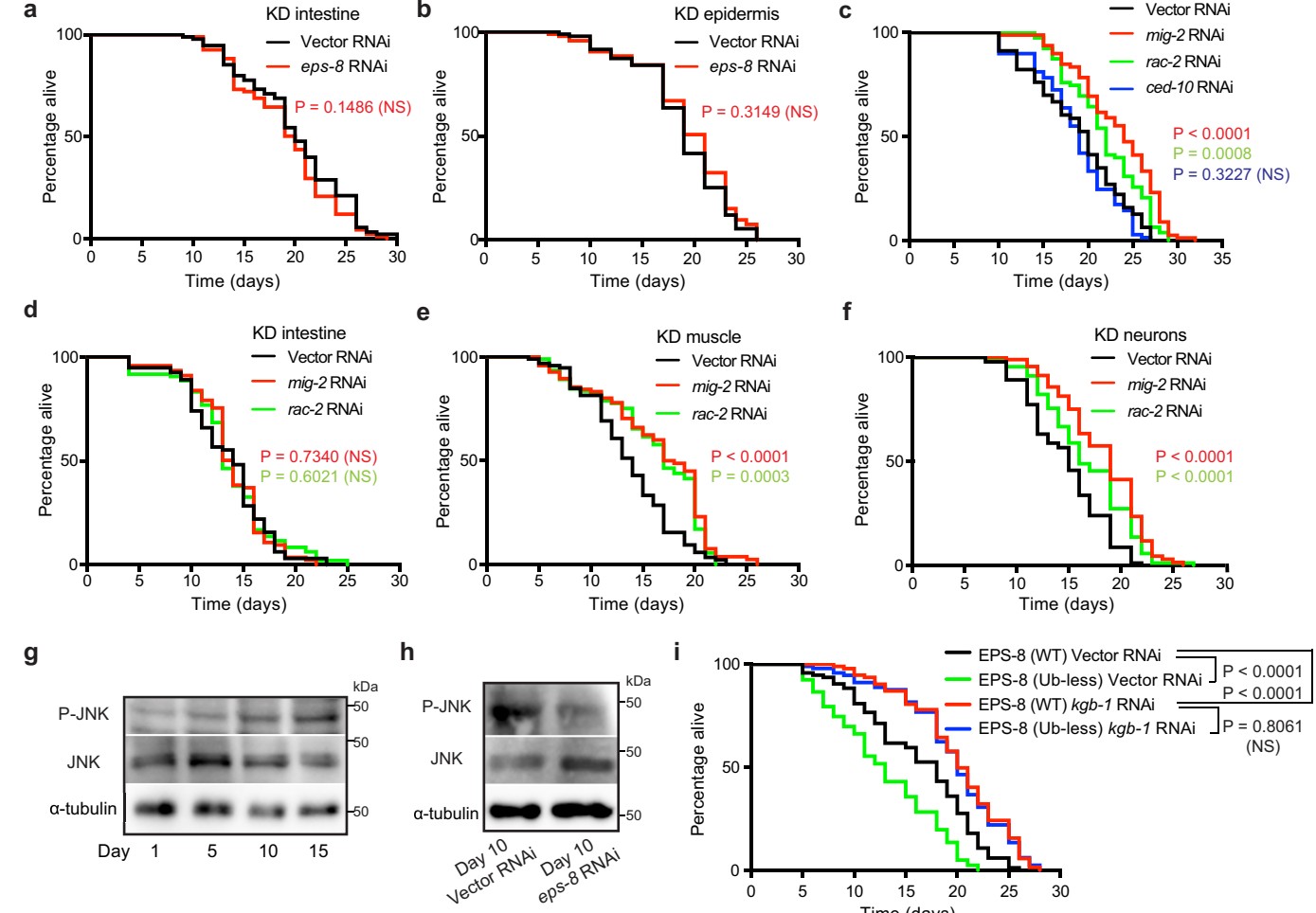

**Extended Data Fig. 9 | Age-related hyperactivation of RAC by increased EPS-8 levels increases JNK phosphorylation. a**, Intestinal-specific knockdown of *eps-8* after development does not affect lifespan (*P* = 0.1486). **b**, Epidermal-specific knockdown of *eps-8* during adulthood does not affect lifespan (*P* = 0.3149). **c**, Single knockdown of RAC orthologues *mig-2* (*P* < 0.0001) and *rac-2* (*P* = 0.0008) during adulthood extends lifespan in wild-type worms. Knockdown of RAC orthologue *ced-10* does not affect lifespan (*P* = 0.3227). **d**, Intestinal-specific knockdown of either *mig-2* (*P* = 0.7340) or *rac-2* (*P* = 0.6021) during adulthood does not affect lifespan. **e**, Muscle-specific knockdown of either *mig-2* (*P* < 0.0001) or *rac-2* (*P* = 0.0003) during adulthood extends lifespan. **f**, Neuronal-specific knockdown of either *mig-2* (*P* < 0.0001) or *rac-2* (*P* < 0.0001) during adulthood extends lifespan. **g**, Western blot analysis with antibodies to phosphorylated JNK (P-JNK), total

JNK and α-tubulin of wild-type worms at different days of adulthood. Representative of three independent experiments. **h**, Western blot analysis with antibodies to P-JNK, total JNK and α-tubulin of wild-type worms at day 10 of adulthood. *eps-8* RNAi was initiated during adulthood. Representative of two independent experiments. **i**, Knockdown of *kgb-1* after development extends longevity (EPS-8 (WT) Vector RNAi versus EPS-8 (WT) *kgb-1* RNAi, *P* < 0.0001) and rescues the short lifespan induced by ubiquitin-less EPS-8 mutant variant (EPS-8 (WT) Vector RNAi versus EPS-8 (Ub-less) Vector RNAi (*P* < 0.0001); EPS-8 (WT) *kgb-1* RNAi versus EPS-8 (Ub-less) *kgb-1* RNAi (*P* = 0.8061)). In each lifespan experiment, RNAi was initiated at day 1 of adulthood. *P* values were determined by two-sided log-rank test, *n* = 96 worms per condition. Supplementary Table 11 contains statistics and replicate data of independent lifespan experiments. For gel source data, see Supplementary Fig. 1.

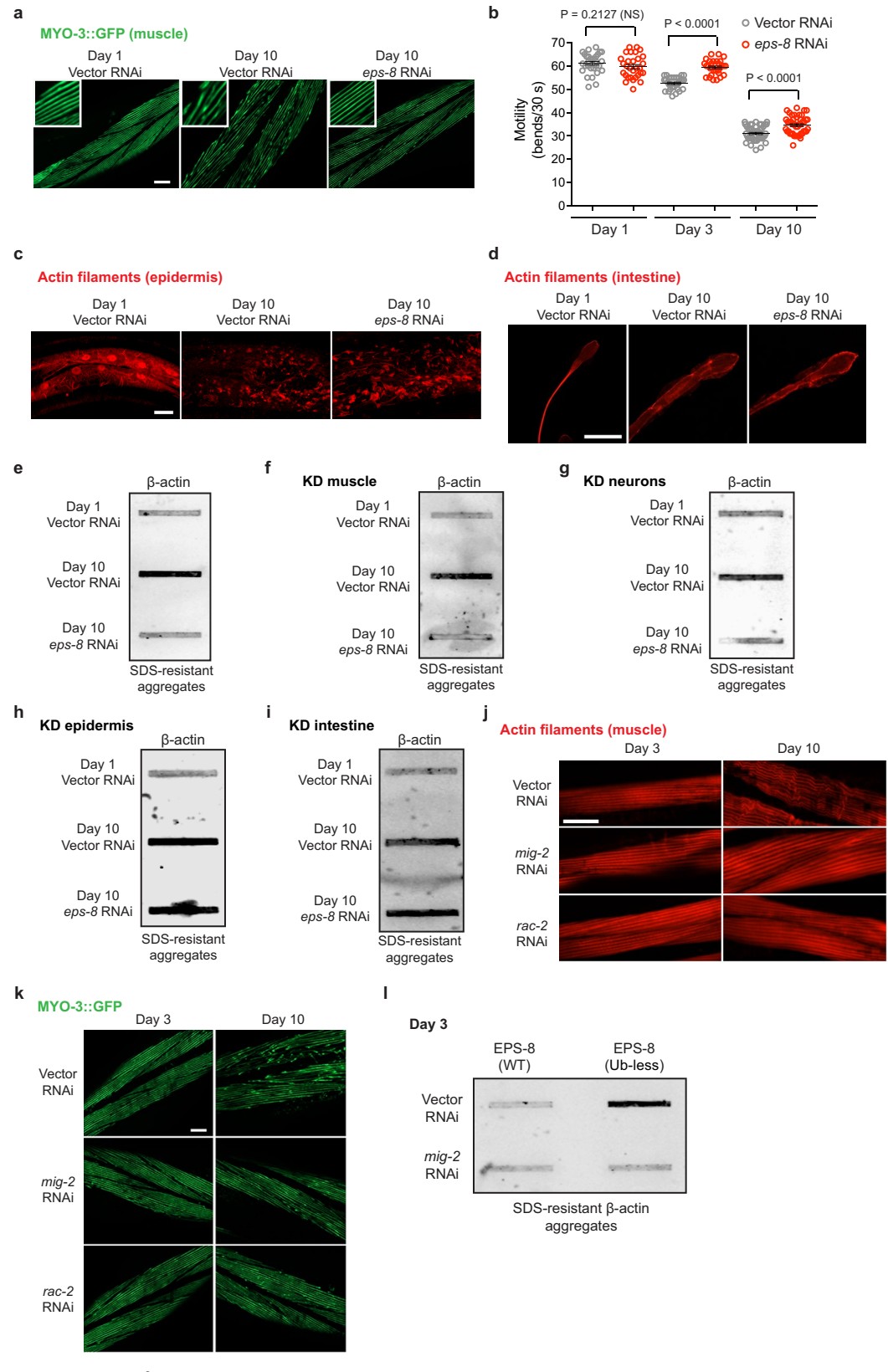

**Extended Data Fig. 10** | See next page for caption.

**Extended Data Fig. 10 | Lowering hyperactivated RAC signalling prevents alterations in actin networks induced by increased EPS-8 levels. a**, Myosin heavy chain tagged to GFP indicates destabilization of myosin filaments in muscle cells during ageing, whereas knockdown of *eps-8* during adulthood maintains organization of the myosin network. Scale bar, 20 μm. Representative of three independent experiments. **b**, Data are mean ± s.e.m. thrashing movements over a 30-s period on day 1 (*n* = 30 worms per condition, three independent experiments), day 3 (*n* = 30 worms per condition, three independent experiments) and day 10 (*n* = 45 worms per condition, three independent experiments) of adulthood. Knockdown of *eps-8* after development ameliorates the age-associated decline in motility (day 1 vector RNAi versus day 1 *eps-8* RNAi, *P* = 0.2127; day 3 Vector RNAi versus day 3 *eps-8* RNAi, *P* < 0.0001; day 10 Vector RNAi versus day 10 *eps-8* RNAi, *P* < 0.0001). *P* values were determined by two-sided *t*-test. **c**, Images of transgenic worms expressing LifeAct::mRuby in the epidermis. Scale bar, 20 μm. Representative of two independent experiments. **d**, Images of transgenic worms expressing LifeAct::mRuby in the intestine. Scale bar, 100 μm. Representative of two independent experiments. **e**, Filter trap analysis with an antibody to β-actin in wild-type worms. Knockdown of *eps-8* during adulthood reduces the amounts of actin aggregates in aged worms. Representative of six independent experiments. **f**, Filter trap analysis with an antibody to β-actin. Muscle-specific knockdown of *eps-8* during adulthood reduces the amounts of actin aggregates in aged worms. Representative of three independent experiments. **g**, Neuronal-specific knockdown of *eps-8* results in decreased actin aggregates during ageing. Representative of three independent experiments. **h**, Epidermal-specific knockdown of *eps-8* does not decrease age-related actin aggregation. Representative of three independent experiments. **i**, Intestinal-specific knockdown of *eps-8* does not decrease age-related actin aggregation. Representative of three independent experiments. **j**, Knockdown of either *mig-2* or *rac-2* during adulthood ameliorates the age-associated destabilization of actin filaments in muscle cells. RNAi treatment was initiated at day 1 of adulthood. Scale bar, 20 μm. Representative of two independent experiments. **k**, Knockdown of either *mig-2* or *rac-2* during adulthood ameliorates the age-associated destabilization of myosin filaments in muscle cells. Scale bar, 20 μm. Representative of two independent experiments. **l**, Filter trap with an antibody to β-actin in worms expressing endogenous wild-type EPS-8 or mutant Ub-less EPS-8 at day 3 of adulthood. Representative of three independent experiments. In all the experiments, RNAi was initiated at day 1 of adulthood.

# nature research

# Reporting Summary

Nature Research wishes to improve the reproducibility of the work that we publish. This form provides structure for consistency and transparency in reporting. For further information on Nature Research policies, see our Editorial Policies and the Editorial Policy Checklist.

## Statistics

For all statistical analyses, confirm that the following items are present in the figure legend, table legend, main text, or Methods section.

| n/a | Confirmed | |
|---|---|---|
| ☐ | ☒ | The exact sample size ($n$) for each experimental group/condition, given as a discrete number and unit of measurement |
| ☐ | ☒ | A statement on whether measurements were taken from distinct samples or whether the same sample was measured repeatedly |
| ☐ | ☒ | The statistical test(s) used AND whether they are one- or two-sided *Only common tests should be described solely by name; describe more complex techniques in the Methods section.* |
| ☒ | ☐ | A description of all covariates tested |
| ☐ | ☒ | A description of any assumptions or corrections, such as tests of normality and adjustment for multiple comparisons |
| ☐ | ☒ | A full description of the statistical parameters including central tendency (e.g. means) or other basic estimates (e.g. regression coefficient) AND variation (e.g. standard deviation) or associated estimates of uncertainty (e.g. confidence intervals) |
| ☐ | ☒ | For null hypothesis testing, the test statistic (e.g. $F$, $t$, $r$) with confidence intervals, effect sizes, degrees of freedom and $P$ value noted *Give P values as exact values whenever suitable.* |
| ☒ | ☐ | For Bayesian analysis, information on the choice of priors and Markov chain Monte Carlo settings |
| ☒ | ☐ | For hierarchical and complex designs, identification of the appropriate level for tests and full reporting of outcomes |
| ☒ | ☐ | Estimates of effect sizes (e.g. Cohen's $d$, Pearson's $r$), indicating how they were calculated |

*Our web collection on statistics for biologists contains articles on many of the points above.*

## Software and code

Policy information about availability of computer code

| | |
|---|---|
| Data collection | No software was used |
| Data analysis | We used GraphPad PRISM 6 for statistical analysis. GraphPad PRISM 6 software was also used to determine median lifespan and generate lifespan graphs. OASIS software (version 1) was used for statistical analysis to determine mean lifespan. For quantification of mCherry fluorescence signal, animals were outlined and quantified using ImageJ software (version 1.51s). For protein identification and label-free quantification (LFQ) in ubiquitin-proteomics experiments, we used the LFQ mode and MaxQuant (version 1.5.3.8) default settings. For proteomics datasets of total protein levels in aging and proteasome-less worms, we used Spectronaut 11 (Biognoys) with the BGS Factory Settings and MaxQuant (version 1.5.3.8) with default settings, respectively. <br><br> All the downstream analyses of the resulting output were performed with R program (version 4.0.5) and Perseus (version 1.6.2.3). Custom code used in this article can be accessed at https://github.com/Vilchezlab/UbProteomics2021. For tissue expression analysis, we used the website http://worm.princeton.edu10. |

For manuscripts utilizing custom algorithms or software that are central to the research but not yet described in published literature, software must be made available to editors and reviewers. We strongly encourage code deposition in a community repository (e.g. GitHub). See the Nature Research guidelines for submitting code & software for further information.

## Data

Policy information about availability of data

All manuscripts must include a data availability statement. This statement should provide the following information, where applicable:

- Accession codes, unique identifiers, or web links for publicly available datasets
- A list of figures that have associated raw data
- A description of any restrictions on data availability

There is no restriction on data availability. Source data are provided with this paper. Readers can interact with the ubiquitin and global proteomics data using the following Shiny Web apps by downloading the datasets provided in the apps: https://vilchezlab.shinyapps.io/shiny-volcanoplot/ and https://vilchezlab.shinyapps.io/shiny-heatmap/. All the proteomics data have been deposited to the ProteomeXchange Consortium via the PRIDE partner repository with the dataset identifiers PXD024338 (ubiquitin proteomics of aging and long-lived worms), PXD025128 (global protein proteomics of aging and long-lived worms), PXD024094 (ubiquitin proteomics upon rpn-6 RNAi), PXD024095 (global protein proteomics upon rpn-6 RNAi), PXD024093 (immunoprecipitation Lys48-linked polyUb), and PXD024045 (immunoprecipitation Lys63-linked polyUb). MS2 spectra in proteomics experiments were searched against the C. elegans Uniprot database (https://www.uniprot.org/proteomes/UP000001940). For worm tissue expression analysis, we used the database http://worm.princeton.edu.

# Field-specific reporting

Please select the one below that is the best fit for your research. If you are not sure, read the appropriate sections before making your selection.

☒ Life sciences ☐ Behavioural & social sciences ☐ Ecological, evolutionary & environmental sciences

For a reference copy of the document with all sections, see nature.com/documents/nr-reporting-summary-flat.pdf

# Life sciences study design

All studies must disclose on these points even when the disclosure is negative.

| | |
|---|---|
| Sample size | No statistical methods were used to predetermine sample size. Exact sample sizes are indicated in the corresponding figure legends. For ubiquitin and total proteomics experiments, sample sizes were chosen based on prior studies on ubiquitin proteomics in mammalian cells (Kim W et al, Molecular Cell 44, 325-340 (2011)), global proteomics changes during aging in C. elegans (Narayan V et al, Cell Systems 3, 144-159 (2016); Walther D.M. et al, Cell 161, 919-932 (2015)) and our previous work on proteomics analysis of C. elegans (Lee H.J. et al, Nature Metabolism 1, 790-810 (2019)). Sample sizes for filter traps, western blot, qPCR, motility, lifespan and quantification of fluorescence reporters in C. elegans were determined according to our extensive laboratory experience and other studies using these assays (Lee H.J. et al, Nature Metabolism 1, 790-810 (2019); Noormohammadi A et al, Nature Communications 7, 13649 (2016); Amrit F.R. et al, Methods 68, 465–475 (2014); Zheng Q et al, Cell 174, 870-883 (2018)). |
| Data exclusions | No data were excluded from the analyses. |
| Replication | At least three independent experiments for each assay were performed to verify the reproducibility of the findings (if there were two independent experiments, this was also noticed in the figure legend). All the attempts of replication gave a similar outcome. Lifespan assays were done at least 2 times with 96 animals per each condition. Exact numbers and statistics are provided in figure legends and supplementary data. |
| Randomization | For lifespan experiments, imaging experiments, and motility assays, worms were synchronized by picking young hermaphrodites adults and let them lay eggs for 6 hours. These young hermaphrodites were randomly picked from our maintenance plates. After egg laying for 6 hours, larvae were raised until adulthood and adult worms were then randomly assigned to the different treatment conditions. The different conditions were assessed in random order. For proteomics, filter trap assays and western blot experiments, worms were synchronized by bleaching of young hermaphordite worms followed by L1 starvation standard procedures. These young hermaphrodite worms were obtained by transferring random chunks of agar from maintenance plates and let them grow until we have sufficient young hermaphrodites worms for bleaching. After bleaching worms and obtaining synchronized adults, these young worms were randomly assigned to the different treatment conditions. Then, the samples were collected, lysed and analyzed in random order. |
| Blinding | For proteomics experiments, sample collection was not performed in a blinded manner as aged wild-type worms, long-lived mutants and RNAi-treated worms have obvious phenotypes that revealed the sample identity (i.e., ages, strains and treatments). Once the samples were processed for proteomics, the mass spectrometry was performed by the staff of the CECAD Proteomics Facility in a blinded manner. Blinded analysis of proteomics data was not feasible as it required integrative analysis of different conditions.<br><br>qPCR, filter trap and western blot experiments were not performed in a blinded manner as they rely on objective instrument measurements and/or provide indirect outputs. For lifespan, microscopy, motility and bacterial colonization, the experiments were not performed in a blinded manner given the nature of the reagents used (e.g., HT115 E. coli carrying empty vector or RNAi clones have to be refreshed every day during early adulthood and very other day at later stages). However, worms were randomly assigned to the different treatment conditions and the different conditions were assessed in random order. Moreover, all the critical experiments were repeated independently by at least 2 of the investigators. |

The investigators were not blinded during data analysis due to feasibility of the analysis.

# Reporting for specific materials, systems and methods

We require information from authors about some types of materials, experimental systems and methods used in many studies. Here, indicate whether each material, system or method listed is relevant to your study. If you are not sure if a list item applies to your research, read the appropriate section before selecting a response.

## Materials & experimental systems

| n/a | Involved in the study |
|-----|----------------------|
| ☐ | ☒ Antibodies |
| ☒ | ☐ Eukaryotic cell lines |
| ☒ | ☐ Palaeontology and archaeology |
| ☐ | ☒ Animals and other organisms |
| ☒ | ☐ Human research participants |
| ☒ | ☐ Clinical data |
| ☒ | ☐ Dual use research of concern |

## Methods

| n/a | Involved in the study |
|-----|----------------------|
| ☒ | ☐ ChIP-seq |
| ☒ | ☐ Flow cytometry |
| ☒ | ☐ MRI-based neuroimaging |

## Antibodies

**Antibodies used**

We used the following antibodies in this study:
anti-IFB-2 (Developmental Studies Hybridoma Bank, MH33, 1:1,000. RRID: AB_528311).
anti-EPS8L2 (Abcam, ab85960, 1:1,000. RRID: AB_1924963)
anti-α-tubulin (Sigma, T6199, 1:5,000. RRID: AB_477583)
anti-JNK (Cell Signaling, #9252, 1:1,000. RRID: AB_2250373)
anti-Phospho-JNK (Thr183/Tyr185) (Cell Signaling, #9251, 1:1,000. RRID: AB_331659)
anti-ubiquitin (Sigma, #05-944, Clone P4D1-A11, 1:1,000. RRID: AB_441944)
anti-GFP (AMSBIO, 210-PS-1GFP, 1:5,000. RRID: AB_10013682)
anti-β-actin (Abcam, ab8226, 1:5,000. RRID: AB_306371)
anti-Ubiquitin Antibody, Lys48-Specific, clone Apu2 (Merck, #05-1307, 1:50. RRID: AB_1587578)
anti-Ubiquitin Antibody, Lys63-Specific, clone Apu3 (Merck, #05-1308, 1:50. RRID: AB_1587580)
anti-FLAG antibody (SIGMA, F7425, 1:100. RRID: AB_439687)

**Validation**

Validation of antibodies were done by the stated manufacturer's and supported by the publications indicated in the manufacturer's website, the Resource Identification Portal (RRID) and other publications using C. elegans as a model organism (including our previous publications).

* anti-IFB-2 (Developmental Studies Hybridoma Bank, MH33, 1:1,000. RRID: AB_528311). The antibody was used according to the manufacturer's instructions for western blot in C. elegans. References: PMID:31414984, PMID:31414984.

* anti-EPS8L2 (Abcam, ab85960, 1:1,000. RRID: AB_1924963) was validated in C. elegans by the data presented in this manuscript (1- the increased in EPS-8 levels observed by proteomics during aging correlated with increased levels by western blot, 2- ubiquitin-less EPS-8 protein cannot be degraded by the proteasome and accordingly western blot shows increase levels of EPS-8 in ubiquitin-less EPS-8 mutant strain).

* anti-α-tubulin (Sigma, T6199, 1:5,000. RRID: AB_477583). The antibody was validated as a loading control for western blot analysis in C. elegans in our previous publications: PMID: 32451438, PMID: 27892468.

* anti-JNK (Cell Signaling, #9252, 1:1,000. RRID: AB_2250373). According to the manufacturer and supported by previous studies, this antibody has been validated for western blot in multiple species including C. elegans (PMID: 15767565, PMID:23525221, PMID:23715867, PMID:24773344, PMID:25209287, PMID:25849727, PMID:26153447, PMID:26295369, PMID:26881311, PMID:27145004, PMID:27253999).

* anti-Phospho-JNK (Thr183/Tyr185) (Cell Signaling, #9251, 1:1,000. RRID: AB_331659). According to the manufacturer and supported by published studies, this antibody has been validated for western blot in multiple species including C. elegans (PMID: 15767565, PMID:24635351, PMID:24773344, PMID:25164676, PMID:25209287, PMID:25849727, PMID:25885794, PMID:26132918, PMID:26153447, PMID:27145004, PMID:27253999).

*anti-ubiquitin (Sigma, #05-944, Clone P4D1-A11, 1:1,000. RRID: AB_441944). Since ubiquitin is evolutionary conserved among species, this antibody is predicted to work on a wide range of species. Accordingly, it has been validated for western blot application in multiple species (PMID:27552055, PMID:29499138, PMID:29547723). Its application for western blot in C. elegans has been validated in the experiments presented in this manuscript. For instance, it can be used to detect ubiquitinated proteins as demonstrated in the western blots presented in Figs. 1g-i where global changes in ubiquitination correlated with the results observed by ubiquitin proteomics Moreover, we could reverse loss of Ub-protein levels detected by western blot by adding broad-spectrum deubiquitinase inhibitor (Fig. 1j).

* anti-GFP (AMSBIO, 210-PS-1GFP, 1:5,000. RRID: AB_10013682). This antibody has been validated for filter trap and western blot in C. elegans in our previous publications: PMID: 27892468; PMID: 30038412

* anti-β-actin (Abcam, ab8226, 1:5,000. RRID: AB_306371). This antibody has been validated for use in C. elegans in our previous publication PMID: 30038412

* anti-Ubiquitin Antibody, Lys48-Specific, clone Apu2 (Merck, #05-1307, 1:50. RRID: AB_1587578). Since Lys48-linked polyubiquitin chains are evolutionary conserved among species, this antibody is predicted to work on a wide range of species. Accordingly, it has been validated for western blot and immunoprecipitation application in multiple species (PMID:27523608, PMID:28347402, PMID:28594325, PMID:28712572, PMID:28943312, PMID:29024643, PMID:29153505, PMID:30581143, PMID:31042464, PMID:31613024, PMID:31825842). We have validated its use for immunoprecipitation experiments in C. elegans in this manuscript by confirming that age-dysregulated proteasome targets contain Lys48-linked polyubiquitin chains (Fig. 2b).

* anti-Ubiquitin Antibody, Lys63-Specific, clone Apu3 (Merck, #05-1308, 1:50. RRID: AB_1587580). Since Lys63-linked polyubiquitin chains are evolutionary conserved among species, this antibody is predicted to work on a wide range of species. Accordingly, it has been validated for western blot and immunoprecipitation application in multiple species (PMID:27523608, PMID:28244869, PMID:28594325, PMID:28712572, PMID:28943312, PMID:29153505, PMID:29547723, PMID:29576527, PMID:29861391, PMID:30893611, PMID:30901564, PMID:31042464, PMID:31606272, PMID:31613024, PMID:31825842). We have validated its use for immunoprecipitation experiments in C. elegans in this manuscript by confirming that most of the age-dysregulated proteasome targets do not contain Lys63-linked polyubiquitin chains (Extended Data Fig. 4d).

* anti-FLAG antibody (SIGMA, F7425, 1:100. RRID: AB_439687). We have validated the use of anti-FLAG antibody as a negative control for immunopreciptation experiments followed by label-free proteomics in our previous publications (PMID: 32451438, PMID: 30038412)

# Animals and other organisms

Policy information about <u>studies involving animals</u>; <u>ARRIVE guidelines</u> recommended for reporting animal research

| Laboratory animals | Caenorhabditis elegans strains were used in this study. For all the experiments, we used hermaphrodites worms. For label-free quantitative proteomics, wild-type, daf-2(e1370) and eat-2(ad1116) worms were collected with M9 buffer at day 1, 5, 10 and 15 of adulthood. For proteomics analysis of proteasome-less worms, we collected day 5 adult worms treated with either Vector RNAi or rpn-6 RNAi. Lifespan analysis was started from day 1 of adulthood. For all the other experiments, the specific age is indicated in the corresponding figures and/or figure legends.

The C. elegans strains used in this study are:

Wild-type (N2)
DA1116 (eat-2 (ad1116)II)
RW1596 (myo-3(st386)V; stEx30[myo-3p::GFP:: myo-3 + rol-6(su1006)])
CF1041 (daf-2(e1370)III)
BJ49 (kcIs6[ifb-2p::ifb-2a::CFP]IV)
BJ186 (kcIs30[ifo-1p::ifo-1::YFP;myo-3p::mCherry::unc-54]III)
BJ324 (kcEx78[ifc-1p::ifc-1::eGFP; unc- 119(ed3)+];unc-119(ed3)III)
BJ316 (ifc-2(kc16[ifc-2a/e::YFP])X)
BJ312 (kcIs40[ifp-1p::ifp-1::eGFP]IV)
AGD1657 (unc-119(ed3)III; uthSi13[gly-19p::LifeAct::mRuby::unc-54 3'UTR::cb-unc-119(+)]IV)
AGD1654 (unc-119(ed3)III; uthSi10[col-19p::LifeAct::mRuby::unc-54 3'UTR::cb-unc-119(+)]IV)
DVG197 (N2, ocbEx162[sur-5p::ifb-2, myo-3p::GFP])
DVG198 (N2, ocbEx163[sur-5p::ifb-2, myo-3p::GFP])
DVG9 (N2, ocbEx9[myo3p::GFP])
VP303 (rde-1(ne219)V; kbIs7[nhx-2p::rde-1 + rol-6(su1006)])
WM118 (rde-1(ne300)V; neIs9[myo-3p::HA::RDE-1 + rol-6(su1006)])
TU3401 (sid-1(pk3321)V; uIs69[pCFJ90(myo-2p::mCherry)+unc-119p::sid-1])
NR222 (rde-1(ne219)V; kzIs9 [(pKK1260) lin-26p::NLS::GFP + (pKK1253) lin-26p::rde-1 + rol-6(su1006)])
VDL07 (ifb-2(syb2876)II)
VDL05 (eps-8(syb2901)IV)
VDL06 (eps-8(syb2901, syb3149)IV)
VDL08 (ifb-2(syb3973)II)) |

| Wild animals | The study did not involve wild animals. |

| Field-collected samples | The study did not involve samples collected from the field. |

| Ethics oversight | In this research, we used invertebrate C. elegans as an organismal model and no ethical approval was required. According to the "Zentrale Kommission für die Biologische Sicherheit" (ZKBS), the responsible entity inside the Bundesamt für Verbraucherschutz und Lebensmittelsicherheit to assess the risk of Genetically Modified Organisms (GMO), genetic work with C. elegans is classified as risk group 1 (biological safety level 1: S1). Accordingly, we performed work on C. elegans in a S1-laboratory. The use of GMO in Germany |

is regulated by the "Gentechnik-Gesetz", and we followed the guidelines applying to S1 work with GMO (i.e., documentation of the project and of the, exact description of the creation and maintenance of the genetic modification or correct waste treatment).

Note that full information on the approval of the study protocol must also be provided in the manuscript.

