## [Peer Review File · Nature]

Manuscript Title: Rewiring of the ubiquitinated proteome determines aging in *C. elegans*

Reviewer Comments & Author Rebuttals

Reviewer Reports on the Initial Version:

Referees' comments:

Referee #1 (Remarks to the Author):

In this study, Koyunku et al. performed multiple proteomics screens to characterize the dysregulation of ubiquitination during aging. The immense amount of data in the manuscript is commendable, starting first with analyzing ubiquitin status of the proteome during four time points of aging, and then extending these studies to two additional paradigms of longevity: calorie restriction (*eat-2*) and IIS (*daf-2*). To further add weight to their findings, they performed general proteomics analysis during aging to complement their ubiquitin screens to directly determine whether changes in ubiquitin status affected protein levels during aging. They followed up these beautifully crafted datasets with a tremendous amount of analysis and description, which is truly spectacular, although on the very rare occasion, there is some over-analysis of data and premature conclusions made (see specific points below). As if this curation of data were not enough, the authors then followed up on two specific candidate proteins, and performed a very thorough analysis of the genetic mechanism whereby IFB-2 and EPS-8 affect the aging process. This manuscript is very well put together with a very rich and valuable dataset for the field of protein homeostasis and aging and is further strengthened by the characterization of two novel mechanistic pathways that alter aging through cytoskeletal remodeling. It is sure to be a major contribution to the fields of aging, protein homeostasis, and cytoskeletal biology and will make a very strong publication for Nature. Below are just some points that can be easily addressed during revision to strengthen the manuscript even further.

Major comments:

- 1) I am not convinced with the analysis of tissue-specific changes to the ubiquitin proteome. The authors performed this analysis by using previously published datasets to determine which proteins are expressed in a tissue-specific manner, then analyzing the changes that happened for these proteins. The correct way to perform this would be to use tissue-specific protocols and performing proteomics in a tissue-specific manner. This is of course too large an ask for a revision, but minimally, the authors need to address the limitations of their analysis. Ideally, it would be best to remove this analysis and data completely, as the conclusions drawn from this analysis are not strong – showing all tissues displayed changes in ubiquitin status with age with perhaps muscle and intestine showing slightly more changes than neurons or germline – is not a novel or interesting conclusion, especially from an analysis method with numerous caveats.
- 2) The manuscript is so data-heavy in the beginning, and a lot of the data is presented in a manner that is not an easy read for a broad audience. For example, it requires the downloading of large excel spreadsheets that we must comb through to get an understanding of the data. I recommend that the authors present some heat maps or other ways to translate some of the spreadsheet data into an easily digestible format. The authors have plenty of room in the Extended Data figures to try and present the data in an easier format, especially because most of the current Extended Data figures are very small (some only on panel), and can be combined into 2-3 figures.
- 3) One concern with the IFB-2 and EPS-8 ubiquitinless mutants is that it may impact the function of the molecules beyond just preventing its degradation. The authors should at least textually comment on why standard overexpression constructs were not synthesized. Ideally, if they can perform validations that these mutations don't affect canonical function of the proteins (e.g. rescue experiments), that would be helpful.
- 4) The authors perform visualization of actin filaments solely in the muscle, despite seeing that *eps-8* knockdown in muscle and neurons are sufficient to cause a functional phenotype. While neuronal actin integrity is challenging due to the small size of neurons, this phenotype begs to interrogate actin in

other tissues. Can the authors attempt to image actin in other cell types, such as the hypodermis? Inclusion of the intestine may also be a great control since they see no lifespan effect with intestine-specific knockdown. This case is made even more important considering the SDS-resistant actin aggregates found in the EPS-8 ubiquitin-less mutants. According to the muscle actin images, no aggregates are visible (just a deformed cytoskeletal network). Where are these aggregates coming from? Are there aggregates in other cell types?

5) RAC components traditionally are positive regulators of actin homeostasis, yet the authors show that knockdown of RAC components can phenocopy eps-8 knockdown, a condition that promotes cytoskeletal integrity. This needs to be addressed. It would also be useful to see cytoskeletal imaging with at least one of the RAC component knockdowns.

Minor comments:

1) The presentation of the volcano plots does not seem very helpful for Figure 1. While I appreciate the authors trying to give an overview of the data, it does not really highlight anything important or useful for the reader. Is it possible to make the graphs more meaningful or simply present the data in a different way? Perhaps highlighting some critical targets that were identified – especially the ones that were discussed in the text (e.g. similar to Fig. 3e-f).

2) Combine extended data figures into 2-3 figures to make space for more important things (see major comment 2).

Referee #2 (Remarks to the Author):

This manuscript from Vilchez and colleagues describes the finding that steady state levels of ubiquitinated substrates are altered in *C. elegans* as a function of aging. Moreover, these ubiquitinated substrates can be rescued through dietary restriction or in genetic models of lifespan extension. The authors came to this observation through diGLY proteomics, a mass spectrometry method which allowed them to survey thousands of site specific ubiquitination events. Differentially ubiquitinated substrates were subsequently confirmed using linkage specific antibody pulldowns and western blot analysis of total protein levels. Functionally, the ubiquitin substrates IFB-2 and EPS-8 were both tested by genetic knockdown to demonstrate that suppressing their accumulation extended lifespan. Interestingly, the effects of each substrate manifests in a tissue dependent manner, within intestine and muscle/neuronal cells respectively.

A key challenge for this paper is that the mechanism broadly decreasing protein ubiquitination remains to be resolved, and may itself be tissue dependent. These findings come on the heels of previous work from a subset of these authors involving the proteasome assembly factor Rpn6, suggesting that proteasome activity plays at least some role. Steady state levels of ubiquitinated substrates may be influenced by many other activities and there is likely a complex interplay involving transcription, translation, ubiquitination, degradation. All of these processes are dynamic, meaning that demonstration of the proposed mechanism requires an understanding of what process becomes deficient in aging cells to tilt the equilibrium in the way described:

- It seems surprising that IFB-2 and EPS-8 would not have been previously identified as factors whose knockdown would extend lifespan, or whose protein levels increase with age. Please explain how or why these have been overlooked in previous screening work by the field.

- Given the complex temporal dynamics observed in Fig 1e-g, a more granular time course should be performed to understand when declines occur in WT worms and how Ub levels change as a function of time, particularly in the *daf-2* model.

- As written, the text strongly emphasizes the ubiquitinated proteome, in spite of having performed global proteome analysis in parallel with some or all of the diGLY studies. From the figures and text, it was difficult to ascertain how protein and diGLY levels compare for individual proteins and how different lysine modification sites respond within affected proteins. While reflected in tabular format, this information is more important than many of the summary panels presented in figures 1-2.

- In terms of system dynamics, what does mRNA expression look like for each of the different ubiquitin genes? Are either the ribosomal fusion or stress response ubiquitin genes differentially regulated with age in the WT or mutant strains?
- In terms of system dynamics, what is the t_{1/2} of ubiquitin itself in worms of different ages and genetic backgrounds tested here? This is particularly important for the claims made in lines 368-374.
- Can the authors differentiate between the possibilities that the decrease in conjugated ubiquitin observed in aged WT worms results from elevated DUB activity, rather than a deficiency in the ubiquitin conjugation system?
- The statement on lines 125-127 does not accurately reflect what the western blots show.
- The assertions in lines 144-145 seem to be assumptions, since the proteome and diGLY proteome of muscles and neurons were not directly measured.
- Do the authors believe the underlying mechanism proposed to alter steady state protein ubiquitination levels for proteins such as IFB-2 and EPS-8 is conserved across different cell types? Or is it possible that this is tissue specific? In a similar vein, are these effects observed in a cell autonomous manner or only when cells are living in the context of an intact organism?
- The assertion on lines 172-175 is overly simplistic and inappropriate given the endpoint experiment carried out. The ubiquitin system functions on the order of minutes to degrade individual substrates and is deeply intertwined with transcription, translation, protein localization and turnover processes. This paper would be improved if it approached modelling of this equilibrium in a more sophisticated manner.
- It would help if the text explained in more detail how Rpn6 knockdown alters the baseline proteome. For example, what is the proposed impact of USP5 expression changes on phenomena being studied here?
- The concluding statement in lines 208-209 oversimplifies a complex and dynamic process.
- The text indicates that the data will be made available in PRIDE, although this should have been done prior to initial submission and data access provided to reviewers through in a PW protected manner as is now standard. More concerning is the comment that upon publication, additional data will be available upon request. For a paper such as this, this approach is no longer sufficient. The authors should assemble an interactive tool (e.g. Shiny Web app) so that readers of the paper can interact with the global proteome and diGLY proteome data and draw their own conclusions.
- Extending from the concern above, the Supplemental Tables have substantive deficits that make it impossible for an expert in mass spectrometry to judge the quality of these data. For example:
 - o Supplemental Tables 1 & 3 do not provide the peptide sequences that were observed in the diGLY data. This is arguably the most critical piece of information to have in this table for purposes of assessing data integrity.
 - o Supplemental Table 2 does not provide any indication of the number of peptides used in assembling the quantification data per protein.

Referee #3 (Remarks to the Author):

Review Koyuncu et al Nature

The authors report the results of a systematic proteomic analysis in which they quantified changes in the ubiquitin-modified signatures during *C. elegans* ageing. They find that ubiquitination gets

downregulated or upregulated for hundreds of proteins between day-5 and day-15 of adulthood. They focus on 10 specific proteins that become under-ubiquitinated during ageing, or hyper-ubiquitinated when the proteasome is inhibited, and show that knocking down the corresponding genes extends lifespan. For two of them, the intermediate filament IFB-2 enriched in the intestine and the actin-binding protein EPS-8, they further identify the lysine residues mediating ubiquitination and show that CRISPR/Cas9 changes of these lysines into arginines reduces lifespan. Last but not least, they examine why changes in the ubiquitination pattern of both proteins alters lifespan. They find that overexpression of IFB-2 cause the aggregation of other intestinal intermediate filaments and reduce animal resistance to intestinal colonization by *E. coli*. Likewise, they find that EPS-8 knockdown upregulates Rac and JNK/KGB-1 signaling in neurons and muscles, and reduces the disorganization of muscle myofilaments.

Hence the authors went from a proteome-wide study of protein ubiquitination during ageing to the cellular and molecular characterization of two proteins, showing why their ubiquitination matters during normal or genetically modified ageing. This is a remarkable achievement, which will be of interest to a very wide audience as a central quest in the ageing field is to define how to prolong healthy ageing and to identify pathways that could mediate so. Furthermore, experiments are overall very well done and the manuscript is well written. I only have rather minor comments. Of note, *C. elegans* intermediate filaments in the intestine, epidermis and excretory cell do not resemble any classical vertebrate intermediate filaments; in fact, they are related to lamins, such that the significance of their findings on IFB-2 will await confirmation of a related study in mouse (for instance) with keratins, vimentin, desmin, nestin, GFAP and the like. On the other hand, the results obtained with EPS-8 should be transposable in vertebrates.

Comments:

1. The authors underline that muscles and intestine exhibited a higher number of proteins with altered ubiquitination, with neurons and germline cells coming next. I find it surprising that the skin would not be represented in their list (if anything given the billions involved in the cosmetic industry to make our skin looking forever young!). For instance, EPS-8 plays also an important role in the epidermis and remains expressed in adults.
2. A question concerning the method used to synchronize animals for proteomic studies. The authors used FUDR to prevent the development of the progeny, which is indeed a classical method in the field. I have a potential concern with this approach, inasmuch FUDR inhibits mitochondria and makes animals sterile, both of which have impact on ageing. The issue is whether the treatment could affect the proteins identified in their screens. The ten proteins on which they focus are ok, but what about the hundred others? It would be important to have an assessment of it by probing a small subset of their dataset taking proteins for which antibodies are available.
3. The procedure used for immunoblots involves a centrifugation of worm extracts at 8000g for 5 min to collect the supernatant. The antibody MH33, which they used to probe the abundance of IFB-2, was generated against a highly insoluble worm fraction (Francis and Waterston, JCB 1991). The results shown in Fig. 4b,d are compelling but I was wondering whether the bulk of IFB-2 changes and whether they could correspond to squiggles.
4. Some lifespan assays have been done with the non-integrated lines DVG9, DVG197, and DVG198. The legends do not specify which panels report assays using these strains.
5. Related to this issue, the *kcIs6* transgene was generated starting from an extrachromosomal array and is thus present in multiple copies. Could this induce aggregation on its own, and would IFB-2::CFP expressed as a single copy be prone to aggregation as well?
6. I randomly checked the sequences used for qPCR assays taking *eps-8* as a test and could not find the position of the forward primer AGAAGAAAAGAAGTGGATTCCGA ACT. Please check all sequences mentioned in Table S11.

Author Rebuttals to Initial Comments:

Referees' comments:

Referee #1 (Remarks to the Author):

In this study, Koyunku et al. performed multiple proteomics screens to characterize the dysregulation of ubiquitination during aging. The immense amount of data in the manuscript is commendable, starting first with analyzing ubiquitin status of the proteome during four time points of aging, and then extending these studies to two additional paradigms of longevity: calorie restriction (*eat-2*) and IIS (*daf-2*). To further add weight to their findings, they performed general proteomics analysis during aging to complement their ubiquitin screens to directly determine whether changes in ubiquitin status affected protein levels during aging. They followed up these beautifully crafted datasets with a tremendous amount of analysis and description, which is truly spectacular, although on the very rare occasion, there is some over-analysis of data and premature conclusions made (see specific points below). As if this curation of data were not enough, the authors then followed up on two specific candidate proteins, and performed a very thorough analysis of the genetic mechanism whereby IFB-2 and EPS-8 affect the aging process. This manuscript is very well put together with a very rich and valuable dataset for the field of protein homeostasis and aging and is further strengthened by the characterization of two novel mechanistic pathways that alter aging through cytoskeletal remodeling. It is sure to be a major contribution to the fields of aging, protein homeostasis, and cytoskeletal biology and will make a very strong publication for Nature. Below are just some points that can be easily addressed during revision to strengthen the manuscript even further.

Major comments:

- 1) I am not convinced with the analysis of tissue-specific changes to the ubiquitin proteome. The authors performed this analysis by using previously published datasets to determine which proteins are expressed in a tissue-specific manner, then analyzing the changes that happened for these proteins. The correct way to perform this would be to use tissue-specific protocols and performing proteomics in a tissue-specific manner. This is of course too large an ask for a revision, but minimally, the authors need to address the limitations of their analysis. Ideally, it would be best to remove this analysis and data completely, as the conclusions drawn from this analysis are not strong – showing all tissues displayed changes in ubiquitin status with age with perhaps muscle and intestine showing slightly more changes than neurons or germline – is not a novel or interesting conclusion, especially from an analysis method with numerous caveats.

*Reviewer #1 is absolutely right and this concern was also raised by Reviewer #2. We have now made clear in the text that **Extended Data Fig. 6a** presents a bioinformatic analysis combining our proteomics data with previously published datasets of tissue-specific expression and toned down the conclusions from this analysis. For instance, we have removed the statement regarding that the muscle and*

intestine have slightly more changes than neurons or germline. We agree that this conclusion requires ubiquitin proteomics experiments in a tissue-specific manner, which are not feasible due to the lack of protocols to isolate most of the tissues (e.g., muscle, epidermis) or obtain sufficient protein amounts of other tissues such as the germline and intestine for diGly enrichment.

*Nevertheless, we think that it is important to keep this bioinformatic analysis as a first indication that the decline in the steady-state levels of ubiquitination is not tissue-specific and occur through the entire organism. In these lines, we have now included the epidermis in the bioinformatic analysis as requested by Reviewer #3 (please see **Extended Data Fig. 6a and Supplementary Table 10**). To confirm that the decline in ubiquitination occurs across tissues, we have now assessed global levels of ubiquitinated proteins in distinct isolated tissues and body parts of *C. elegans* (**Extended Data Fig. 6b**). The text now says: "Using previously published datasets of tissue-specific expression³⁷, we classified proteins that exhibit differences in their Ub-peptides according to the tissues where these proteins are expressed. The bioinformatic analysis indicated that distinct tissues such as the germline, muscle, intestine, epidermis or neurons express multiple proteins that contain downregulated Ub-peptides with age (**Extended Data Fig. 6a and Supplementary Table 10**). Similar to whole organism lysates, we observed a decrease in the global amounts of Ub- proteins in isolated germlines, intestines and heads during aging (**Extended Data Fig. 6b**), supporting that this process occurs through the organism".*

- 2) The manuscript is so data-heavy in the beginning, and a lot of the data is presented in a manner that is not an easy read for a broad audience. For example, it requires the downloading of large excel spreadsheets that we must comb through to get an understanding of the data. I recommend that the authors present some heat maps or other ways to translate some of the spreadsheet data into an easily digestible format. The authors have plenty of room in the Extended Data figures to try and present the data in an easier format, especially because most of the current Extended Data figures are very small (some only on panel), and can be combined into 2-3 figures.

We really appreciate this suggestion from Reviewer #1. We have now replaced volcano plots by heat maps allowing us to present more visibly the global changes in the ubiquitinated proteome discovered in our proteomics experiments. We have also included additional heat maps to visualize other relevant comparisons at the proteome level from all the proteomics analysis included in the Supplementary Tables:

- *We have now replaced the 9 individual volcano plots presented in Figure 1b of our first submission by heat maps (please see **Fig. 1b**). These heat maps show in an easier format the widespread changes identified by ubiquitin proteomics:*
 - 1) *The steady-state amounts of numerous Ub-peptides significantly changed with age in wild-type and long-lived mutants when compared with their respective genetic background at day 1 of adulthood,*
 - 2) *In wild-type worms, the total number of differentially abundant Ub-peptides increased after day 5 of adulthood and most of these changes were linked with a downregulation in the steady-state levels of Ub-peptides,*
 - 3) *In contrast, long-lived mutants had fewer*

number of downregulated Ub-peptides during aging. In fact, *daf-2* had an increase in the number of upregulated Ub-peptides with age. Together with the bar graphs presented in **Fig. 1c-d**, these heat maps provide a meaningful summary of the main findings from **Supplementary Table 1** at the whole proteome level.

- To ascertain how ubiquitination and protein levels compare for individual proteins through the proteome, we have now included heat maps that integrate ubiquitin and global proteomics data comparing different ages of the distinct strains with day 1 worms of the respective genetic background (**Extended Data Fig. 1b**). These heat maps visualize that differences in the levels of Ub-peptides often do not correlate with a similar change in the total amounts of the respective protein. As such, these graphs summarize the information included in **Supplementary Tables 1-3**.
- We have now replaced the volcano plot presented in the Fig. 2a of our first submission showing widespread changes in the levels of Ub-peptides comparing aged (day 15) and young (day 5) wild-type worms by a heat map for better visualization of these global changes (**Fig. 1e**). Moreover, this heat map also integrates data of the corresponding total protein levels for all the differentially abundant Ub-peptides in aged worms. Besides showing that aging is particularly associated with a decline of Ub-peptides in wild-type worms, the heat map further supports that not all the differences in Ub-peptides can be simply ascribed to a similar change in the total protein levels. Thus, **Fig. 1e** summarizes **Supplementary Tables 4-5**.
- We have now included a heat map presenting differentially abundant Ub-peptides in aged (day 15) wild-type worms and comparison with age-matched *eat-2* and *daf-2* mutants (**Fig. 1f**). This heat map shows that DR and reduced IIS rescues age-related changes in the Ub-proteome. Detailed information about the specific number of rescued changes by each pro-longevity pathway are presented in Venn diagrams (**Extended Data Fig. 1d**). Taken together, these graphs summarize **Supplementary Table 6**.
- In addition, we have also included heat maps representing significant age-related changes in the protein levels of E3 ubiquitin ligases and DUBs (**Extended Data Fig. 4a-b**), summarizing **Supplementary Table 7**.
- We have now added a heat map representing differentially abundant proteins upon *rpn-6* RNAi (**Extended Data Fig. 5a**). Moreover, the Venn diagram presented in **Fig. 2b** summarizes our integrated analysis of proteomics data from aging and proteasome-less worms to define proteasome targets that become less ubiquitinated and degraded with age. The Venn diagram also highlights the 10 age-dysregulated proteasome targets identified in our analysis. Thus, these graphs summarize **Supplementary Table 8**.

3) One concern with the IFB-2 and EPS-8 ubiquitinless mutants is that it may impact the function of the molecules beyond just preventing its degradation. The authors should at least textually comment on why standard overexpression constructs were

not synthesized. Ideally, if they can perform validations that these mutations don't affect canonical function of the proteins (e.g. rescue experiments), that would be helpful.

Although ubiquitin-less mutants are more relevant to determine whether there is a direct link between loss of ubiquitination in IFB-2 and EPS-8 with increased protein levels of these factors and regulation of longevity, we agree with Reviewer #1 that standard overexpression constructs could further support our conclusions.

*We found that ubiquitin-less ifb-2 mutations increased IFB-2 protein levels in young adult worms and shortened lifespan (**Fig. 2g-h and Extended Data Fig. 7a**), indicating that upregulation of IFB-2 levels decreases lifespan. In support of this conclusion, we have observed that IFB-2 overexpression is sufficient to shorten lifespan (**Extended Data Fig. 7b**).*

*Likewise, ubiquitin-less eps-8 mutant animals exhibited dysregulated high levels of EPS-8 protein at young adult stages and a short-lived phenotype (**Fig. 2i-j**). Unfortunately, we tried to generate eps-8 overexpressing constructs for over 2 years but we were not able to synthesize them. This could be due to the fact that EPS-8 has many different isoforms and it is not clear which ones are the dominant isoforms. Moreover, the eps-8 gene itself has large introns making cloning pretty much impossible. However, we agree with Reviewer #1 that it is important to discard that the short lifespan phenotype of ubiquitin-less eps-8 mutant animals is not due to a loss of the normal function of EPS-8 protein. Since EPS-8 function is essential for embryonic survival and larval development (Croce et al, Nat. Cell Biol. 2004; Ding et al, Plos One 2008), we have now assessed whether ubiquitin-less eps-8 mutations impair these processes. Importantly, ubiquitin-less eps-8 mutant animals did not exhibit embryonic lethality and were able to develop into adult worms at the same extent than control worms (**Extended Data Fig. 7c-d**), indicating that ubiquitin-less mutations do not reduce the normal function of the protein. Moreover, we have now observed that knockdown of eps-8 after development is sufficient to rescue the short-lived phenotype of ubiquitin-less eps-8 mutant animals (**Extended Data Fig. 7e**). These data further support that the upregulation of EPS-8 protein levels underlies the short lifespan phenotype of ubiquitin-less eps-8 mutant animals.*

- 4) The authors perform visualization of actin filaments solely in the muscle, despite seeing that eps-8 knockdown in muscle and neurons are sufficient to cause a functional phenotype. While neuronal actin integrity is challenging due to the small size of neurons, this phenotype begs to interrogate actin in other tissues. Can the authors attempt to image actin in other cell types, such as the hypodermis? Inclusion of the intestine may also be a great control since they see no lifespan effect with intestine-specific knockdown. This case is made even more important considering the SDS-resistant actin aggregates found in the EPS-8 ubiquitin-less mutants. According to the muscle actin images, no aggregates are visible (just a deformed cytoskeletal network). Where are these aggregates coming from? Are there aggregates in other cell types?

To visualize actin filaments in the intestine and the epidermis, we have now used transgenic *C. elegans* strains that express the actin-binding construct LifeAct::mRuby specifically in these tissues (Higuchi-Sanabria et al Mol. Biol. Cell 2018). In contrast to the robust beneficial effects in muscle cells (Fig. 5d), knockdown of *eps-8* did not prevent or only slightly ameliorated age-associated changes in actin organization within intestinal and epidermal cells, respectively (Extended Data Fig. 10b-c). In this regard, it is important to note that similar to knockdown of *eps-8* in the intestine alone, we have now observed that tissue-specific knockdown of *eps-8* in the epidermis does not extend lifespan (Extended Data Fig. 10a).

We agree with Reviewer #1 that the most prominent age-related phenotype in the images of muscle cells is a deformed actin cytoskeletal network. However, we also find areas with higher accumulation of actin compared with the rest of the muscle actin cytoskeleton that could partially correspond to the aggregating actin detected by filter trap, whereas the actin staining is more uniform in young worms (Fig. 5d and 5h). The areas of concentrated actin observed in aged worms are rescued by knockdown of either *eps-8* and RAC orthologues (Fig. 5d and 5h). Moreover, ubiquitin-less EPS-8 accelerated the appearance of concentrated actin areas (Fig. 5i). Please see also below the aforementioned figures, where we have highlighted and showed higher magnification of the concentrated actin areas for the Reviewer (Fig. R1a-c). Thus, we cannot discard that aggregation of actin protein occurs in muscle cells.

Figure R1. Besides a deformed actin cytoskeletal network phenotype, the aging process also results in areas with higher accumulation of actin. a-c, Staining of filamentous actin with phalloidin in muscle cells. Areas with concentrated actin are highlighted and showed at higher bars, 20 μ m. Images are representative of two independent experiments. These images are presented in Fig. 5d, 5h and 5i of the manuscript without highlighting concentrated actin areas

To determine in which tissues the age-dysregulated levels of EPS-8 promote actin aggregation, we have now performed filter trap assay upon tissue-specific RNAi of eps-8. We found that knockdown of eps-8 in the muscle reduces the amounts of actin aggregates during aging (**Extended Data Fig. 10d**). Likewise, downregulation of eps-8 in neurons also resulted in decreased actin aggregates (**Extended Data Fig. 10e**). However, knockdown of eps-8 in the epidermis or intestine did not reduce age-related actin aggregation (**Extended Data Fig. 10f-g**). Taken together, these results indicate a link between upregulated eps-8 and aggregation of actin in both muscle cells and neurons during aging.

- 5) RAC components traditionally are positive regulators of actin homeostasis, yet the authors show that knockdown of RAC components can phenocopy eps-8 knockdown, a condition that promotes cytoskeletal integrity. This needs to be addressed. It would also be useful to see cytoskeletal imaging with at least one of the RAC component knockdowns.

As Reviewer #1 indicates, active RAC induces actin polymerization and remodeling. EPS-8 not only regulates the actin network through activation of RAC signaling but also by directly interacting with actin, promoting the localization of active RAC to sites of actin remodeling (Scita et al, J. Cell Biol. 2001). Actin polymerization and remodeling are essential for development, for instance by governing cell division and movement processes. According to the positive role of RAC/EPS-8 activity in actin cytoskeleton, RAC genes and EPS-8 are essential for development (Lundquist et al, Development 2001; Croce et al, Nat. Cell Biol. 2004; Ding et al, Plos One 2008). Whereas these factors endow benefits early in life, our results indicate that the age-associated increase of EPS-8 protein levels and subsequent upregulation of RAC activity have detrimental effects on actin networks and adult lifespan. Conversely, reducing the age-dysregulated increased levels of EPS-8 after development was sufficient to ameliorate actin destabilization during aging. As suggested by Reviewer #1, we have now performed knockdown experiments of the RAC orthologues mig-2 and rac-2 after development. Similar to EPS-8 knockdown, RNAi against RAC genes also prevented the age-associated destabilization of actin networks and subsequent breakdown of myosin filaments in muscle cells (please see **Fig. 5h and Extended Data Fig. 10h**). Since EPS-8 activates RAC and acts as a potent inducer of RAC-mediated actin polymerization and remodeling (Scita et al, J. Cell Biol. 2001), a potential possibility is that hyperactivation of RAC by upregulated EPS-8 results in an excessive polymerization and remodeling of actin filaments, eventually leading to the collapse of actin networks with age. We have now discussed this in the Discussion section.

Minor comments:

- 1) The presentation of the volcano plots does not seem very helpful for Figure 1. While I appreciate the authors trying to give an overview of the data, it does not really

highlight anything important or useful for the reader. Is it possible to make the graphs more meaningful or simply present the data in a different way? Perhaps highlighting some critical targets that were identified – especially the ones that were discussed in the text (e.g. similar to Fig. 3e-f).

*As indicated in our response to major comment #2, we have now replaced all the Volcano plots previously presented in Fig. 1-2 by heat maps. These heat maps show in an easier format the global changes in the Ub-proteome and how they compare with total protein levels. While **Figure 1** and related **Extended Figure 1** summarize global changes across the proteome, we have now focused **Figure 2** on the specific age-dysregulated proteasome targets identified in our proteomics analysis. As such, the different graphs presented in **Figure 2** already highlight these critical targets early in the manuscript.*

In addition, we have now assembled a Shiny Web app where the readers can generate volcano plots and interact with the ubiquitin and global proteomics data by downloading the datasets provided within the app. The Shiny Web app also allows for searching genes of interest and highlight them in the Volcano plot:

<https://vilchezlab.shinyapps.io/shiny-volcanoplot/>

- 2) Combine extended data figures into 2-3 figures to make space for more important things (see major comment 2).

We have now combined Extended Data Figures as much as possible. However, it is important to note that we have now increased the number to 10 Extended Data Figures to include the new experiments suggested by the Reviewers. We have also combined and reduced the number of main figures to conform the format instructions of Nature, transferring some of the panels to Extended Data Figures.

Referee #2 (Remarks to the Author):

This manuscript from Vilchez and colleagues describes the finding that steady state levels of ubiquitinated substrates are altered in *C. elegans* as a function of aging. Moreover, these ubiquitinated substrates can be rescued through dietary restriction or in genetic models of lifespan extension. The authors came to this observation through diGLY proteomics, a mass spectrometry method which allowed them to survey thousands of site specific ubiquitination events. Differentially ubiquitinated substrates were subsequently confirmed using linkage specific antibody pulldowns and western blot analysis of total protein levels. Functionally, the ubiquitin substrates IFB-2 and EPS-8 were both tested by genetic knockdown to demonstrate that suppressing their accumulation extended lifespan. Interestingly, the effects of each substrate manifest in a tissue dependent manner, within intestine and muscle/neuronal cells respectively. A key challenge for this paper is that the mechanism broadly decreasing protein ubiquitination remains to be resolved, and may itself be tissue dependent. These findings come on the heels of previous work from a subset of these authors involving the proteasome assembly factor Rpn6, suggesting that proteasome activity plays at

least some role. Steady state levels of ubiquitinated substrates may be influenced by many other activities and there is likely a complex interplay involving transcription, translation, ubiquitination, degradation. All of these processes are dynamic, meaning that demonstration of the proposed mechanism requires an understanding of what process becomes deficient in aging cells to tilt the equilibrium in the way described:

- It seems surprising that IFB-2 and EPS-8 would not have been previously identified as factors whose knockdown would extend lifespan, or whose protein levels increase with age. Please explain how or why these have been overlooked in previous screening work by the field.

** Regarding IFB-2 and EPS-8 as factors whose knockdown extends lifespan:*

There are four large-scale RNAi longevity screens in C. elegans (Lee et al, Nat. Genet. 2003; Hamilton et al, Genes Dev. 2005; Hansen et al, PLoS Genet. 2005; Curran et al, PLoS Genet. 2007). However, it is important to note that our results cannot be directly compared with these studies because the experimental conditions are substantially different:

a) All the previous large-scale RNAi lifespan screens used either sterile mutant worms (Hansen et al, PLoS Genet. 2005) or FUdR treatment (Lee et al, Nat. Genet. 2003; Hamilton et al, Genes Dev. 2005; Curran et al, PLoS Genet. 2007), which are common methods to inhibit progeny and facilitate longevity screens. However, sterility and FUdR treatment can affect distinct biological processes (Feldman et al, PLoS One 2014; Scott et al, Cell 2017; Lee et al, Nat. Metabolism 2019). Since our targeted RNAi screen against age-dysregulated proteasome targets was focused on 10 genes, we performed the lifespan experiments in fertile wild-type worms without FUdR treatment to avoid potential effects of sterility and FUdR treatment.

b) Since we defined changes in ubiquitination and protein levels during the aging of adult worms, we started the RNAi treatment for lifespan assays after development (i.e. day 1 of adulthood). This is the main difference with previous large-scale RNAi longevity screens, as they started the RNAi treatment at embryonic (Hansen et al, PLoS Genet. 2005) or early larval stages (Lee et al, Nat. Genet. 2003; Hamilton et al, Genes Dev. 2005). As such, these large-scale RNAi screens overlook regulators of adult lifespan if these genes are essential for embryo and/or larval development. A fourth screen started RNAi treatment during later larval stages, but they used a different genetic background (eri-1 mutant worms) and FUdR treatment (Curran et al, PLoS Genet. 2007) making difficult to compare it with our lifespan experiments. In summary, essential genes for development are underrepresented in genome-wide RNAi screens for post-developmental phenotypes such as aging whereas our study circumvents this limitation. This is particularly important for proteins such as EPS-8 and IFB-2, which have a role in normal development. For instance, loss of eps-8 during development triggers embryonic lethality and defects in the morphogenesis of distinct tissues in C. elegans (Croce et al, Nat. Cell Biol. 2004; Ding

et al, Plos One 2008). Likewise, IFB-2 is necessary for intestinal morphogenesis during *C. elegans* development (Geisler et al, Sci. Rep. 2020, Karabinos et al, Eur. J. Cell Biol. 2004). We have now made more clear in the main text that several age- dysregulated proteasome targets are essential for development and we started our targeted RNAi screen at day 1 of adulthood. For example, the text now says: “Several age-dysregulated proteasome targets have an essential role during development. For instance, loss of EPS-8^{29,30}, IFB-2³¹, RPL-4³² or F54D1.6³³ before reaching adulthood leads to embryogenic and developmental defects. Whereas these factors endow benefits early in life, we asked whether the age-associated increase in their levels have detrimental effects for adult lifespan. We hypothesized that if age-dysregulated proteasome targets are not essential for adult viability, genetic interventions that specifically diminish their levels only during adulthood can prolong longevity. To this end, we performed single RNAi treatment against the age-dysregulated proteasome targets after development into adult worms”. We have also specified in the Methods section and Figure Legends that we started RNAi treatment after development.

* Regarding IFB-2 and EPS-8 as factors whose protein levels increase with age:

Our proteomics analysis of global protein levels indicates that both IFB-2 and EPS-8 amounts increase with age, as we validated by western blot experiments. A previous study from Prof. Kenyon’s laboratory carried out one of the most comprehensive proteomics analysis of protein levels in *C. elegans* (Narayan et al, Cell Systems 2016). In this work, the authors performed a SILAC-based deep analysis of protein levels comparing young (day 1) and older (day 10) adult worms. Although the authors mostly focused on enriched gene ontology (GO) term analysis to identify age-related processes, other proteins that did not belong to enriched GO terms were also changed with age. Table S3 of the aforementioned paper contains log₂ Fold Changes and statistical analysis of all the quantified proteins including IFB-2 (Protein Accessions: Q19286;M1Z854) and EPS-8 (G5ED33;O18250;Q7YTG1;Q7YTG2;G5EFE0). When compared with young worms, old worms (day 10) exhibit increased protein levels of both IFB-2 (log₂ Fold Change: 0.72, FDR-corrected *p* value < 0.0001) and EPS-8 (log₂ Fold Change: 0.51, FDR-corrected *p* value < 0.001). Therefore, this proteomics study supports that IFB-2 and EPS-8 protein levels increased with age, as we observed in our proteomics and western blot experiments.

- Given the complex temporal dynamics observed in Fig 1e-g, a more granular time course should be performed to understand when declines occur in WT worms and how Ub levels change as a function of time, particularly in the *daf-2* model.

We have now assessed the levels of ubiquitinated proteins at day 1, 2, 3, 4, 5, 6, 7, 8, 9, 10, 12, and 15 in wild-type, *eat-2(ad1116)* and *daf-2(e1370)* worms (please see **Extended Data Fig. 2a-c**). The text now says: “A more granular time-course analysis indicated that the robust decline in Ub-protein levels of wild-type worms

occurs after day 8 of adulthood, whereas the increase in Ub-proteins of daf-2 mutant animals starts from day 2 (Extended Data Fig. 2a-c)”.

- As written, the text strongly emphasizes the ubiquitinated proteome, in spite of having performed global proteome analysis in parallel with some or all of the diGLY studies. From the figures and text, it was difficult to ascertain how protein and diGLY levels compare for individual proteins and how different lysine modification sites respond within affected proteins. While reflected in tabular format, this information is more important than many of the summary panels presented in figures 1-2.

We have now re-written the text and included new figures to emphasize the comparison between diGLY and total protein levels for individual proteins:

- *To ascertain how ubiquitination and protein levels compare for individual proteins across the proteome, we have now included heat maps that integrate ubiquitin and global proteomics data comparing different ages of the distinct strains with day 1 worms of the respective genetic background (Extended Data Fig. 1b). These heat maps provide a direct visual comparison of ubiquitination vs total protein levels, showing that differences in the levels of Ub-peptides often do not correlate with a similar change in the total amounts of the respective protein.*
- *We have now replaced the volcano plot presented in the Fig. 2a of our first submission showing widespread changes in the levels of Ub-peptides comparing aged (day 15) and young (day 5) wild-type worms by a heat map for better visualization of these global changes (Fig. 1e). In this graph, we have also integrated data of total protein levels for all the differentially abundant Ub-peptides in aged worms, allowing direct visual comparison of ubiquitination vs. total protein in aged worms. Besides showing that aging is particularly associated with a decline of Ub-peptides in wild-type worms, this heat map summarizes in an visible manner that not all the differences in Ub-peptides can be simply ascribed to a similar change in the total protein levels.*
- *We have now added Tables with examples of distinct proteins to show how different lysine modification sites respond within these proteins (please see Extended Data Fig. 1c)*
- *In the main text, we have now expanded on the comparison of ubiquitination vs. total protein levels. The text now says: “Given that *C. elegans* undergo a widespread proteome remodeling during aging^{14,15}, we assessed how age-related changes in Ub-peptides compare with the total levels of the corresponding protein. To this end, we quantified the global amounts of individual proteins in aging wild-type and long-lived mutant worms (Fig. 1a and Supplementary Table 2). When we integrated these data with diGLY proteomics, we found that differences in the levels of Ub-peptides often do not correlate with a similar change in the total amounts of the respective protein (Extended Data Fig. 1b and Supplementary Table 3). Since we detected a*

higher number of changes in the levels of Ub-peptides after day 5 of adulthood (**Fig. 1c**), we directly compared young (day 5) with aged (day 15) wild-type worms for further analysis of age-related differences. Indeed, old worms underwent a deep remodeling of their Ub-modified proteome when compared with day 5-adults (**Fig. 1e and Supplementary Table 4**). In aged wild-type worms, the steady-state levels of 1813 Ub-peptides were downregulated whereas 350 Ub-peptides were upregulated (**Fig. 1e and Supplementary Table 4**), further supporting that aging is particularly associated with a decline of ubiquitinated peptides. Among these age-related differences, only 582 downregulated and 123 upregulated Ub-modified peptides correlated with a change in the total levels of the protein in the same direction (**Fig. 1e and Supplementary Table 5**). However, 905 downregulated and 189 upregulated Ub-peptides corresponded to proteins that did not change in abundance with age (**Fig. 1e and Supplementary Table 5**). For instance, the protein levels of myosin MYO-2 and paramyosin UNC-15 remained similar during aging, but they contained multiple downregulated and upregulated Ub-sites (**Extended Data Fig. 1c**). The rest of differentially abundant Ub-peptides were inversely correlated with the corresponding protein levels (**Fig. 1e and Supplementary Table 5**). For example, EPS-8 protein became more abundant with age while most of its Ub-sites were significantly downregulated (**Extended Data Fig. 1c**). Thus, our integrative analysis demonstrated that not all the differences in Ub-peptides can be simply ascribed to a similar change in the total protein levels”.

- In terms of system dynamics, what does mRNA expression look like for each of the different ubiquitin genes? Are either the ribosomal fusion or stress response ubiquitin genes differentially regulated with age in the WT or mutant strains?

We have now assessed the mRNA levels for the ubiquitin-encoding gene *ubq-1*, the ubiquitin-ribosomal fusion genes *ubq-2* and *ubl-1* as well as the ubiquitin-stress response gene *usp-14* in wild-type and long-lived mutant strains at day 1 and day 15 of adulthood (please see **Fig. 1j and Extended Data Fig. 3a-c**). The text now says: “With the global decline of Ub-proteins in aged wild-type worms, we asked whether this process directly correlates with changes in the expression of ubiquitin itself. Ubiquitin is encoded by three different genes in *C. elegans*, i.e. *ubq-1* and the ubiquitin-ribosomal fusion genes *ubq-2* and *ubl-1*. In contrast to the global downregulation of Ub-peptides, aged wild-type worms had higher transcript levels of both *ubq-1* and *ubq-2* when compared with young worms, whereas *ubl-1* mRNA levels remained similar (**Extended Data Fig. 3a**). We also observed a significant upregulation of *ubq-1* transcript levels in long-lived *daf-2* mutant animals with age (**Extended Data Fig. 3b-c**). Nevertheless, aged wild-type worms expressed higher or similar levels of *ubq-1*, *ubq-2* and *ubl-1* when compared with age-matched *eat-2* and *daf-2* mutant animals (**Fig. 1j**). Moreover, aging and longevity pathways did not change the expression of *usp-14* (**Fig. 1j and Extended Data Fig. 3a-c**), a ubiquitin-stress response gene induced by ubiquitin deficiency²¹”.

- In terms of system dynamics, what is the $t_{1/2}$ of ubiquitin itself in worms of different ages and genetic backgrounds tested here? This is particularly important for the claims made in lines 368-374.

We have now treated the worms with cycloheximide to block ubiquitin synthesis (Hanna et al, Mol. Cell Biol. 2003) and assessed $t_{1/2}$ of protein ubiquitin itself in wild-type, eat-2 and daf-2 animals at day 1 and 10 of adulthood (Extended Data Fig. 3e-f). We found that the half-life of ubiquitin is approximately 2 hours in young wild-type worms and approximately 30 minutes longer in aged wild-type worms (Extended Data Fig. 3e-f). Likewise, long-lived eat-2 and daf-2 mutants exhibited similar half-lives for protein ubiquitin when compared with age-matched wild-type worms (Extended Data Fig. 3e-f). Taken together, these data indicate that the downregulated amounts of ubiquitinated proteins characteristic of aged wild-type worms is not associated with a decrease in the half-life of ubiquitin itself.

- Can the authors differentiate between the possibilities that the decrease in conjugated ubiquitin observed in aged WT worms results from elevated DUB activity, rather than a deficiency in the ubiquitin conjugation system?

*We really appreciate this comment from Reviewer #2 because it led to exciting findings supporting a role of elevated DUB activity in the age-associated decline of ubiquitinated proteins. Using our proteomics data, we have now identified differentially expressed E3 ubiquitin ligases and DUBs with age. In *C. elegans*, there are over 170 genes that encode for E3 ligases (Kipreos et al WormBook 2005). However, we only found significant changes in the expression of 12 E3 enzymes with age (Extended Data Fig. 4a and Supplementary Table 7). On the other hand, there are 45 DUBs in *C. elegans* (Papaevgeniou Redox. Biol. 2014), but a higher proportion (i.e. 14 DUBs) were significantly upregulated in aged wild-type worms (Extended Data Fig. 4b and Supplementary Table 7). Among them, dietary restriction rescued changes in CSN-6 levels whereas reduced IIS was sufficient to prevent the upregulation of most of the age-dysregulated DUBs including CSN-6, USP-48, USP-50, USP-5 and H34C03.2 (Extended Data Fig. 4b).*

Prompted by these results, we assessed whether elevated DUB activity could contribute to the age-associated decline in Ub-protein levels of wild-type animals. To this end, we treated the worms with PR-619, a broad-spectrum DUB inhibitor (Altun et al, Chem. Biol. 2011, https://www.merckmillipore.com/DE/en/product/DUB-Inhibitor-V-PR-619-Calbiochem,EMD_BIO-662141). Notably, the treatment with DUB inhibitor for 4 h rescued ubiquitination levels in old worms (Fig. 1k). We found that single knockdown of csn-6/COPS6, H34C03.2/USP4, F07A11.4/USP19, math-33/USP7, usp-5/USP5, usp-48/USP48 or otub-3/OTUD6A ameliorates the decline of Ub-protein levels in aged wild-type worms, whereas downregulation of other DUBs did not prevent this phenotype (Extended Data Fig. 4c-g). Altogether, these results suggest that elevated DUB activity could tilt the equilibrium towards the global decrease in ubiquitination levels characteristic of aged worms.

- The statement on lines 125-127 does not accurately reflect what the western blots show.

We have now rewritten these lines. The text now says: “Among the numerous ubiquitination changes in wild-type worms during aging, the proteomics data indicated that the equilibrium is tilted towards a decline in ubiquitination levels. To further assess these changes, we performed western blot analysis at the same ages investigated in our proteomics assay. Indeed, day 10 and day 15 wild-type worms exhibited a dramatic decrease in the global levels of Ub-proteins compared with day 5 and day 1 wild-type worms (Fig. 1g). On the other hand, day 10 and 15 eat-2 mutant worms did not have a strong downregulation in Ub-protein levels compared with younger ages (Fig. 1h). In daf-2 mutant worms, the levels of Ub-proteins were upregulated compared with day1 adults of the same genetic background, a phenotype that was already acute at day 5 of adulthood (Fig. 1i). A more granular time-course analysis indicated that the robust decline in Ub-protein levels of wild-type worms occurs after day 8 of adulthood, whereas the increase in Ub-proteins of daf-2 mutant animals starts from day 2 (Extended Data Fig. 2a-c)”.

- The assertions in lines 144-145 seem to be assumptions, since the proteome and diGLY proteome of muscles and neurons were not directly measured.

*Reviewer #2 is absolutely right and this concern was also raised by Reviewer #1. As indicated in our response to Reviewer #1, we have now made clear in the text that **Extended Data Fig. 6a** presents a bioinformatic analysis combining our proteomics data with previously published datasets of tissue-specific expression and toned down the conclusions from this analysis. For instance, we have removed the statement regarding that the muscle and intestine have slightly more changes than neurons or germline. We agree that this conclusion requires ubiquitin proteomics experiments in a tissue-specific manner, which are not feasible due to the lack of protocols to isolate most of the tissues (e.g., muscle, epidermis) or obtain sufficient protein amounts of other tissues such as the germline and intestine for diGly enrichment.*

*Nevertheless, we think that it is important to keep this bioinformatic analysis as a first indication that the decline in the steady-state levels of ubiquitination is not tissue-specific and occur through the entire organism. In these lines, we have now included the epidermis in the bioinformatic analysis as requested by Reviewer #3 (please see **Extended Data Fig. 6a and Supplementary Table 10**). To confirm that the decline in ubiquitination occurs across tissues, we have now assessed global levels of ubiquitinated proteins in distinct isolated tissues and body parts of *C. elegans* (**Extended Data Fig. 6b**). The text now says: “Using previously published datasets of tissue-specific expression³⁷, we classified proteins that exhibit differences in their Ub-peptides according to the tissues where these proteins are expressed. The bioinformatic analysis indicated that distinct tissues such as the germline, muscle, intestine, epidermis or neurons express multiple proteins that contain downregulated*

Ub-peptides with age (Extended Data Fig. 6a and Supplementary Table 10). Similar to whole organism lysates, we observed a decrease in the global amounts of Ub- proteins in isolated germlines, intestines and heads during aging (Extended Data Fig. 6b), supporting that this process occurs through the organism”.

- Do the authors believe the underlying mechanism proposed to alter steady state protein ubiquitination levels for proteins such as IFB-2 and EPS-8 is conserved across different cell types? Or is it possible that this is tissue specific? In a similar vein, are these effects observed in a cell autonomous manner or only when cells are living in the context of an intact organism?

*As mentioned above, we have now extruded different tissues and body parts of *C. elegans* to assess whether the age-associated decline in the steady-state levels of ubiquitination is conserved across tissues. Similar to lysates from the entire animal, there was a pronounced decrease in the global amounts of Ub-proteins in isolated germlines, intestines and heads (Extended Data Fig. 6b). These data support that age-associated downregulation in global ubiquitination levels occurs in different cell types. Moreover, we have now examined age-related changes in the EPS-8 and IFB-2 protein levels of isolated tissues. EPS-8 is ubiquitously expressed in the soma (Dinget al, Plos One 2008; Stetak et al, EMBO J. 2006) and its protein amounts were increased in different somatic tissues with age (Extended Data Fig. 6c), correlating with the decline in ubiquitination levels across tissues. Since the *ifb-2* gene is specifically expressed in intestinal cells (Bossinger et al, Dev. Biol. 2004), its protein levels were upregulated in isolated intestines of old worms (Extended Data Fig. 6d). However, we did not detect IFB-2 in a distinct somatic tissue even during the aging process (Extended Data Fig. 6d).*

*To further assess the impact of the ubiquitin-proteasome system in the intracellular regulation of IFB-2 and EPS-8 protein levels, we have now performed tissue-specific knockdown of *rpn-6*. Loss of *rpn-6* in the intestine, epidermis, neurons or muscle increased EPS-8 levels in young worms, indicating that the ubiquitin- proteasome system modulates EPS-8 in all these tissues (Extended Data Fig. 6e-h). However, only intestinal loss of *rpn-6* was sufficient to upregulate IFB-2 levels in young worms (Extended Data Fig. 6e-h), according to the specific expression of IFB-2 in the intestine. Taken together, these experiments support that the ubiquitin-proteasome system acts in a cell autonomous manner to regulate IFB-2 and EPS-8 levels.*

*Since we performed our proteomics experiments in the context of an intact organism, it is possible that cell non-autonomous mechanisms also regulate steady- state levels of ubiquitination in distal tissues during aging. Indeed, growing evidence demonstrates that interorgan communication is also an important determinant of organismal aging (Taylor et al, Nat. Rev. Mol. Cell Biol. 2014). For instance, the nervous system elicits signals that modulate aging of distal tissues (Taylor et al, Nat. Rev. Mol. Cell Biol. 2014). To assess whether cell non-autonomous events influence the age-associated decline in ubiquitination levels, we have now examined *unc-13* mutant worms which are deficient in the release of neurotransmitters from small clear*

vesicles (Richmond et al, Nature Neuroscience 1999). Importantly, blocking neurotransmitter release did not affect the steady-state levels of ubiquitination in young worms, but it exacerbated the age-associated decline in older worms (**Extended Data Fig. 6i**). Concomitantly, the protein levels of ubiquitously expressed EPS-8 and intestinal-specific IFB-2 were further upregulated in aged *unc-13* mutants when compared with age-matched wild-type worms (**Extended Data Fig. 6j-k**), suggesting a role of cell non-autonomous mechanisms in this process.

- The assertion on lines 172-175 is overly simplistic and inappropriate given the endpoint experiment carried out. The ubiquitin system functions on the order of minutes to degrade individual substrates and is deeply intertwined with transcription, translation, protein localization and turnover processes. This paper would be improved if it approached modelling of this equilibrium in a more sophisticated manner.

Reviewer #2 is absolutely right. We have now re-written the text and added new experimental data, performing a more sophisticated approach for the modelling of changes in the steady-state of ubiquitination and protein levels:

- *We have now removed the specific sentences indicated by Reviewer #2: “Thus, differences in the amounts of these Ub-modified peptides could ensue from age-related changes in the transcriptional or translational regulation of protein levels. In addition, 905 downregulated and 189 upregulated Ub-modified peptides did not result in alterations of protein abundance with age (Fig. 3a and Supplementary Data 7), suggesting that these ubiquitination events are not a mark for protein degradation”. Instead, we have now focused on expanding the main conclusion from the comparison between ubiquitination and total protein levels, i.e. our analysis demonstrates that not all the differences in Ub-modified peptides can be simply ascribed to a similar change in the total protein levels.*
- *Besides protein degradation, we have now indicated in the main text that the steady-state levels of ubiquitinated substrates can be influenced by many other activities, including transcription, translation, protein localization and ubiquitination/deubiquitination. For instance, the text now says: “The steady-state levels of Ub-peptides can be influenced by processes such as transcription, translation, protein localization and proteolysis. All these processes are dysregulated with age and rescued by longevity paradigms^{5,10,16-20}. To assess whether DR and reduced IIS prevent dysregulation of the Ub-modified proteome, we directly compared aged wild-type worms with age-matched *eat-2* and *daf-2* mutants. Among the 1813 downregulated ubiquitin modifications identified in old wild-type worms, age-matched *eat-2* and *daf-2* mutants exhibited increased ubiquitination for 952 and 336 peptides, respectively (**Fig. 1f, Extended Data Fig. 1d and Supplementary Table 6**). Among the 350 upregulated Ub-peptides in aged wild-type worms, *eat-2* and *daf-2* animals had decreased ubiquitination for 234 and 251 peptides, respectively (**Fig. 1f, Extended Data Fig. 1d and Supplementary Table 6**). Thus, pro-longevity signaling pathways rescued age-*

related changes in the Ub-proteome, further supporting that alterations in regulated mechanisms contribute to this process”.

- We have now performed a series of experiments that indicate that elevated DUB activity is an important factor to tilt the equilibrium towards the decrease in ubiquitination levels during aging. For instance, we have now treated the worms with a broad-spectrum DUB inhibitor (PR-619) and found that this treatment rescues the age-associated decline in ubiquitination levels (**Fig. 1k**). Using our proteomics data of global protein levels, we observed that 14 DUBs were significantly upregulated in aged wild-type worms (**Extended Data Fig. 4b**). Among them, single knockdown of *csn-6/COPS6*, *H34C03.2/USP4*, *F07A11.4/USP19*, *math-33/USP7*, *usp-5/USP5*, *usp-48/USP48* or *otub-3/OTUD6A* ameliorates the age-associated decline of Ub-protein levels in wild-type worms (**Extended Data Fig. 4c-g**).
- Among the numerous age-related changes in the ubiquitinated proteome, our aim was to define proteasome targets that become dysregulated with age. Therefore, we have kept the focus on the ubiquitin-proteasome system. By integrating ubiquitin and total protein proteomics of both aging and proteasome-less worms with immunoprecipitation experiments of Lys48-linked polyUb chains, we identified age-dysregulated proteasome targets such as *IFB-2* and *EPS-8*. As we have now discussed in the text, we cannot discard that other activities could also influence the levels of these proteins. The Discussion section says: “Although our findings support at least a partial role of the ubiquitin-proteasome system in regulating the amounts of specific longevity modulators, it is important to note that changes in other processes such as translation or protein localization could also contribute to regulating the ubiquitination, protein levels and activity of these factors”.
- Nevertheless, we believe that the upregulation of proteins such as *IFB-2* and *EPS-8* upon *rpn-6* RNAi supports at least a partial a role of the proteasome in regulating their levels. We further strengthened this conclusion with supporting data from other experiments: 1) the mRNA levels of *EPS-8* and *IFB-2* did not increase upon aging or *rpn-6* RNAi, 2) blocking ubiquitination of *EPS-8* and *IFB-2* by gene editing resulted in increased levels of these proteins and 3) knockdown of *rpn-6* in the intestine, epidermis, muscle and neurons increased the levels of the ubiquitously expressed *EPS-8* whereas only knockdown of *rpn-6* in the intestine upregulated the amounts of the intestinal protein *IFB-2*.
- We have now explained in more detail our criteria to identify potential proteasome targets, while indicating that we cannot discard that other ubiquitinated proteins could also be modulated by the proteasome. The text now says: “Given the high number of downregulated Ub-peptides during aging, we hypothesized that a subset of these events could reduce selective degradation of specific proteins by the proteasome. Our integrated analysis revealed age-related changes in multiple Ub-peptides that directly correlated with similar differences in the protein levels or corresponded to proteins with unchanged abundance, making difficult to interpret whether these Ub-sites modulate proteasomal degradation of the protein (**Fig. 2a**

and Supplementary Table 5). To identify downregulated ubiquitin marks involved in protein degradation, we focused on changes in Ub-peptide levels that inversely correlated with protein amounts. Notably, we found that 192 proteins exhibited less ubiquitination in at least one of their lysine sites during aging, while the total levels of the protein increased (**Fig. 2a and Supplementary Table 5**). Thus, these events could indicate a loss of ubiquitin marks in target proteins, resulting in diminished degradation of the protein with age. If Ub-proteins are indeed proteasomal targets, defects in proteasome activity could reduce their degradation in young animals, leading to increased amounts of both the protein and the corresponding Ub-peptides”.

- It would help if the text explained in more detail how Rpn6 knockdown alters the baseline proteome. For example, what is the proposed impact of USP5 expression changes on phenomena being studied here?

We have now added a heat map representing differentially abundant proteins upon *rpn-6* RNAi treatment (please see **Extended Data Fig. 5a**) and explained in more detail how *rpn-6* knockdown alters the baseline proteome. As suggested by Reviewer#2, we have also discussed about the potential impact of *usp-5* expression changes upon *rpn-6* knockdown. The text now says: “To decrease proteasome function in young adult worms, we knocked down *rpn-6*, a specific activator of 26S proteasome assembly and activity^{10,27}. Given the role of the proteasome in multiple biological pathways, *rpn-6* RNAi resulted in widespread changes in the proteome baseline of young adults (day5), including 297 downregulated proteins and 509 upregulated proteins (**Extended Data Fig. 5a and Supplementary Table 8**). Besides potential indirect effects on protein levels caused by proteasome dysfunction, we hypothesized that these upregulated proteins also include direct proteasome targets which are less degraded upon *rpn-6* RNAi, particularly if they also have increased Ub-peptides”. (...) “Likewise, loss of *rpn-6* did not upregulate the transcript levels of these targets, with the only exception of the deubiquitinase *usp-5* (**Extended Data Fig. 5c**). Given the role of USP-5 in protein ubiquitin homeostasis²⁸, a process challenged by deficits in proteasome activity, the increased expression of *usp-5* gene could indicate a compensatory mechanism to protect from the deleterious effects induced by *rpn-6* RNAi”.

- The concluding statement in lines 208-209 oversimplifies a complex and dynamic process.

We apologize for this oversimplification. We have now replaced the concluding statement “Concomitantly, these proteins cannot be recognized and degraded by the proteasome, resulting in upregulated levels of the protein in aged animals” by “Altogether, our data indicate that the profound rewiring of the Ub-proteome during aging could also decrease the steady-state ubiquitination levels of distinct proteasome targets. In turn, lower ubiquitination marks could reduce the proportion of these targets

for recognition and subsequent degradation by the proteasome, eventually contributing to the upregulation in their total protein amounts with age”.

- The text indicates that the data will be made available in PRIDE, although this should have been done prior to initial submission and data access provided to reviewers through in a PW protected manner as is now standard. More concerning is the comment that upon publication, additional data will be available upon request. For a paper such as this, this approach is no longer sufficient. The authors should assemble an interactive tool (e.g. Shiny Web app) so that readers of the paper can interact with the global proteome and diGLY proteome data and draw their own conclusions.

We apologize for not providing access to the raw proteomics data in our initial submission. We have now deposited all the proteomics data in PRIDE with access for the reviewers using the following reviewer account details:

- **Ubiquitin proteomics of aging and long-lived worms**

Username: reviewer_pxd024338@ebi.ac.uk

Password: XAeITYdr

- **Global protein proteomics of aging and long-lived worms**

Username: reviewer_pxd025128@ebi.ac.uk

Password: Elpbsdsf

- **Ubiquitin proteomics upon *rpn-6* RNAi**

Username: reviewer_pxd024094@ebi.ac.uk

Password: Gnko9wXQ

- **Global protein proteomics upon *rpn-6* RNAi**

Username: reviewer_pxd024095@ebi.ac.uk

Password: dEJixzn4

- **Immunoprecipitation Lys48-linked polyUb**

Username: reviewer_pxd024093@ebi.ac.uk

Password: vRXnU9lw

- **Immunoprecipitation Lys63-linked polyUb**

Username: reviewer_pxd024045@ebi.ac.uk

Password: oGcqr2YH

*We have now included source data for all the gels (please see **Supplementary Fig. 1**) and graphs (**Source Data file**) of the Main Figures and Extended Data Figures. Therefore, all data are now present in the Main Figures, Extended Data and*

Supplementary Information. We have now assembled two Shiny Web apps where the readers can interact with the ubiquitin and global proteomics data by downloading the datasets provided within the apps:

<https://vilchezlab.shinyapps.io/shiny-volcanoplot/>
<https://vilchezlab.shinyapps.io/shiny-heatmap/>

The Data Statement now says: “ There is no restriction on data availability. Source data are provided with this paper. Readers can interact with the ubiquitin and global proteomics data using the following Shiny Web apps by downloading the datasets provided in the apps: <https://vilchezlab.shinyapps.io/shiny-volcanoplot/> and <https://vilchezlab.shinyapps.io/shiny-heatmap/>. All the proteomics raw data will be made publicly available in the ProteomeXchange Consortium via the PRIDE partner repository under accession codes PXD024338 (ubiquitin proteomics of aging and long-lived worms), PXD025128 (global protein proteomics of aging and long-lived worms), PXD024094 (ubiquitin proteomics upon rpn-6 RNAi), PXD024095 (global protein proteomics upon rpn-6 RNAi), PXD024093 (immunoprecipitation Lys48-linked polyUb), PXD024045 (immunoprecipitation Lys63-linked polyUb)”.

- Extending from the concern above, the Supplemental Tables have substantive deficits that make it impossible for an expert in mass spectrometry to judge the quality of these data. For example:
 - o Supplemental Tables 1 & 3 do not provide the peptide sequences that were observed in the diGLY data. This is arguably the most critical piece of information to have in this table for purposes of assessing data integrity.

*We apologize for not providing peptide sequences for the diGLY proteomics data in our first submission. We have now included the peptide sequences in the corresponding Supplementary Tables for all the ubiquitin-proteomics experiments: **Supplementary Tables 1, 4 and 8.***

- o Supplemental Table 2 does not provide any indication of the number of peptides used in assembling the quantification data per protein.

*We are very sorry for not including this relevant information in our first submission. We have now included the total number of identified peptides per protein as well as the number of peptides used for quantification of individual protein levels in **Supplementary Tables 2, 8 and 9.***

Referee #3 (Remarks to the Author):

Review Koyuncu et al Nature

The authors report the results of a systematic proteomic analysis in which they

quantified changes in the ubiquitin-modified signatures during *C. elegans* ageing. They find that ubiquitination gets downregulated or upregulated for hundreds of proteins between day-5 and day-15 of adulthood. They focus on 10 specific proteins that become under-ubiquitinated during ageing, or hyper-ubiquitinated when the proteasome is inhibited, and show that knocking down the corresponding genes extends lifespan. For two of them, the intermediate filament IFB-2 enriched in the intestine and the actin-binding protein EPS-8, they further identify the lysine residues mediating ubiquitination and show that CRISPR/Cas9 changes of these lysines into arginines reduces lifespan. Last but not least, they examine why changes in the ubiquitination pattern of both proteins alters lifespan. They find that overexpression of IFB-2 cause the aggregation of other intestinal intermediate filaments and reduce animal resistance to intestinal colonization by *E. coli*. Likewise, they find that EPS-8 knockdown upregulates Rac and JNK/KGB-1 signaling in neurons and muscles, and reduces the disorganization of muscle myofilaments.

Hence the authors went from a proteome-wide study of protein ubiquitination during ageing to the cellular and molecular characterization of two proteins, showing why their ubiquitination matters during normal or genetically modified ageing. This is a remarkable achievement, which will be of interest to a very wide audience as a central quest in the ageing field is to define how to prolong healthy ageing and to identify pathways that could mediate so. Furthermore, experiments are overall very well done and the manuscript is well written. I only have rather minor comments. Of note, *C. elegans* intermediate filaments in the intestine, epidermis and excretory cell do not resemble any classical vertebrate intermediate filaments; in fact, they are related to lamins, such that the significance of their findings on IFB-2 will await confirmation of a related study in mouse (for instance) with keratins, vimentin, desmin, nestin, GFAP and the like. On the other hand, the results obtained with EPS-8 should be transposable in vertebrates.

Comments:

1. The authors underline that muscles and intestine exhibited a higher number of proteins with altered ubiquitination, with neurons and germline cells coming next. I find it surprising that the skin would not be represented in their list (if anything given the billions involved in the cosmetic industry to make our skin looking forever young!). For instance, EPS-8 plays also an important role in the epidermis and remains expressed in adults.

*Reviewer #3 is absolutely right and we apologize for not including the epidermis in our bioinformatic analysis of proteins with altered ubiquitination levels in our first submission. We have now included the epidermis in the bioinformatic analysis (please see **Extended Data Fig. 6d**). Similar to other tissues, the epidermis also expressed a high number of proteins that contain downregulated Ub-peptides with age (**Extended Data Fig. 6a**).*

As Reviewer #3 indicates, *EPS-8* plays an important role in the epidermis during development and remains expressed in this tissue during adulthood. We have now performed tissue-specific knockdown of *rpn-6* after development to further assess the impact of the ubiquitin-proteasome system in the intracellular regulation of *EPS-8* levels in different tissues, including the epidermis. Notably, loss of *rpn-6* in the intestine, neurons, muscle or epidermis increased *EPS-8* levels in young worms, indicating that the ubiquitin-proteasome system modulates *EPS-8* in all these tissues during adulthood (**Extended Data Fig. 6e-h**).

Although the ubiquitin-proteasome system modulates *EPS-8* levels in the epidermis, we found that epidermal-specific knockdown of *eps-8* does not extend longevity (**Extended Data Fig. 10a**). The text now says: “In contrast to *ifb-2*, tissue-specific knockdown of *eps-8* in the intestine alone did not affect lifespan (**Fig. 4a**). Likewise, tissue-specific knockdown of *eps-8* in the epidermis did not modulate lifespan (**Extended Data Fig. 10a**). However, tissue-specific knockdown in either the muscle or neurons extended adult lifespan (**Fig. 4b-c**), indicating that *EPS-8* activity in these cells regulates organismal longevity”.

In addition, we have now assessed whether knockdown of *eps-8* ameliorates age-associated alterations in actin organization within epidermal cells (**Extended Data Fig. 10b**). The text now says: “In contrast to the robust beneficial effects in muscle cells, knockdown of *eps-8* did not prevent or only slightly ameliorated age-associated changes in actin organization within intestinal and epidermal cells, respectively (**Extended Data Fig. 10b-c**)”. Moreover, we have now observed by filter trap assay that knockdown of *eps-8* in the epidermis alone does not reduce age-related actin aggregation (**Extended Data Fig. 10f**).

2. A question concerning the method used to synchronize animals for proteomic studies. The authors used FUDR to prevent the development of the progeny, which is indeed a classical method in the field. I have a potential concern with this approach, inasmuch FUDR inhibits mitochondria and makes animals sterile, both of which have impact on ageing. The issue is whether the treatment could affect the proteins identified in their screens. The ten proteins on which they focus are ok, but what about the hundred others? It would be important to have an assessment of it by probing a small subset of their dataset taking proteins for which antibodies are available.

This is a very important point raised by Reviewer #3. Given the large amounts of total protein required for the proteomic experiments, the only way to obtain sufficient populations of synchronized worms was to inhibit progeny. As Reviewer #3 indicates, FUDR is extensively used in the aging field to inhibit progeny. Since FUDR does not affect longevity at the standard conditions used in our manuscript (e.g., Mitchell et al, J. Gerontol. 1979; Hosono et al, Exp. Gerontol. 1982; Gandhi et al, Mech. Ageing Dev. 1980; Lee et al, Nat. Metabolism 2019), we used this treatment for the proteomics experiments. However, Reviewer #3 is absolutely right and FUDR can make animals sterile and affect other processes. Beyond the ten main proteins of our study, we agree that it is important to probing another small subset of the proteomics datasets to

examine whether the treatment could affect the identified proteins. Following the Reviewer's suggestion, we have now assessed the levels of three additional proteins by western blot in worms without FUDR treatment. Our proteomics dataset of wild-type worms treated with FUDR indicates that the protein levels of AJM-1 and DLG-1 strongly increase with age, while ERM-1 levels do not significantly increase (please see **Figure R2a** below). Importantly, we obtained similar results by western blot in wild-type worms without FUDR treatment (**Figure R2b**).

Figure R2. Assessment of a subset of proteins by western blot in worms without FUDR treatment. **a**, Proteomics analysis of wild-type worms treated with FUDR indicates that the protein levels of AJM-1 and DLG-1 increase with age, while ERM-1 levels do not significantly increase ($n=4$, False Discovery Rate (FDR) <0.05 was considered significant). **b**, Western blot analysis with antibody to AJM-1, DLG-1 and ERM-1 of wild-type worms without FUDR treatment. The images are representative of two independent experiments. All the antibodies were obtained from Developmental Studies Hybridoma Bank (DSHB).

- The procedure used for immunoblots involves a centrifugation of worm extracts at 8000g for 5 min to collect the supernatant. The antibody MH33, which they used to probe the abundance of IFB-2, was generated against a highly insoluble worm fraction (Francis and Waterston, JCB 1991). The results shown in Fig. 4b,d are compelling but I was wondering whether the bulk of IFB-2 changes and whether they could correspond to squiggles.

As Reviewer #3 indicates, MH33 antibody was generated against a highly insoluble worm fraction and we completely agree that further controls are needed to strengthen our conclusions. We have now performed western blot with antibody against IFB-2 (MH33) of whole *C. elegans* extracts without the 8000xg centrifugation step. Similar to supernatant samples (**Fig. 2e**), we observed an increase in the total levels of IFB-2 in aged worms when the whole *C. elegans* extract was analyzed (please see **Extended Data Fig. 5h**). In addition, analysis of whole worm extracts also supports that ubiquitin-less IFB-2 double mutation hastens the upregulation of IFB-2 protein levels (**Extended Data Fig. 7a**). We have indicated in the corresponding

Figures when we used whole worm lysates and also made this point clear in the Methods section: “When indicated in the corresponding figure, we also performed analysis of total IFB-2 levels with antibody MH33 in whole *C. elegans* lysates without the centrifugation step”.

To circumvent any interference on the results ensued from a potential higher affinity of MH33 antibody towards the IFB-2 of the highly insoluble fraction, we have now performed western blot using anti-GFP antibody in worms expressing IFB-2 tagged with either CFB or GFP. To this end, we analyzed both the transgenic strain BJ49 (*kcls6[ifb-2p::ifb-2a::CFP]IV*) from Prof. Leube’s lab and also a new strain that expresses endogenous IFB-2 fused to GFP generated by CRISPR/Cas-9. Importantly, the western blots using GFP antibody further validated that IFB-2 protein levels increase with age (please see **Extended Data Fig. 5f-g**).

4. Some lifespan assays have been done with the non-integrated lines DVG9, DVG197, and DVG198. The legends do not specify which panels report assays using these strains.

We apologize for not making more clear which panel reports lifespan assays with the DVG9, DVG197, and DVG198 in our first submission. Lifespan assays with DVG9, DVG197 and DVG198 strains are now presented in **Extended Data Fig. 7b**. The main text now says: “These results indicate that upregulation of IFB-2 levels is sufficient to decrease lifespan, as further supported by experiments with transgenic strains that overexpress IFB-2 (**Extended Data Fig. 7b**)”. We have also specified the used of these specific strains in the corresponding figure legend: “Non-integrated transgenic DVG197 and DVG198 strains overexpressing *ifb-2* under *sur-5* promoter exhibit a short lifespan phenotype compared to the control strain ($P < 0.0001$)”.

5. Related to this issue, the *kcls6* transgene was generated starting from an extrachromosomal array and is thus present in multiple copies. Could this induce aggregation on its own, and would IFB-2::CFP expressed as a single copy be prone to aggregation as well?

It is important to note that the filter trap assay with antibody to IFB-2 (MH33) presented in **Fig. 3e** was performed in wild-type worms. Thus, these data support that endogenous IFB-2 protein aggregates during aging. We have now made clear that these experiments were performed using wild-type worms in the main text, **Fig. 3e** panel and corresponding Figure Legend.

Moreover, we have now tagged endogenous IFB-2 using CRISPR-Cas9 methodology and performed GFP imaging of these worms at different ages. Similar to multi-copy transgenic IFB-2::CFP, endogenous IFB-2 tagged with GFP also showed mislocalization and accumulation into aggregates during aging (please see **Extended Data Fig. 8a**). Moreover, we have performed filter trap assays with antibody to GFP of worms expressing endogenous IFB-2::GFP and confirmed aggregation of IFB-2 protein (**Extended Data Fig. 8b**).

6. I randomly checked the sequences used for qPCR assays taking eps-8 as a test and could not find the position of the forward primer AGAAGAAAAGAAGTGGATTCCGAAC. Please check all sequences mentioned in Table S11.

We have double checked all the primer sequences indicated in Table S11 (now Supplementary Table 12) and confirmed that all of them are correct. It is important to note that EPS-8 has many different isoforms and we designed the qPCR primers for the transcript Y57G11C.24g.1 (please see Figure R3 below), which is the most abundant isoform in C. elegans according to our proteomics data. We apologize for not specifying the specific transcript for EPS-8 in our first submission. We have now included this relevant information in Supplementary Table 12.

```
tttcagccagtc aaatgccatttgcttaaaaaagc gaggaaat tttatcatcagtg aagaagtgt aaaaacatctatt
aaaaatgATGCGTCGAGGTGGATCGATGGGTCCACCGAGCGGGGATCCATATCAGAGTCGCCCATCCCCGGAGG
CTACTACTACAACCGTTCAACGCCTGGTGGTCAGCCAGCTCCATCACCCCTCACACAGTCAACAATCCGCATCTTC
ACATCATCCAAGAGGAGTGCCAATGTCACAACCAATCGCCCGCCGATCGGACTACCGTACTGGAAGTGAACAAAT
GACTCCACGATCCGATCATCGTGGCCCATCGATGGGTACGGTAATGGGGGTCTGTGGATCAACGTGTTCGATGA
CGTCACGCCGTCTACTACGTGGAACATCTTGCCACGTTTGCAGTTGGAAGACAGTTTGGACTCACTTTTCCAGC
CGATGGCATCAGAAAGTTGAAACAAATGGAAGAATTAGCCATTTGGGCGCAACCTCTAATTCTTCGGTTTCG
ACACAACGCAGTGACGGTAGAAGACGACAACGGGAGCTCGTCGAGCAATTTCCGTTGGAATTAATCGAGCAGCC
GACGGCTCACGTGAGCAATGATTCTCGTGAGACTTACAACAATGTGTCCTTTTTGTGTTCGGGAGGACCGGAA
GAGGATGAGCACGCCACCGAGATGCACATTTTCCAGTGTATCCGTGTATCCGCTACCGATGTGGCCGAGGACCT
CAAAAACCTACGTACAAGGTCAATTTCCGTCGTGTGCGTAATGGCCGACGTACCGCCGCGCCGACGCATCTCAAGC
TCAACAACAACAATGCCATTCTACCCACCGGACGACGCTTCAATCAGCAGTGAAACATCTGAAATGTTCGAACG
AGATGTGAATACACTGAATCGTTGTTTCGATGATATCGAACGATTTGTGGCGAGAATTCATCGGCCCACTGGC
TCAGCGAGAAATGAGCAACAGAATCATAGATATCGAACCGGAATCGTCGGGACAAGAAGAACAGCAGCCACC
AGATCCGAATGGCATCTATTTATGAGAGCTCAACTCCATTGGAGTCTGAATTTGTTCGATATTTGAAGAAGTT
CAAGCTCTCCTTTAATCTACTGGCCAAACTCAAAAACCACTCCACGAGCCAAATGCTCCGGAACTTTTGCACCTT
TCTCTTACGCCACTAGCTGTGATCCTTGAAGCATGCCACTGGGGCTCGGAAGAAATGTTGCTCCAACCTGTTGC
CTCGCCGCTGCTCTCGTTGGAAGCTCGTGAGCTCATGCAAAAACCTGTTTGAAGAAGTCAAGAGTGACATTTGGAT
GAGCCTCGGAGAAGCCTGGAGAACTCCACCAGAGGACTGGACAAAACCACTTCCACCACCATATCGTCCAATCTT
CAATGATGGGTTTGGCCCTTACGGGGTCGCCGACAGGCCATGGCAACTCCAAATCAAAATCCATCGTGGACATTC
TGCTCCGCCAGAGCACTTCCGTCAGCCACCGCCAGAGAGAGGAATATGGTGGATACGCTCGAATTCGATCGATT
AACTCTGGAACCGCAGGACTCGAATTTGAAAAGGCAAAGATTATGGAAAGGGAGAGTCTGTTTGAGGCAATGAGGA
GAAGCAGATTGAAGATGAAAAGCGACGAATGCACGCCGAGAAGGATCTCATCACAAAAGAGACAACACAACCAGT
TCCACCACCAGCTGCAGTTCATACATCAACCAATCACCAAGCGATATGATCCACCAATTTCCATCTCTCCACC
ACCACAACGTAACAACCTATTCACACGTGAAGGTTACTGTAGATTCTGACACATCGCCACGTCAGCAGGCATTCAT
TGATGACATCGTGGCAAAAGGTGGCAAGTTGGCACTGGTACCTATGATAGAGGAGGTCAGAATACGAAGGAGCT
GACTGTTTCAAGGGGGAGTATTTGGAGGTTACTTTCGACGAGCGCAACTGGTGGGATGCAAGAATATGCATCA
AAGAGTCGGATACGTTCCACACACAATTTTGTCAATGGTCCATTTGAGCAACAACAATATGCGTCACCAAACCA
TCACAACAATTCCTCATCAACCGGTGGCTACAACAACGGACATCATCAAGGCCAGCGCAAAAACGCTATCGTCC
ACCCCCACCACCCTAGTATCCGATGCAGGTGTCCAGGTCGAAATCCGCCGGGAACATGTTGCTCCGCCACCACC
TCCAGTTGTAATTCACCACCACCACCACCAGTGCGAACCTCAACTATGGAAGAGTTGCTCCGCATGCAACAACA
ACAACAGCAGAAGCAGAGAAAACCACCAGTTGAAGAACCAGTCTACCAACCGCAACCCGCTCAACGTGCCGATT
GGAAATCTGGAGACCGAATCAGAATCCTCAATTGATTCAAGAAGTGACTGAAACTGTGGGACAACACAGTGGAGA
CATTTTAAGACCAGCAATGCAAAGGGCCACTCGAGTGGCAATCAACGAAAAGTCTTCTCCGGAAGACGTGACTCG
TTGGTTACAGGAGAAGGGATTCTACCAAGAGTAATTGACCTATTGGATGGTCAAGATGGTGGCAATCTATTTCTC
CTTGTCAAAATTCATCTGCAACAAGCTTGTGGAAGAGATGAGGGTGGATATTTGTACAGTCAATTTGTTGGTTCA
GAAGAAAAGAAGTGGATTCCGAACTCACACTGGCGATGAATTTGAAAGCAATTTCTCAATCACAGACGAACCCACGT
GGAATTTATCAAATGAAGCTGCAGCTGATGAACCTGTATTCACTATTAACCAATTTCTAGGttttttcaaat ttc
tgtctaacttttccaacgcacaaaaatccaaat ttcacaaaaaaaatattgattttaaatattat ttttcgctaattc
ttttggctagctcaacaataagaaatccccgattttccagatgaatactcattccaaacttactttcaatatgt
tcaat tttactgcctccgtaataatccat tttttgtttcagat ttttgaat tttcaacat ttttaattgaattatgtttc
aaataaatgcttctgcgaata
```

Figure R3. Spliced + UTR sequence of the transcript Y57G11C.24g.1 (EPS-8, isoform g). The forward (5'-AGAAGAAAAGAAGTGGATTCCGAACT-3') and reverse (5'-TTCGTCTGTGATTGAGAATTGCTT-3') primers were designed from the sequences highlighted in red. Source:

Wormbase.org

(https://wormbase.org/species/c_elegans/transcript/Y57G11C.24g.1#06--10).

Reviewer Reports on the First Revision:

Referees' comments:

Referee #2 (Remarks to the Author):

This revised manuscript from Koyuncu et al. is an impressive paper that has grown substantively and aesthetically from the initial submission. The authors are commended for being exceptionally thorough in responding to the reviewer queries, adding rich new experimental data and bringing forth information from previously published resource papers in the C.elegans aging literature. Notable in this submission was their handling and accessibility of the 'omics data, which is much improved (although ideally, the RShiny app would have the data pre-populated rather than asking readers to upload a delimited file). Additionally, their biological exploration of broad spanning deubiquitinase activity during aging revealed a new finding that provides a mechanistic underpinning to this work. Additional work will of course be required to explore this complex phenomena in a tissue specific manner. Nonetheless, the authors have sufficiently addressed my concerns and I believe that the paper is appropriate for publication in Nature.

Referee #3 (Remarks to the Author):

The authors have made a very serious effort to take into consideration the three reviewers' comments. I am satisfied with their reply and changes in response to those I had made, except for a few minor points listed below, and thus suggest that the manuscript could be accepted for publication at Nature.

Minor:

- The authors have modified the residues K524R, K583R and K621R. Since eps-8 generates multiple isoforms, the numbering K524, etc depends on the isoform considered (not to mention that it may be missing from some isoforms). Thus the authors need to specify it.
- The new Extended Data Figure 6/panels cd shows "Western blot analysis of isolated intestine and heads". Neither the legend of that figure nor the Methods section indicates how intestines and heads were "isolated". This is certainly not trivial and is unlikely to generate a lot of material.
- The revised Abstract does not read well and should be reconsidered. Below are sentences that would gain from rewriting (COMMENTS IN CAPITAL LETTERS).

Ubiquitin modifications regulate multiple biological processes¹. **However** [WHY THIS "HOWEVER", THE OPPOSITION SEEMS GRATUITOUS GIVEN THE PREVIOUS SENTENCE], the link between ubiquitination and aging remains unclear. Here we perform quantitative analysis of whole-proteome ubiquitin signatures and find that aging induces a rewiring of the ubiquitin-modified proteome in C. elegans, **a process ameliorated by longevity paradigms** [THE LAST BIT OF THE SENTENCE IS UNCLEAR/ WHICH PROCESS DO YOU MEAN?]. Age-related changes in ubiquitin signatures are mostly linked to downregulated ubiquitination levels across the proteome of different tissues. However, inhibiting elevated deubiquitinase activity prevents these alterations. Since ubiquitin modifications can tag specific proteins for recognition by the proteasome¹, a fundamental question is whether age-associated deficits in targeted degradation

influence longevity. By integrating data from animals with a defective proteasome, we identify proteasomal targets that accumulate with age due to decreased ubiquitination and subsequent degradation. Notably, lowering the levels of distinct age-dysregulated proteasome targets prolongs longevity, whereas preventing their proteasomal degradation shortens lifespan. Among them, we find the intermediate filament, IFB-2, and EPS-8, a modulator of guanine nucleotide exchange activity^{3,4}. While increased levels of IFB-2 promote loss of intestinal integrity, [I WOULD REPEAT "INCREASED LEVELS" HERE BEFORE EPS-8] EPS-8 hyperactivates RAC signaling in muscle cells and neurons leading to dysregulated protein kinase JNK and **the collapse of actin cytoskeleton** [I DISAGREE WITH THE WORD "COLLAPSE" AS FIGURE 5 DOES NOT SHOW A COLLAPSE, BUT RATHER AGGREGATES AND A WIGGLY ORGANISATION OF THE ACTIN]. Thus, age-related changes in targeted degradation of structural and regulatory proteins across tissues determine organismal longevity.

Referee #4 (Remarks to the Author):

The authors have addressed the concerns of Reviewer #1 appropriately and extensively.

In response to the Reviewer #1's first remark, they provided new data on the tissue-specificity of the ubiquitination by analyzing isolated tissues and body parts with Western blotting. Also, they emphasized that their initial analysis on tissue-specificity is based on an integration of their proteomic data with previously published datasets that describe tissue-level gene expression. The new heat maps, as suggested by reviewer #1 (and indirectly by Reviewer #2) have now been added to the manuscript and provide a good overview of the proteomics data.

Also the concern of Reviewer #1 about the functionality of the IFB-2 and EPS-8 ubiquitin-less mutants is properly addressed.

Reviewer #1 rightfully wondered if, in old eps-8 knockdown worms, actin integrity is rescued in tissues other than the body wall muscles. The authors have now provided new evidence that this rescue does not occur in the intestine and the epidermis. Also, the lack of obvious actin aggregates in micrographs of old muscle tissue was questioned by Reviewer #1, while whole-worm filter trap analysis suggested these aggregates should occur. The authors now show that eps-8 knockdown specifically in muscle or neurons clears those aggregates, while in epidermis and intestine it does not. However, they also point to micrographs of old worms in which stained F-actin shows undulations and other deformities. In their rebuttal, the authors enlarge sections of these micrographs and point to the higher actin accumulation in certain regions "that could partially correspond to the aggregated actin detected by filter trap" analysis. I am not convinced by this argument and likely these images are being over-interpreted. All I can observe are undulating actin filaments that are not optically resolved due to close proximity.

The comment of Reviewer #1 on the role of RAC components as regulators of actin homeostasis, as well as the subsequent minor comments, are well-addressed by the authors.

Finally, Reviewer #2 made a short but important remark on the relative roles of a decreased ubiquitin conjugation system versus elevated DUB activity to explain the decrease in ubiquitination in aging worms. This led the authors to conduct a new interesting series of experiments that brings this manuscript to (an even) higher level. Their most important finding was that elevated DUB activity may be a major cause of the global decrease in protein ubiquitination in aged worms. Using a broad-spectrum DUB inhibitor, the authors showed that the age-related decrease of ubiquitination can be reversed. The central theme in this manuscript is the link between decreased protein ubiquitination and, in some cases, the resulting protein accumulation, the subsequent functional deterioration, and its final effect on lifespan. Therefore it is difficult to understand why the authors did not include a simple, low-tech, but crucially important lifespan experiment that

compares controls with animals treated (from adulthood) with the DUB inhibitor PR-619. One would expect that the PR-619 treated animals would be long-lived because many proteins (such as EPS-8 and IFB-2) would not tend to accumulate with age, or at least accumulate at a slower pace. Likely, the authors have run this obvious experiment by now and I encourage them to include it in the manuscript.

Author Rebuttals to First Revision:

Referees' comments:

Referee #2 (Remarks to the Author):

This revised manuscript from Koyuncu et al. is an impressive paper that has grown substantively and aesthetically from the initial submission. The authors are commended for being exceptionally thorough in responding to the reviewer queries, adding rich new experimental data and bringing forth information from previously published resource papers in the *C.elegans* aging literature. Notable in this submission was their handling and accessibility of the 'omics data, which is much improved (although ideally, the RShiny app would have the data pre-populated rather than asking readers to upload a delimited file). Additionally, their biological exploration of broad spanning deubiquitinase activity during aging revealed a new finding that provides a mechanistic underpinning to this work. Additional work will of course be required to explore this complex phenomena in a tissue specific manner. Nonetheless, the authors have sufficiently addressed my concerns and I believe that the paper is appropriate for publication in Nature.

We thank Reviewer #2 for their insightful comments that helped us to significantly improve our manuscript. We will assess how elevated deubiquitinase activity regulates aging in a tissue-specific manner in our follow-up projects.

Referee #3 (Remarks to the Author):

The authors have made a very serious effort to take into consideration the three reviewers' comments. I am satisfied with their reply and changes in response to those I had made, except for a few minor points listed below, and thus suggest that the manuscript could be accepted for publication at Nature.

We thank Reviewer #3 for their thoughtful comments that strengthened our conclusions.

Minor:

- The authors have modified the residues K524R, K583R and K621R. Since eps-8 generates multiple isoforms, the numbering K524, etc depends on the isoform

considered (not to mention that it may be missing from some isoforms). Thus the authors need to specify it.

*We have now specified this in **Extended Data Fig. 1f** and its corresponding legend: “Since EPS-8 has many different isoforms, here we show the ubiquitination sites numbered according to the leading isoform identified in our proteomics data (EPS-8 protein isoform g)”.*

Moreover, we have also provided this information in the Methods section. The text now says: “The strain expressing endogenous EPS-8 K524R/K583R/K621R::3xHA (VDL06, eps-8(syb2901, syb3149)IV) was generated using sg1-GGAGAGTCGTTTGAGGCATGAGG and sg2-AGGAGCTGACTGTTTCAAGGGG. Since eps-8 generates multiple different isoforms, the numbering of ubiquitination sites corresponds to the leading EPS-8 isoform identified in our proteomics data (i.e. transcript Y57G11C.24g.1, EPS-8 protein isoform g)”.

- The new Extended Data Figure 6/panels cd shows “Western blot analysis of isolated intestine and heads”. Neither the legend of that figure nor the Methods section indicates how intestines and heads were “isolated”. This is certainly not trivial and is unlikely to generate a lot of material.

We have now included the protocol for isolation of germlines, intestines and heads in the Methods section. We completely agree with Reviewer #3 that it is necessary to explain this in the manuscript, particularly given the difficulty to obtain enough material of isolated tissues for western blot analysis. As explained in the Methods section, we needed to collect approximately 1000 samples of each isolated tissue to have enough material for western blot analysis. That was possible thanks to one of our experienced postdocs who spent 1 month exclusively dedicated to isolate and collect the different tissues.

- The revised Abstract does not read well and should be reconsidered. Below are sentences that would gain from rewriting (COMMENTS IN CAPITAL LETTERS).

Ubiquitin modifications regulate multiple biological processes¹. **However** [WHY THIS "HOWEVER", THE OPPOSITION SEEMS GRATUITOUS GIVEN THE PREVIOUS SENTENCE], the link between ubiquitination and aging remains unclear. Here we perform quantitative analysis of whole-proteome ubiquitin signatures and find that aging induces a rewiring of the ubiquitin-modified proteome in *C. elegans*, **a process ameliorated by longevity paradigms** [THE LAST BIT OF THE SENTENCE IN UNCLEAR/ WHICH PROCESS DO YOU MEAN?]. Age-related changes in ubiquitin signatures are mostly linked to downregulated ubiquitination levels across the proteome of different tissues. However, inhibiting elevated deubiquitinase activity

prevents these alterations. Since ubiquitin modifications can tag specific proteins for recognition by the proteasome¹, a fundamental question is whether age-associated deficits in targeted degradation influence longevity. By integrating data from animals with a defective proteasome, we identify proteasomal targets that accumulate with age due to decreased ubiquitination and subsequent degradation. Notably, lowering the levels of distinct age-dysregulated proteasome targets prolongs longevity, whereas preventing their proteasomal degradation shortens lifespan. Among them, we find the intermediate filament, IFB-22, and EPS-8, a modulator of guanine nucleotide exchange activity^{3,4}. While increased levels of IFB-2 promote loss of intestinal integrity, [I WOULD REPEAT “INCREASED LEVELS” HERE BEFORE EPS-8] EPS-8 hyperactivates RAC signaling in muscle cells and neurons leading to dysregulated protein kinase JNK and **the collapse of actin cytoskeleton** [I DISAGREE WITH THE WORD “COLLAPSE” AS FIGURE 5 DOES NOT SHOW A COLLAPSE, BUT RATHER AGGREGATES AND A WIGGLY ORGANISATION OF THE ACTIN]. Thus, age-related changes in targeted degradation of structural and regulatory proteins across tissues determine organismal longevity.

We have rewritten the Abstract following the indications from Reviewer #3.

Referee #4 (Remarks to the Author):

The authors have addressed the concerns of Reviewer #1 appropriately and extensively.

We thank Reviewer #4 and Reviewer #1 for their insightful comments that helped us to significantly improve our manuscript.

In response to the Reviewer #1's first remark, they provided new data on the tissue-specificity of the ubiquitination by analyzing isolated tissues and body parts with Western blotting. Also, they emphasized that their initial analysis on tissue-specificity is based on an integration of their proteomic data with previously published datasets that describe tissue-level gene expression.

The new heat maps, as suggested by reviewer #1 (and indirectly by Reviewer #2) have now been added to the manuscript and provide a good overview of the proteomics data.

Also the concern of Reviewer #1 about the functionality of the IFB-2 and EPS-8 ubiquitin-less mutants is properly addressed.

Reviewer #1 rightfully wondered if, in old eps-8 knockdown worms, actin integrity is rescued in tissues other than the body wall muscles. The authors have now provided new evidence that this rescue does not occur in the intestine and the epidermis. Also,

the lack of obvious actin aggregates in micrographs of old muscle tissue was questioned by Reviewer #1, while whole-worm filter trap analysis suggested these aggregates should occur. The authors now show that eps-8 knockdown specifically in muscle or neurons clears those aggregates, while in epidermis and intestine it does not. However, they also point to micrographs of old worms in which stained F-actin shows undulations and other deformities. In their rebuttal, the authors enlarge sections of these micrographs and point to the higher actin accumulation in certain regions “that could partially correspond to the aggregated actin detected by filter trap” analysis. I am not convinced by this argument and likely these images are being over-interpreted. All I can observe are undulating actin filaments that are not optically resolved due to close proximity.

We agree with Reviewer #4. We have now made clear in the text that the aggregation of actin was detected by filter trap: “Filter trap experiments indicated that actin protein also aggregates during aging, whereas knockdown of eps-8 rescued this phenotype (Extended Data Fig. 10e)”.

The comment of Reviewer #1 on the role of RAC components as regulators of actin homeostasis, as well as the subsequent minor comments, are well-addressed by the authors.

Finally, Reviewer #2 made a short but important remark on the relative roles of a decreased ubiquitin conjugation system versus elevated DUB activity to explain the decrease in ubiquitination in aging worms. This led the authors to conduct a new interesting series of experiments that brings this manuscript to (an even) higher level. Their most important finding was that elevated DUB activity may be a major cause of the global decrease in protein ubiquitination in aged worms. Using a broad-spectrum DUB inhibitor, the authors showed that the age-related decrease of ubiquitination can be reversed. The central theme in this manuscript is the link between decreased protein ubiquitination and, in some cases, the resulting protein accumulation, the subsequent functional deterioration, and its final effect on lifespan. Therefore it is difficult to understand why the authors did not include a simple, low-tech, but crucially important lifespan experiment that compares controls with animals treated (from adulthood) with the DUB inhibitor PR-619. One would expect that the PR-619 treated animals would be long-lived because many proteins (such as EPS-8 and IFB-2) would not tend accumulate with age, or at least accumulate at a slower pace. Likely, the authors have run this obvious experiment by now and I encourage them to include it in the manuscript.

We have now assessed whether DUB inhibitor PR-619 prolongs lifespan (please see Extended Data Fig. 3h). Notably, we observed that the treatment with PR-619 at day 9 of adulthood for 24 h or from day 9 of adulthood until the end of the lifespan extends lifespan of wild-type worms.